# Pharmacological hallmarks of allostery at the M4 muscarinic receptor elucidated through structure and dynamics

Ziva Vuckovic[1†], Jinan Wang[2†], Vi Pham[1†], Jesse I Mobbs[1,3†], Matthew J Belousoff[1,3], Apurba Bhattarai[2], Wessel AC Burger[1,3], Geoff Thompson[1], Mahmuda Yeasmin[1], Vindhya Nawaratne[1], Katie Leach[1,3], Emma T van der Westhuizen[1], Elham Khajehali[1], Yi-Lynn Liang[1], Alisa Glukhova[1,3], Denise Wootten[1,3], Craig W Lindsley[4], Andrew Tobin[5], Patrick Sexton[1,3], Radostin Danev[6], Celine Valant[1*], Yinglong Miao[2*], Arthur Christopoulos[1,3,7*], David M Thal[1,3*]

[1]Drug Discovery Biology, Monash Institute of Pharmaceutical Sciences, Monash University, Parkville, Australia; [2]Center for Computational Biology and Department of Molecular Biosciences, University of Kansas, Lawrence, United States; [3]ARC Centre for Cryo-electron Microscopy of Membrane Proteins, Monash Institute of Pharmaceutical Sciences, Monash University, Parkville, Australia; [4]Department of Pharmacology, Warren Center for Neuroscience Drug Discovery and Department of Chemistry, Warren Center for Neuroscience Drug Discovery, Vanderbilt University, Nashville, United States; [5]The Centre for Translational Pharmacology, Advanced Research Centre (ARC), College of Medical, Veterinary and Life Sciences, University of Glasgow, Glasgow, United Kingdom; [6]Graduate School of Medicine, University of Tokyo, Tokyo, Japan; [7]Neuromedicines Discovery Centre, Monash University, Parkville, Australia

*For correspondence:
celine.valant@monash.edu (CV);
miao@ku.edu (YM);
Arthur.Christopoulos@monash.
edu (AC);
david.thal@monash.edu (DMT)

[†]These authors contributed equally to this work

**Abstract** Allosteric modulation of G protein-coupled receptors (GPCRs) is a major paradigm in drug discovery. Despite decades of research, a molecular-level understanding of the general principles that govern the myriad pharmacological effects exerted by GPCR allosteric modulators remains limited. The $M_4$ muscarinic acetylcholine receptor ($M_4$ mAChR) is a validated and clinically relevant allosteric drug target for several major psychiatric and cognitive disorders. In this study, we rigorously quantified the affinity, efficacy, and magnitude of modulation of two different positive allosteric modulators, LY2033298 (LY298) and VU0467154 (VU154), combined with the endogenous agonist acetylcholine (ACh) or the high-affinity agonist iperoxo (Ipx), at the human $M_4$ mAChR. By determining the cryo-electron microscopy structures of the $M_4$ mAChR, bound to a cognate $G_{i1}$ protein and in complex with ACh, Ipx, LY298-Ipx, and VU154-Ipx, and applying molecular dynamics simulations, we determine key molecular mechanisms underlying allosteric pharmacology. In addition to delineating the contribution of spatially distinct binding sites on observed pharmacology, our findings also revealed a vital role for orthosteric and allosteric ligand–receptor–transducer complex stability, mediated by conformational dynamics between these sites, in the ultimate determination of affinity, efficacy, cooperativity, probe dependence, and species variability. There results provide a holistic framework for further GPCR mechanistic studies and can aid in the discovery and design of future allosteric drugs.

## Editor's evaluation

This important work advances our understanding of the structural basis of allosteric modulation of the M4 muscarinic receptor but has broad implications for GPCRs. The evidence supporting the conclusions is exceptional, with multiple cryo-EM structures that are complemented by excellent pharmacological and dynamics studies.

## Introduction

Over the past 40 y, there have been major advances to the analytical methods that allow for the quantitative determination of the pharmacological parameters that characterize G protein-coupled receptor (GPCR) signaling and allosteric modulation (*Figure 1A and B*). These analytical methods are based on the operational model of agonism (*Black and Leff, 1983*) and have been extended or modified to account for allosteric modulation (*Leach et al., 2007*), biased agonism (*Kenakin, 2012*), and even biased allosteric modulation (*Slosky et al., 2021*). Collectively, these models and subsequent key parameters (*Figure 1B*) are used to guide allosteric drug screening, selectivity, efficacy, and ultimately, clinical utility, and provide the foundation for modern GPCR drug discovery (*Wootten et al., 2013*). Yet, a systematic understanding of how these pharmacological parameters relate to the molecular structure and dynamics of GPCRs remains elusive.

The muscarinic acetylcholine receptors (mAChRs) are an important family of five Class A GPCRs that have long served as model systems for understanding GPCR allostery (*Conn et al., 2009*). The mAChRs have been notoriously difficult to exploit therapeutically and selectively due to high-sequence conservation within their orthosteric binding domains (*Burger et al., 2018*). However, the discovery of highly selective positive allosteric modulators (PAMs) for some mAChR subtypes has paved the way for novel approaches to exploit these high-value drug targets (*Chan et al., 2008*; *Gentry et al., 2014*; *Marlo et al., 2009*). X-ray crystallography and cryo-electron microscopy (cryo-EM) have been used to determine inactive state structures for all five mAChR subtypes (*Haga et al., 2012*; *Kruse et al., 2012*; *Thal et al., 2016*; *Vuckovic et al., 2019*) and active state structures of the $M_1$ and $M_2$ mAChRs (*Maeda et al., 2019*). For the $M_2$ mAChR, this includes structures co-bound with the high-affinity agonist iperoxo (Ipx) and the PAM LY2119620 in complex with a G protein mimetic nanobody (*Kruse et al., 2013*) and the transducers $G_o$ (*Maeda et al., 2019*) and β-arrestin1 (*Staus et al., 2020*). These $M_2$ mAChR structures were foundational to validating the canonical mAChR allosteric site but are limited to only one agonist (iperoxo) and one PAM (LY2119620) and do not account for the vast pharmacological properties of ligands targeting mAChRs. A recent nuclear magnetic resonance (NMR) study of the $M_2$ mAChR revealed differences in the conformational landscape of the $M_2$ mAChR when bound to different agonists, but no clear link was established between the properties of the ligands and the conformational states of the receptor (*Xu et al., 2019*). The $M_4$ mAChR subtype is of major therapeutic interest due to its expression in regions of the brain that are rich in dopamine and dopamine receptors, where it regulates dopaminergic neurons involved in cognition, psychosis, and addiction (*Bymaster et al., 2003*; *Dencker et al., 2011*; *Foster et al., 2016*; *Tzavara et al., 2004*). Importantly, these findings have been supported by studies utilizing novel PAMs that are highly selective for the $M_4$ mAChR (*Bubser et al., 2014*; *Chan et al., 2008*; *Leach et al., 2010*; *Suratman et al., 2011*). Among these, LY2033298 (LY298) was the first reported highly selective PAM of the $M_4$ mAChR and displayed antipsychotic efficacy in a preclinical animal model of schizophrenia (*Chan et al., 2008*). Despite LY298 being one of the best characterized $M_4$ mAChR PAMs, its therapeutic potential has been limited by numerous factors, including its chemical scaffold, which has been difficult to optimize with respect to its molecular allosteric parameters (*Figure 1C*) and variability of response between species (*Suratman et al., 2011*; *Wood et al., 2017b*). In the search for better chemical scaffolds, the PAM, VU0467154 (VU154), was subsequently discovered. VU154 showed robust efficacy in preclinical rodent models; however, it also exhibited species selectivity that prevented its clinical translation (*Bubser et al., 2014*). Collectively, LY298 and VU154 are exemplar tool molecules that highlight the promises and the challenges in understanding and optimizing allosteric GPCR drug activity for translational and clinical applications.

Herein, by examining the pharmacology of the PAMs LY298 and VU154 with the agonists ACh and Ipx across radioligand binding assays and two different signaling assays and analyzing these results with modern analytical methods, we determined the key parameters that describe signaling

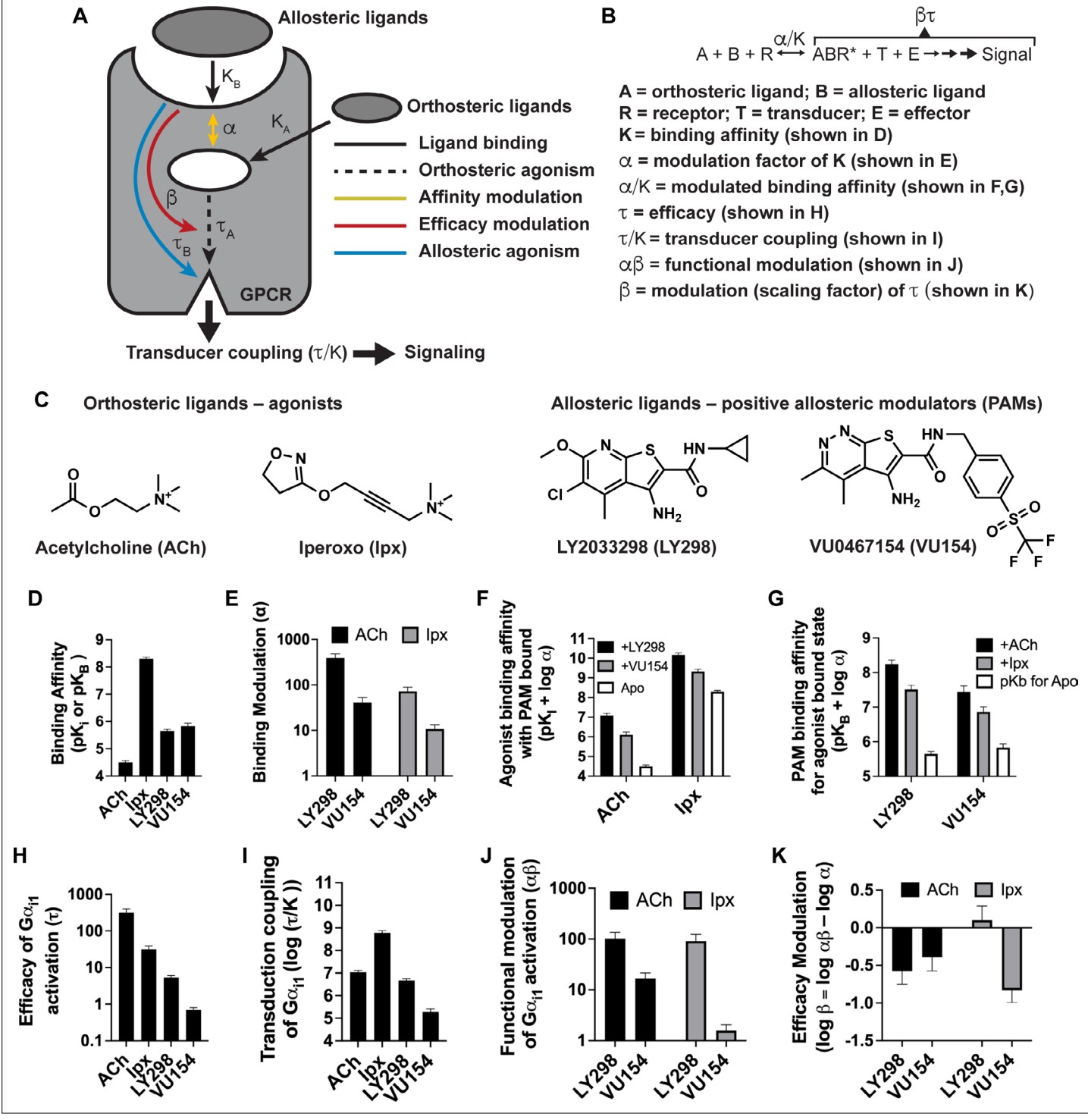

**Figure 1.** Pharmacological characterization of the positive allosteric modulators (PAMs), LY298 and VU154, with acetylcholine (ACh) and iperoxo (Ipx) at the human $M_4$ muscarinic acetylcholine receptor (mAChR). (**A**) Schematic of the pharmacological parameters that define effects of orthosteric and allosteric ligands on a G protein-coupled receptor (GPCR). (**B**) A simplified schematic diagram of the Black–Leff operational model to quantify agonism, allosteric modulation, and agonist bias with pharmacological parameters defined (***Black and Leff, 1983***). (**C**) 2D chemical structures of the orthosteric and allosteric ligands used in this study. (**D–G**) Key pharmacological parameters for interactions between orthosteric and allosteric ligands in [³H]-N-methylscopolamine ([³H]-NMS) binding assays. (**D**) Equilibrium binding affinities ($pK_i$ and $pK_B$) and (**E**) the degree of binding modulation ($\alpha$) between the agonists and PAMs resulting in the modified binding affinities (**F**) $\alpha/K_A$ and (**G**) $\alpha/K_B$. (**H–K**) Key pharmacological parameters relating to $G\alpha_{i1}$ activation for interactions between orthosteric and allosteric ligands measured with the TruPath assay (***Figure 1—figure supplement 1***). (**H**) The signaling efficacy

*Figure 1 continued on next page*

*Figure 1 continued*

($\tau_A$ and $\tau_B$) and (**I**) transduction coupling coefficients (log ($\tau$/K)) of each ligand. (**J**) The functional cooperativity (αβ) between ligands and (**K**) the efficacy modulation (β) between ligands. All data are mean ± SEM of three or more independent experiments performed in duplicate or triplicate with the pharmacological parameters determined using a global fit of the data. The error in (**F, G, K**) was propagated using the square root of the sum of the squares. See *Table 1*. Concentration–response curves are shown in *Figure 1—figure supplement 1*.

The online version of this article includes the following source data and figure supplement(s) for figure 1:

**Source data 1.** Related to *Figure 1D–K*.

**Figure supplement 1.** Concentration–response curves of interactions between the orthosteric ligands (acetylcholine [ACh], iperoxo [Ipx]) and the allosteric ligands (LY298, VU154) at the human M$_4$ muscarinic acetylcholine receptor (mAChR).

**Figure supplement 1—source data 1.** Related to *Figure 1—figure supplement 1*.

**Figure supplement 2.** Pharmacological characterization of the positive allosteric modulators (PAMs), LY298 and VU154, with acetylcholine (ACh) and iperoxo (Ipx) in pERK1/2 signaling assays.

**Figure supplement 2—source data 1.** Related to *Figure 1—figure supplement 2*.

and allostery for these ligands. To investigate a structural basis for these pharmacological parameters, we used cryo-EM to determine high-resolution structures of the M$_4$ mAChR in complex with a cognate G$_{i1}$ heterotrimer and ACh and Ipx. We also determined structures of receptor complexes with Ipx co-bound with the PAMs LY298 or VU154. Moreover, because protein allostery is a dynamic process (*Changeux and Christopoulos, 2016*), we performed all-atom simulations using the Gaussian accelerated molecular dynamics (GaMD) enhanced sampling method (*Draper-Joyce et al., 2021*; *Miao et al., 2015*; *Wang et al., 2021a*) on the M$_4$ mAChR using the cryo-EM structures. The structures and GaMD simulations, in combination with detailed molecular pharmacology and receptor mutagenesis experiments, provide fundamental insights into the molecular mechanisms underpinning the hallmarks of GPCR allostery. To further validate these findings, we investigated the differences in the selectivity of VU154 between the human and mouse receptors and established a structural basis for species selectivity. Collectively, these results will enable future GPCR drug discovery research and potentially lead to the development of next generation M$_4$ mAChR PAMs.

## Results

### Pharmacological characterization of M$_4$ mAChR PAMs with ACh and Ipx

The pharmacology of LY298 or VU154 interacting with ACh has been well characterized in binding and functional assays at the M$_4$ mAChR (*Bubser et al., 2014*; *Chan et al., 2008*; *Gould et al., 2016*; *Leach et al., 2010*; *Suratman et al., 2011*; *Thal et al., 2016*). However, their pharmacology with Ipx has not been reported. Therefore, we characterized both PAMs with ACh and Ipx in binding and in two different functional assays to provide a thorough foundational comparative characterization of the pharmacological parameters of these ligands from the same study.

We first used radioligand binding assays (*Figure 1—figure supplement 1A*) to determine the *binding affinities* (i.e., equilibrium dissociation constants) of ACh and Ipx (K$_A$) for the orthosteric site and of LY298 and VU154 (K$_B$) for the allosteric site of the unoccupied human M$_4$ mAChR (*Figure 1D*), along with the degree of *binding cooperativity* (α) between the agonists and PAMs when the two are co-bound (*Figure 1E*). Analysis of these experiments revealed that LY298 and VU154 have very similar binding affinities for the allosteric site with values (expressed as negative logarithms; pK$_B$) of 5.65 ± 0.07 and 5.83 ± 0.12, respectively (*Table 1*), in accordance with previous studies (*Bubser et al., 2014*; *Leach et al., 2011*). Both PAMs potentiated the binding affinity of ACh and Ipx (*Figure 1E*), with the effect being greatest between LY298 and ACh (~400-fold increase in binding affinity). Comparatively, the positive cooperativity between VU154 and ACh was only 40-fold. When Ipx was used as the agonist, the binding affinity modulation mediated by both PAMs was more modest, characterized by an approximately 72-fold potentiation for the combination of Ipx and LY298, and 10-fold potentiation for the combination of Ipx and VU154. These results indicate *probe-dependent* effects (*Valant et al., 2012*) with respect to the ability of either PAM to modulate the affinity of each agonist (*Figure 1F and G*). A probe-dependent effect was also observed with the radioligand, [³H]-NMS, evidenced by a reduction in specific radioligand binding due to negative cooperativity between the antagonist probe and LY298, which has been previously reported (*Chan et al., 2008*; *Leach et al., 2010*; *Suratman*

**Table 1.** Pharmacological parameters from radioligand binding and functional experiments.

[³H]-NMS saturation binding on stable M₄ mAChR CHO cells

| Constructs | Sites per cell* | pK_D[†] |
|---|---|---|
| Human WT M₄ mAChR | 598,111 ± 43,067 (7) | 9.76 ± 0.05 (7) |
| Mouse WT M₄ mAChR | 21,027 ± 2188 (3) | 9.76 ± 0.05 (3) |
| Human D432E M₄ mAChR | 126,377 ± 10,066 (3) | 9.60 ± 0.07 (3) |
| Human T433R M₄ mAChR | 157,442 ± 36,658 (6) | 9.64 ± 0.09 (6) |
| Human V91L, D432E, T433R M₄ mAChR | 205,771 ± 20,975 (4) | 9.58 ± 0.08 (4) |

[3H]-NMS interaction binding assays between ACh or Ipx and LY298 or VU154 on stable M₄ mAChR constructs in Flp-In CHO cells

| Constructs | PAM | pK_i ACh [‡] | pK_i Ipx [‡] | pK_B PAM [‡] | log α_ACh [§] | log α_Ipx [§] |
|---|---|---|---|---|---|---|
| Human WT M₄ mAChR | LY298 | 4.50 ± 0.06 (4) | 8.30 ± 0.06 (4) | 5.65 ± 0.07 (8) [¶] | 2.59 ± 0.10 (4) | 1.86 ± 0.10 (4) |
| | VU154 | 4.40 ± 0.09 (4) | 8.19 ± 0.06 (8) | 5.83 ± 0.11 (12) [¶] | 1.61 ± 0.13 (4) | 1.03 ± 0.10 (8) |
| Mouse WT M₄ mAChR | LY298 | 4.52 ± 0.07 (4) | 8.55 ± 0.06 (4) | 5.74 ± 0.07 (8) [¶] | 1.78 ± 0.10 (4) | 1.30 ± 0.11 (4)* |
| | VU154 | 4.59 ± 0.06 (4) | 8.57 ± 0.06 (3) | 6.07 ± 0.09 (7) [¶] | 2.43 ± 0.10 (4) | 1.75 ± 0.12 (3)* |
| Human D432E M₄ mAChR | LY298 | N.T. | 8.28 ± 0.04 (5) | 5.86 ± 0.07 (5) | N.T. | 1.59 ± 0.06 (5) |
| | VU154 | N.T. | 8.27 ± 0.06 (6) | 6.21 ± 0.12 (6) | N.T. | 1.04 ± 0.09 (6) |
| Human T433R M₄ mAChR | LY298 | N.T. | 8.05 ± 0.08 (5) | 5.04 ± 0.04 (5)* | N.T. | 1.91 ± 0.11 (5) |
| | VU154 | N.T. | 7.88 ± 0.04 (5) | 5.50 ± 0.08 (5) | N.T. | 1.67 ± 0.07 (5)* |
| Human V91L, D432E, T433R M₄ mAChR | LY298 | N.T. | 7.95 ± 0.10 (4) | 5.29 ± 0.26 (4) | N.T. | 1.80 ± 0.22 (4) |
| | VU154 | N.T. | 7.89 ± 0.12 (4) | 6.34 ± 0.16 (4)* | N.T. | 1.35 ± 0.16 (4) |

Gα_{i1} activation (TruPath) interaction assays between ACh or Ipx and LY298 or VU154 on transiently expressed M₄ mAChR constructs in HEK293A cells

| Constructs | PAM | log τ ACh** | log τ Ipx** | pK_B PAM [‡] | log τ PAM** | log αβ_ACh [††] | log αβ_Ipx [††] |
|---|---|---|---|---|---|---|---|
| Human WT M₄ mAChR | LY298 | | | = 5.65 | 1.02 ± 0.03 (8) [¶] | 2.01 ± 0.14 (4) | 1.96 ± 0.16 (4) |
| | VU154 | 2.71 ± 0.14 (4) | 1.49 ± 0.12 (4) | = 5.83 | –0.55 ± 0.08 (8) [¶] | 1.22 ± 0.13 (4) | 0.20 ± 0.13 (4) |

pERK1/2 interaction assays between ACh or Ipx and LY298 or VU154 on stable M₄ mAChR constructs in Flp-In CHO cells

| Constructs | PAM | log τ ACh** | log τ Ipx** | pK_B PAM [‡] | log τ_C PAM [‡‡] | log αβ_ACh [††] | log αβ_Ipx [††] |
|---|---|---|---|---|---|---|---|
| Human WT M₄ mAChR | LY298 | | | = 5.65 | 1.19 ± 0.05 (12)** | 2.29 ± 0.22 (4) | 1.08 ± 0.28 (8) |
| | VU154 | 3.27 ± 0.06 (8) [¶] | 1.74 ± 0.03 (16) [¶] | = 5.83 | 0.11 ± 0.05 (12)** | 0.88 ± 0.23 (4) | 0.66 ± 0.15 (8) |
| Mouse WT M₄ mAChR | LY298 | N.T. | N.D. | = 5.74 | 1.32 ± 0.07 (5) | N.T. | 1.24 ± 0.12 (4) |
| | VU154 | N.T. | N.D. | = 6.07 | 1.47 ± 0.08 (5) [§§] | N.T. | 2.08 ± 0.15 (5) [§§] |
| Human D432E M₄ mAChR | LY298 | N.T. | N.D. | = 5.86 | 1.34 ± 0.08 (5) | N.T. | 1.37 ± 0.28 (5) |
| | VU154 | N.T. | N.D. | = 6.21 | 0.78 ± 0.08 (5) [§§] | N.T. | 1.02 ± 0.15 (5) |
| Human T433R M₄ mAChR | LY298 | N.T. | N.D. | = 5.04 | 1.73 ± 0.13 (5) [§§] | N.T. | 1.85 ± 0.28 (5) |
| | VU154 | N.T. | N.D. | = 5.50 | 0.95 ± 0.12 (5) [§§] | N.T. | 1.18 ± 0.14 (5) |
| Human V91L, D432E, T433R M₄ mAChR | LY298 | N.T. | N.D. | = 5.29 | 1.62 ± 0.09 (5) [§§] | N.T. | 1.64 ± 0.30 (5) |
| | VU154 | N.T. | N.D. | = 6.34 | 0.68 ± 0.06 (5) [§§] | N.T. | 1.34 ± 0.11 (5) [§§] |

Values represent the mean ± SEM with the number of independent experiments shown in parenthesis.

N.T.: not tested; N.D.: not determined; Ach, acetylcholine; Ipx: iperoxo; PAM: positive allosteric modulator.

*Number of [³H]-NMS binding sites per cell.

[†]Negative logarithm of the radioligand equilibrium dissociation constant.

[‡]Negative logarithm of the orthosteric (pK_i) or allosteric (pK_B) equilibrium dissociation constant.

[§]Logarithm of the binding cooperativity factor between the agonist (ACh or Ipx) and the PAM (LY298 or VU154).

[¶]Parameter was determined in a shared global analysis between agonists.

**Logarithm of the operational efficacy parameter determined using the Operational Model of Agonism.

[††]Logarithm of the functional cooperativity factor between the agonist (ACh or Ipx) and the PAM (LY298 or VU154).

[‡‡]log τ_C = logarithm of the operational efficacy parameter corrected for receptor expression (methods in Appendix 1).

[§§]Values from pK_B PAM, log α_Ipx, log τ_C PAM, and log αβ_Ipx that are significantly different from human WT M₄ mAChR (p<0.05) calculated by a one-way ANOVA with a Dunnett's post-hoc test.

*et al., 2011*; *Thal et al., 2016*). It is important to note that binding affinity modulation is thermodynamically reciprocal at equilibrium, and the affinities of LY298 and VU154 were thus also increased in the agonist bound state (*Figure 1—figure supplement 1A*). This results in LY298 having a fivefold higher binding affinity than VU154 when agonists are bound (*Table 1*).

We subsequently used the BRET-based TruPath assay (*Olsen et al., 2020*) as a proximal measure of G protein activation with $G\alpha_{i1}$ (*Figure 1—figure supplement 1B*). We also used a more amplified downstream signaling assay, extracellular signal-regulated kinases 1/2 phosphorylation (pERK1/2), that is also dependent on $G_i$ activation (*Figure 1—figure supplement 2A*), to measure the cell-based activity of each PAM with each agonist. These signaling assays allowed us to determine the *efficacy* of the agonists ($\tau_A$) and the PAMs ($\tau_B$) (*Figure 1H*, *Figure 1—figure supplement 2B*). Importantly, efficacy ($\tau$), as defined from the Black–Leff operational model of agonism (*Black and Leff, 1983*), is determined by the ability of an agonist to promote an active receptor conformation, the receptor density ($B_{max}$), and the subsequent ability of a cellular system to generate a response (*Figure 1B*). Notably, in both signaling assays, the rank order of efficacy was ACh > Ipx > LY298 > VU154. We subsequently calculated the *transducer coupling coefficient* ($\tau$/K) (*Figure 1I*, *Figure 1—figure supplement 2C*), a parameter often used as a starting point to quantify biased agonism (*Kenakin et al., 2012*) and that is specific to the intact cellular environment in which a given response occurs. Thus the dissociation constant (K) in the transduction coefficient subsumes the affinity for the ground state (non-bound) receptor, in addition to any isomerization states of the receptor that ultimately yield cellular responses (*Kenakin and Christopoulos, 2013*). Consequently, in both assays, the rank order of transducer coupling was Ipx >> ACh ~ LY298 > VU154 due to Ipx having a higher binding affinity for the receptor. Overall, these results indicate that although ACh is a more efficacious agonist than Ipx, it has lower transducer coupling coefficient. In contrast, LY298 has both better efficacy and transducer coupling coefficient than VU154 (*Table 1*).

The signaling assays and use of an operational model of allosterism also allowed for the determination of the *functional cooperativity* (αβ) exerted by the PAMs (*Figure 1J*, *Figure 1—figure supplement 2D*), which is a composite parameter accounting for both binding (α) and efficacy (β) modulation. Notably, VU154 displayed lower positive functional cooperativity with ACh than LY298. Strikingly, VU154 had negligible functional modulation with Ipx, in contrast to the cooperativity observed with ACh in the TruPath assay. The tenfold difference in αβ values for VU154 between ACh and Ipx highlights the dependence of the orthosteric probe used in the assay (i.e. probe dependence); on this basis, VU154 would be classified as a *'neutral' allosteric ligand* (not a PAM) with Ipx in the TruPath assay, that is, VU154 still binds to the allosteric site, but displays neutral cooperativity (αβ = 1) with Ipx (*Table 1*).

The degree of *efficacy modulation* (β) that the PAMs have on the agonists can be calculated by subtracting the binding modulation (α) from the functional modulation (αβ) (*Figure 1K*, *Figure 1—figure supplement 2E*). A caveat of this analysis is that errors for β are higher due to the error being propagated between calculations. Ideally, the degree of efficacy modulation would be determined in an experimental system where the maximal efficacy of system is not reached by the agonists alone (*Berizzi et al., 2016*). Nevertheless, our analysis shows the PAMs LY298 and VU154 appear to have a slight negative to neutral effect on agonist efficacy in the $G_{i1}$ TruPath and pERK1/2 assays (*Table 1*), suggesting that the predominant allosteric effect exerted by these PAMs is mediated through modulation of binding affinity.

Collectively, our extensive analysis on the pharmacology of LY298 and VU154 with ACh and Ipx offers detailed insight into the key differences between these ligands across a range of pharmacological properties: ligand binding, probe dependence, efficacy, agonist–receptor–transducer interactions, and allosteric modulation (*Figure 1*, *Table 1*). We hypothesized that structures of the human $M_4$ mAChR in complex with different agonists and PAMs combined with molecular dynamic simulations could provide high-resolution molecular insights into the different pharmacological profiles of these ligands.

## Determination of $M_4R$-$G_{i1}$ complex structures

Similar to the approach used in prior determination of active-state structures of the $M_1$ and $M_2$ mAChRs (*Maeda et al., 2019*), we used a human $M_4$ mAChR construct that lacked residues 242–387 of the third intracellular loop to improve receptor expression and purification, and made complexes of the

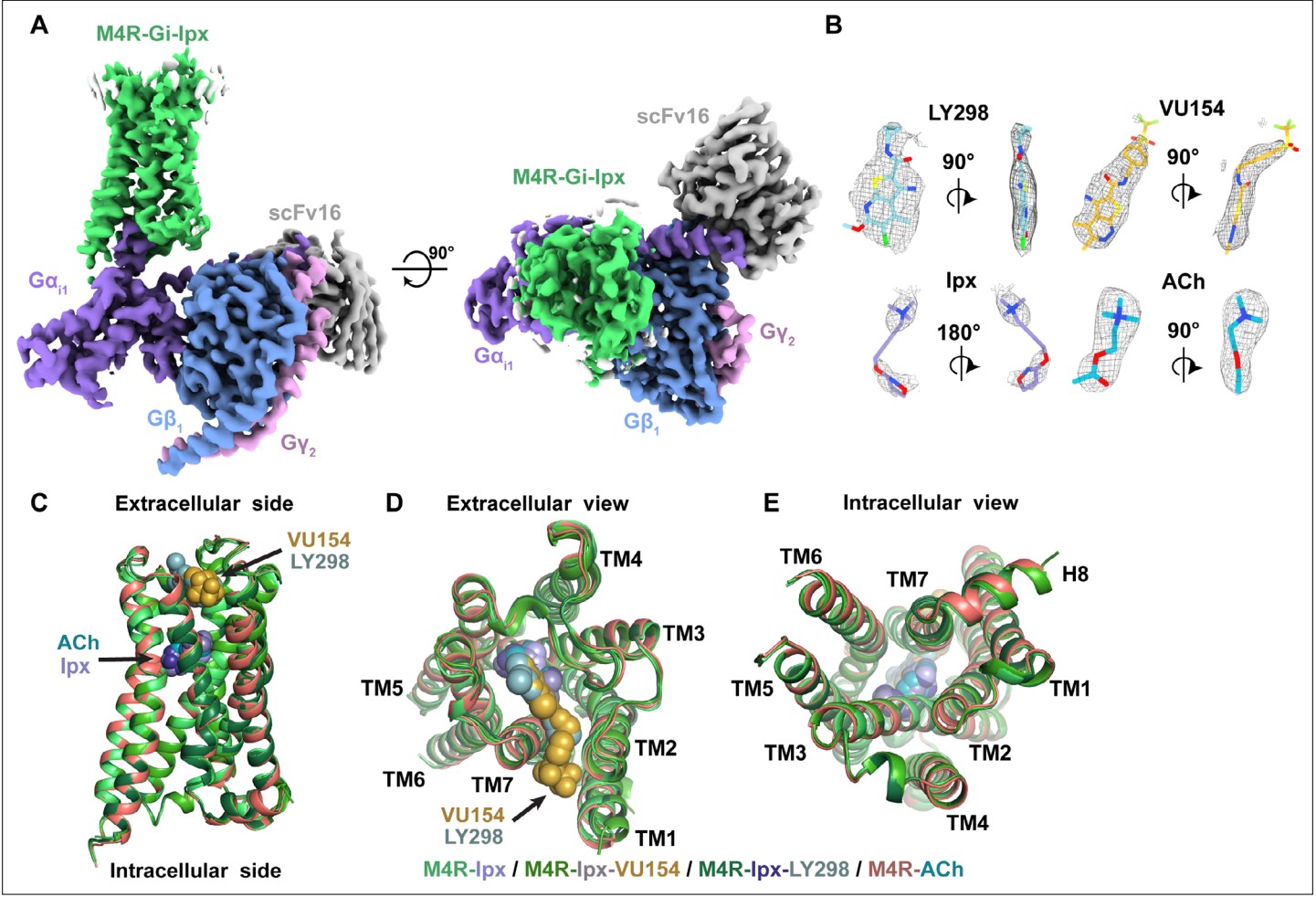

**Figure 2.** Cryo-electron microscopy (cryo-EM) structures of the $M_4R$-$G_{i1}$-scFv16 complexes. (**A**) Cryo-EM maps of Ipx-bound $M_4R$-$G_{i1}$-scFv16 complex with views from the membrane and the extracellular surface. Cryo-EM maps of the other ligand-bound structures are shown in *Figure 2—figure supplement 1*. (**B**) Representative EM density around the ligands in this study. EM-maps of Ipx-, LY298-Ipx-, and VU154-Ipx were set to a contour level of 0.011 and the receptor-focused map of ACh- was set to 0.32. (**C–E**) Comparison of the receptor models with bound ligands and views from the (**C**) membrane, (**D**) extracellular surface, and (**E**) intracellular surface.

The online version of this article includes the following figure supplement(s) for figure 2:

**Figure supplement 1.** Cryo-electron microscopy (cryo-EM) structures of the $M_4R$-$G_{i1}$-scFv16 complexes.

**Figure supplement 2.** Cryo-electron microscopy (cryo-EM) data processing and analysis.

**Figure supplement 3.** Cryo-electron microscopy (cryo-EM) density maps.

**Figure supplement 4.** Comparison of active state muscarinic acetylcholine receptor (mAChR) structures.

**Figure supplement 5.** Comparison of active state $M_4$ muscarinic acetylcholine receptor (mAChR) structures.

receptor with $G_{i1}$ protein and either the endogenous agonist, ACh, or Ipx. Due to the higher affinity of Ipx compared to ACh (*Schrage et al., 2013*), we utilized Ipx to form additional $M_4R$-$G_{i1}$ complexes with or without the co-addition of either LY298 or VU154. In all instances, complex formation was initiated by combining purified $M_4$ mAChR immobilized on anti-FLAG resin with detergent solubilized $G_{i1}$ membranes, a single-chain variable fragment (scFv16) that binds $G\alpha_i$ and $G\beta$, and the addition of apyrase to remove guanosine 5'-diphosphate (*Maeda et al., 2018*). For this study, we used a $G_{i1}$ heterotrimer composed of a dominant negative form of human $G\alpha_{i1}$, and human $G\beta_1$ and $G\gamma_2$. (*Liang et al., 2018b*). Vitrified samples of each complex were imaged using conventional cryo-TEM on a Titan Krios microscope (*Danev et al., 2021*).

The structures of ACh-, Ipx-, LY298-Ipx-, and VU154-Ipx-bound $M_4R$-$G_{i1}$ complexes were determined to resolutions of 2.8, 2.8, 2.4, and 2.5 Å, respectively (*Figure 2A*, *Figure 2—figure supplement*

**Table 2.** Cryo-electron microscopy (cryo-EM) data collection, refinement, and validation statistics.

| | M4R-G$_{i1}$-Ipx | M4R-G$_{i1}$-Ipx-LY298 | M4R-G$_{i1}$-Ipx-VU154 | M4R-G$_{i1}$-ACh |
|---|---|---|---|---|
| **Data collection & refinement** | | | | |
| EMD code | 26,099 | 26,100 | 26,101 | 26,102 |
| Micrographs | 5056 | 5121 | 6021 | 5913 |
| Electron dose (e$^-$/A$^2$) | 66 | 66 | 59.5 | 53.6 |
| Voltage (kV) | 300 | 300 | 300 | 300 |
| Pixel size (Å) | 0.83 | 0.83 | 0.83 | 0.83 |
| Spot size | | | | |
| Exposure time | 4 | 4 | 3 | 5 |
| Movie frames | 76 | 76 | 75 | 71 |
| K3 CDS mode | No | No | No | Yes |
| Defocus range (μm) | 0.5–1.5 | 0.5–1.5 | 0.5–1.5 | 0.5–1.5 |
| Symmetry imposed | C1 | C1 | C1 | C1 |
| Particles (final map) | 415,743 | 617,793 | 677,392 | 315,595 |
| Resolution @0.143 FSC (Å) | 2.8 | 2.4 | 2.5 | 2.8 |
| Refinement | | | | |
| CC$_{map-model}$ | 0.87 | 0.87 | 0.88 | 0.82 |
| Map sharpening B factor (Å$^2$) | –80.9 | –60.8 | –46.6 | –85.1 |
| **Model quality** | | | | |
| PDB code | 7TRK | 7TRP | 7TRQ | 7TRS |
| R.M.S. deviations | | | | |
| Bond length (Å) | 0.004 | 0.004 | 0.005 | 0.006 |
| Bond angles (°) | 0.849 | 0.811 | 0.826 | 0.773 |
| Ramachandran | | | | |
| Favored (%) | 98.38 | 99.14 | 98.02 | 98.10 |
| Outliers (%) | 0 | 0 | 0 | 0 |
| Rotamer outliers (%) | 0.11 | 0.21 | 0 | 0 |
| C-beta deviations (%) | 0 | 0 | 0 | 0 |
| Clashscore | 2.69 | 2.62 | 2.26 | 4.08 |
| MolProbity score | 1.06 | 1.05 | 1.00 | 1.19 |

mAChR: muscarinic acetylcholine receptor; ACh: acetylcholine; Ipx: iperoxo; FSC: Fourier shell correlation.

*1*, *Table 2*). For the ACh-bound M$_4$R-G$_{i1}$ complex, an additional focus refinement yielded an improved map of the receptor and binding site (2.75 Å) for modeling (*Figure 2—figure supplements 2 and 3*). The cryo-EM density maps for all complexes were sufficient for confident placement of backbone and sidechains for most of the receptor, G$_{i1}$, and scFv16, and the bound ligands with exception of the alkyne bond of Ipx, which was consistent with prior cryo-EM studies (*Maeda et al., 2019*; *Figure 2B*, *Figure 2—figure supplement 3*).

In all four structures, EM density beyond the top of transmembrane helix 1 (TM1) and the third intracellular loop (ICL3) of the receptor was poorly observed and not modeled. Similarly, the EM density of the α-helical domain of Gα$_{i1}$ was poor and not modeled. These regions are highly dynamic and typically not modeled in many class A GPCR-G protein complex structures. Apart from these

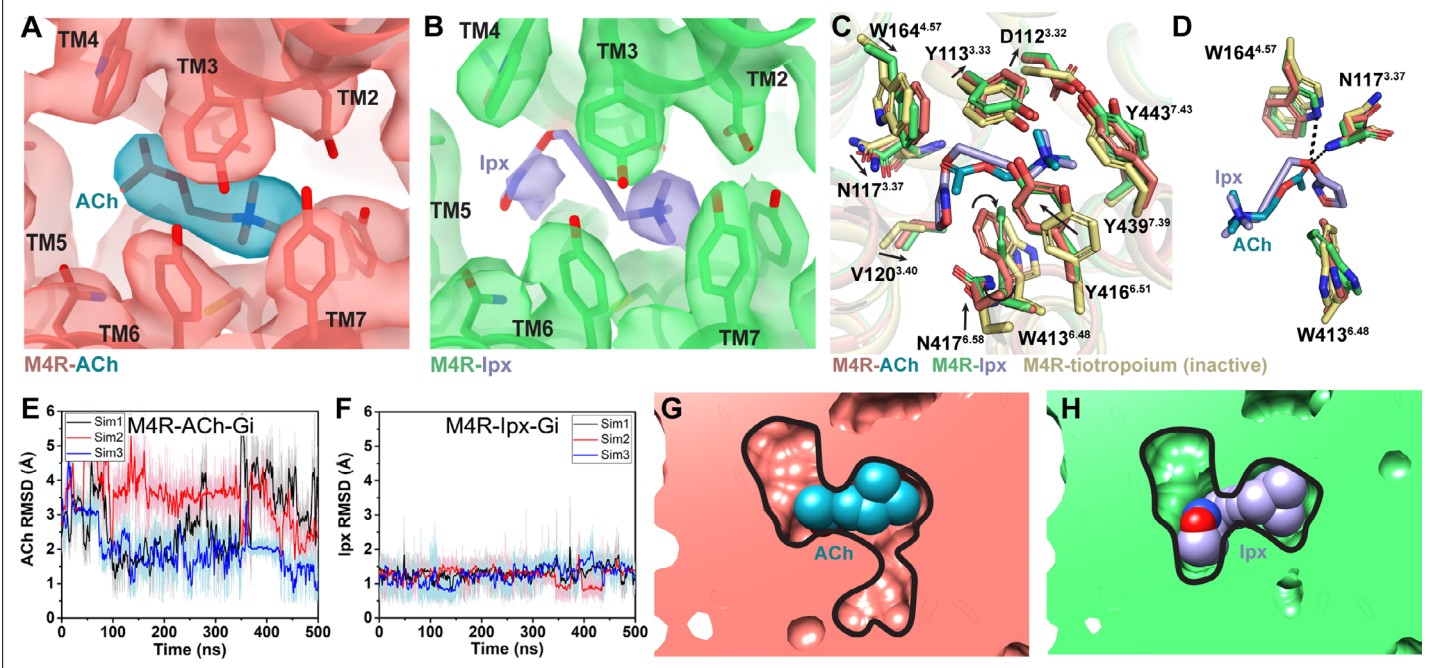

**Figure 3.** Interactions of acetylcholine (ACh) and iperoxo (Ipx) with the receptor. (**A, B**) Cryo-electron microscopy (cryo-EM) density of the (**A**) ACh- and (**B**) Ipx-bound structures. (**C, D**) Interactions at the orthosteric binding site comparing the active state ACh- and Ipx-bound structures with the inactive state tiotropium-bound structure (PDB: 5DSG). Arrows denote relative movement of residues between the inactive and active states. (**D**) Detailed interactions of ACh and Ipx. Hydrogen bonds are shown as black dashed lines. (**E, F**) Time courses from Gaussian accelerated molecular dynamics (GaMD) simulations of the ACh- and Ipx- bound $M_4R$-$G_{i1}$ cryo-EM structures, each performed with three separate replicates. Individual replicate simulations are illustrated with different colors. The heading of each plot refers to the specific model used in the simulations. Root mean square deviations (RMSDs) of (**E**) ACh and (**F**) Ipx from simulations of the cryo-EM structures. (**G, H**) Cross-sections through the ACh- and Ipx-bound structures denoting the relative size of the binding pockets outlined in black.

The online version of this article includes the following figure supplement(s) for figure 3:

**Figure supplement 1.** Interactions of acetylcholine (ACh) and iperoxo (Ipx) with the receptor measured during Gaussian accelerated molecular dynamics (GaMD) simulations.

regions, most amino acid side chains were well resolved in the final EM density maps (*Figure 2—figure supplement 3*).

## Structure and dynamics of agonist binding

Recently, cryo-EM structures of $M_4R$-$G_{i1}$ complexes bound to Ipx, Ipx, and the PAM, LY2119620, and a putative novel allosteric agonist, c110, were determined (*Wang et al., 2022*). Surprisingly, comparison of the $M_4R$-$G_{i1}$ complex structures revealed larger differences in the position of key orthosteric and allosteric site residues than the $M_1R$-$G_{11}$ and $M_2R$-$G_{oA}$ complex structures (*Figure 2—figure supplement 4*). Unfortunately, the quality of density in the EM maps around the orthosteric and allosteric sites of these $M_4R$-$G_{i1}$ structures (*Wang et al., 2022*) was poor, resulting in several key residues being mismodeled in each site (*Figure 2—figure supplement 5*). Therefore, differences between the $M_4R$-$G_{i1}$ structures described herein and those by *Wang et al., 2022* are highly likely to not be due to genuine differences and, as such, we compared the prior $M_1R$-$G_{11}$ and $M_2R$-$G_{oA}$ complex structures (*Maeda et al., 2019*) in this study.

Overall, our $M_4R$-$G_{i1}$ complex structures are similar in architecture to that of other activated class A GPCRs, including the $M_1R$-$G_{11}$ and $M_2R$-$G_{oA}$ complexes (*Figure 2—figure supplement 4*). Superposition of the $M_4R$-$G_{i1}$ complexes revealed nearly identical structures with root mean square deviations (RMSD) of 0.4–0.5 Å for the full complexes and 0.3–0.4 Å for the receptors alone (*Figure 2C*). The largest differences occur around the extracellular surface of the receptors (*Figure 2D*) along with slight displacements in the position of the αN helix of $Gα_{i1}$ and $Gβ_1$, $Gγ_2$, and scFv16 with respect to the receptor (*Figure 2—figure supplement 1D*). The EM density of side chains surrounding the ACh and

**Table 3.** Gaussian accelerated molecular dynamics (GaMD) simulations of the $M_4$ muscarinic acetylcholine receptor (mAChR).

| System | Method |
|---|---|
| M4-$G_{i1}$-Ipx (cryo-EM structure) | GaMD (3 × 500 ns) |
| M4-$G_{i1}$-Ipx-VU154 (cryo-EM structure) | GaMD (3 × 500 ns) |
| M4-$G_{i1}$-Ipx-LY298 (cryo-EM structure) | GaMD (3 × 500 ns) |
| M4-$G_{i1}$-ACh (cryo-EM structure) | GaMD (3 × 500 ns) |
| M4-D432E-$G_{i1}$-Ipx-VU154 | GaMD (3 × 500 ns) |
| M4-T433R-$G_{i1}$-Ipx-VU154 | GaMD (3 × 500 ns) |
| M4-$G_{i1}$-ACh-VU154 | GaMD (3 × 500 ns) |
| M4-$G_{i1}$-ACh-LY298 | GaMD (3 × 500 ns) |
| M4-$G_{i1}$-VU154 | GaMD (3 × 500 ns) |
| M4-$G_{i1}$-LY298 | GaMD (3 × 500 ns) |
| M4-VU154 | GaMD (3 × 1000 ns) |
| M4-LY298 | GaMD (3 ×1000 ns) |

Ipx binding sites (*Figure 3A and B*) was well resolved providing the opportunity to understand structural determinants of orthosteric agonist binding. The orthosteric site of the $M_4$ mAChR, in common with the other mAChR subtypes, is buried within the TM bundle in an aromatic cage that is composed of four tyrosine residues, two tryptophan residues, one phenylalanine residue, and seven other polar and nonpolar residues (*Figure 3C*). Notably, all 14 of these residues are absolutely conserved across all five mAChR subtypes, underscoring the difficulty in developing highly subtype-selective orthosteric agonists (*Burger et al., 2018*). Both ACh and Ipx have a positively charged trimethyl ammonium ion that makes cation-π interactions with Y113$^{3.33}$, Y416$^{6.51}$, Y439$^{7.39}$, and Y443$^{7.43}$ (*Figure 3C*; superscript refers to the Ballesteros and Weinstein scheme for conserved class A GPCR residues; *Ballesteros and Weinstein, 1995*). Likewise, both ACh and Ipx have a polar oxygen atom that can form a hydrogen bond to the indole nitrogen of W164$^{4.57}$ with the oxygen of Ipx also being in position to interact with the backbone of N117$^{3.37}$ (*Figure 3D*). Mutation of any of these contact residues reduces the affinity of ACh, validating their importance for agonist binding (*Leach et al., 2011*; *Thal et al., 2016*). The largest chemical difference between ACh and Ipx is the bulkier heterocyclic isoazoline group of Ipx that makes a π-π interaction with the conserved residue W413$^{6.48}$ (*Figure 3D*). The residue W413$^{6.48}$ is part of the CWxP motif, also known as the rotamer toggle switch, a residue that typically undergoes a change in rotamer between the inactive and active states of class A GPCRs (*Shi et al., 2002*).

To investigate the structural dynamics of the $M_4$ mAChR, we performed three independent 500 ns GaMD simulations on the ACh- and Ipx-bound $M_4$R-$G_{i1}$ cryo-EM structures (*Table 3*). GaMD simulations revealed that ACh undergoes higher fluctuations in the orthosteric site than Ipx (*Figure 3E and F*, *Videos 1 and 2*). Similarly, the interactions of N117$^{3.37}$, W164$^{4.57}$, and W413$^{6.48}$

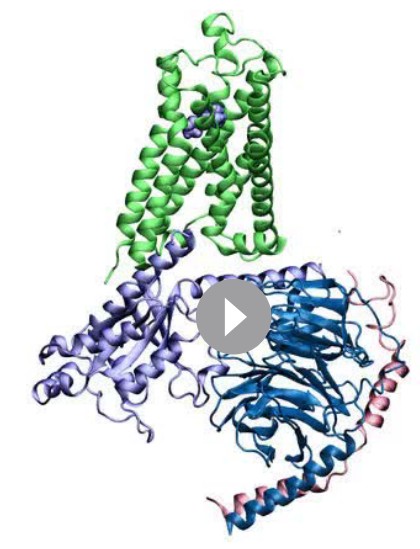

time:  0 ns

**Video 1.** Movie from one Ipx-$M_4$R-$G_{i1}$ Gaussian accelerated molecular dynamics (GaMD) simulation. https://elifesciences.org/articles/83477/figures#video1

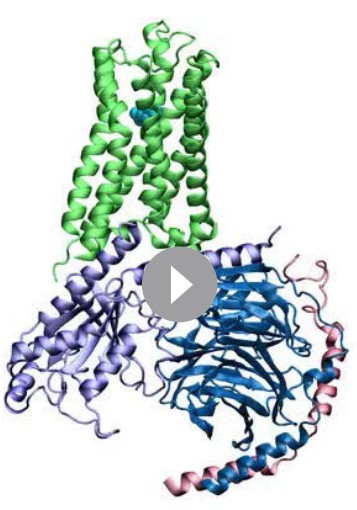

time: 0 ns

**Video 2.** Movie from one ACh-$M_4$R-$G_{i1}$ Gaussian accelerated molecular dynamics (GaMD) simulation.
https://elifesciences.org/articles/83477/figures#video2

with Ipx were more stable than those with ACh (*Figure 3—figure supplement 1*). In the ACh-bound structure, W413[6.48] was in a conformation matching the inactive-state tiotropium-bound structure (*Figure 3C and D*). GaMD simulations also showed that W413[6.48] sampled a larger conformational space in the ACh-bound structure than in the Ipx-bound structure (*Figure 3—figure supplement 1C and G*). The predominate $\chi_2$ angle of W413[6.48] was approximately 60° and 105° in the ACh-bound and Ipx-bound simulations, respectively, corresponding to the cryo-EM conformations.

Located above ACh and Ipx is a tyrosine lid formed by three residues (Y113[3.33], Y416[6.51], and Y439[7.39]) that separate the orthosteric binding site from an extracellular vestibule (ECV) at the top of the receptor and the bulk solvent (*Figure 3C*). In the inactive conformation, the tyrosine lid is partially open due to Y416[6.51] rotating away from the binding pocket to accommodate the binding of bulkier inverse agonists such as tiotropium. In contrast, mAChR agonists are typically smaller in size than antagonists and inverse agonists, and this is reflected in a contraction of the size of the orthosteric binding pocket from 115 Å³ when bound to tiotropium to 77 and 63 Å³ when bound to ACh and Ipx, respectively (*Figure 3G and H*; *Tian et al., 2018*). Together, the smaller binding pocket of Ipx and more stable binding interactions with nearby residues that include W413[6.48] likely explain why Ipx has greater than 1000-fold higher binding affinity than ACh.

## Structure and dynamics of PAM binding and allosteric modulation of agonist affinity

The $M_4$R-$G_{i1}$ structures of LY298 and VU154 co-bound with Ipx are very similar to the Ipx- and ACh-bound structures, as well as to prior structures of the $M_2$ mAChR bound to Ipx and the PAM, LY2119620 (*Figure 2—figure supplement 4*; *Kruse et al., 2013*; *Maeda et al., 2019*). Both LY298 and VU154 bind directly above the orthosteric site in the ECV that is composed of a floor delineated by the tyrosine lid, and 'walls' formed by residues from TM2, TM6, TM7, ECL2, and ECL3 (*Figure 4A and B*). The EM density surrounding the PAM binding site and the ECV of the $M_4$ mAChR were clearly resolved with one exception; in the VU154-bound structure, the EM density begins to weaken around the trifluoromethylsulfonyl moiety (*Figure 2B*, *Figure 4B*). This was likely due to the moiety's ability to freely rotate and a lack of strong interactions with the receptor.

Given the overall similarities revealed by our four cryo-EM structures, we examined whether there were further differences in the dynamics between the PAM-bound structures by performing a 3D multivariance analysis (3DVA) of the principal components of motion within the Ipx-, LY298-Ipx, VU154-Ipx, and ACh-bound $M_4$R-$G_{i1}$ cryo-EM data sets using Cryosparc (*Punjani and Fleet, 2021*); a similar analysis performed previously on cryo-EM structures of class A and class B GPCRs provided important insights into the allosteric motions of extracellular domains and receptor interactions with G proteins (*Josephs et al., 2021*; *Liang et al., 2020*; *Mobbs et al., 2021*; *Zhang et al., 2020*).

In the 3DVA of the Ipx-bound complex, the $M_4$ mAChR appeared less flexible than the receptor in the ACh-bound complex (*Videos 3 and 4*) consistent with Ipx having a higher binding affinity and more stable pose during the GaMD simulations (*Figure 3E and F*). The LY298-Ipx-bound complex appeared similar to the Ipx-bound complex with LY298 being bound in the ECV (*Video 5*). In contrast, the 3DVA of the VU154 structure had more dynamic movements in the allosteric pocket that could reflect partial binding of VU154 (*Video 6*). This observation was in line with our findings that VU154 had

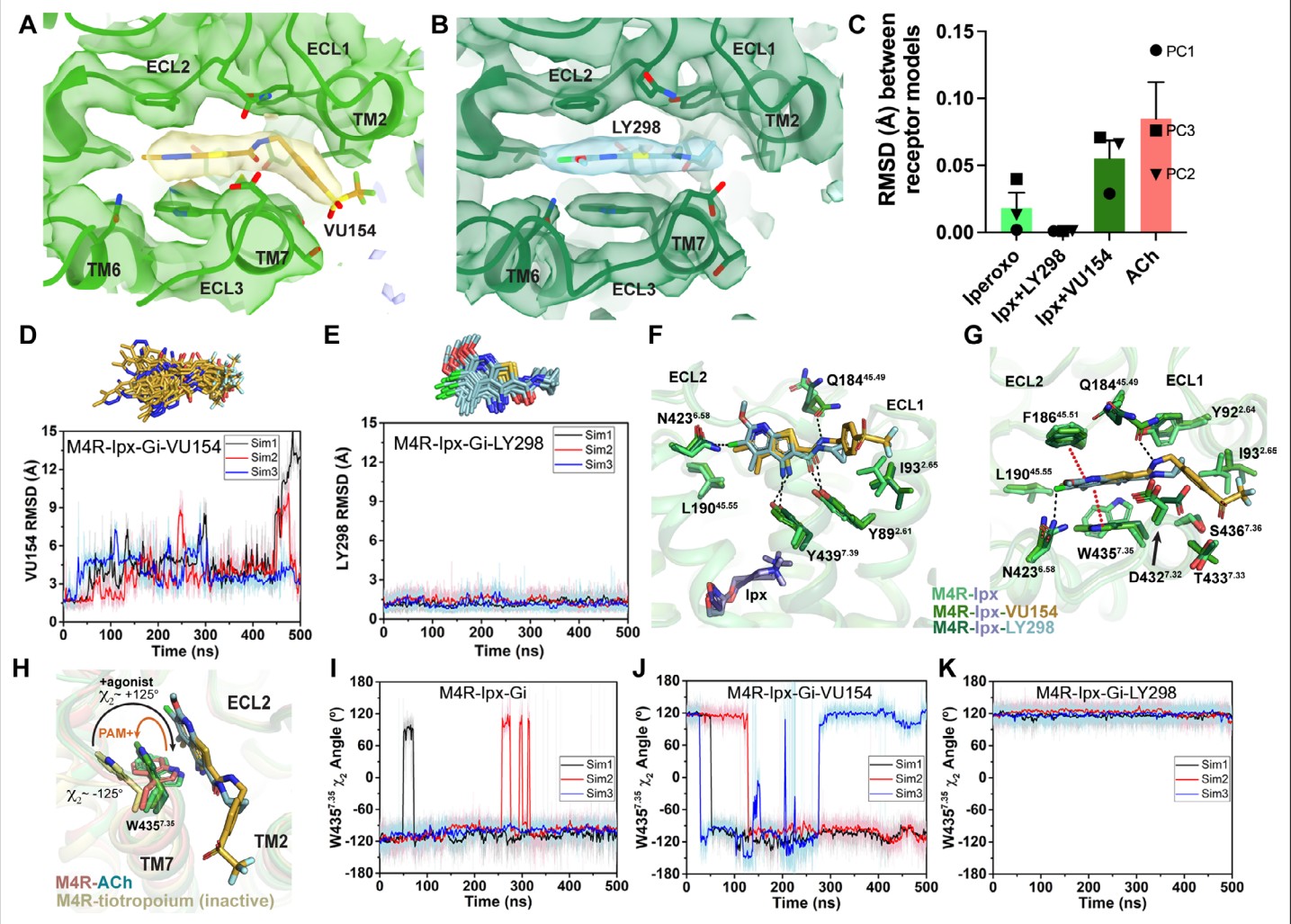

**Figure 4.** Binding and dynamics of LY298 and VU154. (**A, B**) Cryo-electron microscopy (cryo-EM) density of the (**A**) VU154- and (**B**) LY298-binding sites. (**C**) The root mean square deviations (RMSDs) between receptor models of the respective cryo-EM structures that were refined into the first and last frames of the EM maps from each principal component (PC1-PC3) of the 3D variability analysis. Values shown are mean ± SEM. (**D, E**) Top representative binding conformations of (**D**) VU154 and (**E**) LY298 obtained from structural clustering with frame populations ≥1% and time courses of the RMSDs of each positive allosteric modulator (PAM) relative to the cryo-EM structures. (**F, G**) Binding interactions of VU154 and LY298 with views from the (**F**) membrane and (**G**) extracellular surface. (**H**) Position and $\chi_2$ angle of W435[7.35] in the tiotropium-, ACh-, Ipx-, VU154-Ipx-, and LY298-Ipx bound structures. (**I–K**) Time courses of the W435[7.35] $\chi_2$ angle obtained from Gaussian accelerated molecular dynamics (GaMD) simulations on the (**I**) Ipx-, (**J**) VU154-Ipx-, and (**K**) LY298-Ipx-bound cryo-EM structures. See *Table 3*.

The online version of this article includes the following source data and figure supplement(s) for figure 4:

**Figure supplement 1.** Gaussian accelerated molecular dynamics (GaMD) simulations of LY293 and VU154 binding.

**Figure supplement 2.** Key residues for the binding of LY298 and VU154 at the human M4 muscarinic acetylcholine receptor (mAChR).

**Figure supplement 2—source data 1.** Related to *Figure 4—figure supplement 2*.

**Figure supplement 3.** Gaussian accelerated molecular dynamics (GaMD) simulations of M4R complexes with acetylcholine (ACh).

lower binding modulation (*Figure 1E*) and functional modulation with agonists than LY298 (*Figure 1J*, *Figure 1—figure supplement 2D*, *Table 1*). To quantify the differences from the 3DVA, we rigid body fitted and refined the respective M4R-G$_{i1}$ models into the first and last frames of the EM maps from each principal component of the 3DVA and then calculated the RMSD between the receptor models (*Figure 4C*). In agreement with our prior observations, the VU154-Ipx-bound and ACh-bound complexes had greater RMSDs with values of 0.06 and 0.09 Å, respectively. Comparatively, the Ipx-bound and LY298-Ipx-bound complexes had lower RMSD values of 0.02 and 0.001 Å, respectively. The results of the 3DVA do not represent *bona fide* measures of receptor dynamics, rather they are

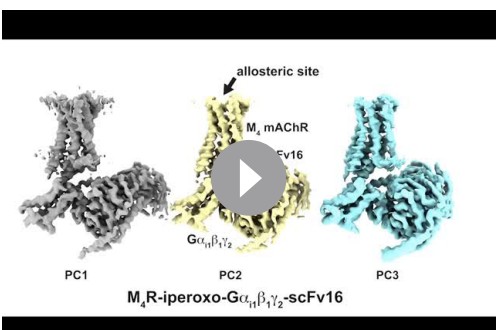

**Video 3.** 3D variability analysis of the Ipx-M$_4$R-G$_{i1}$ cryo-electron microscopy (cryo-EM) structure.
https://elifesciences.org/articles/83477/figures#video3

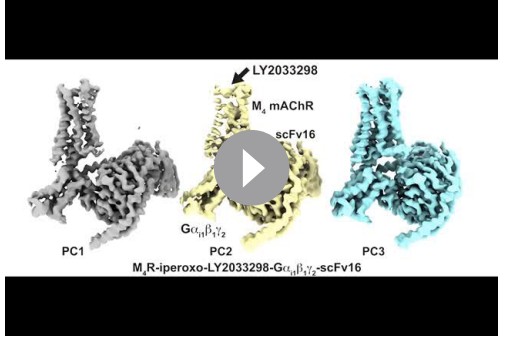

**Video 5.** 3D variability analysis of the LY298-Ipx-M$_4$R-G$_{i1}$ cryo-electron microscopy (cryo-EM) structure.
https://elifesciences.org/articles/83477/figures#video5

suggestive of differences between the collected data sets that led to the structures. To support these findings, we compared the GaMD simulations of all four cryo-EM structures (*Table 3*). Notably, VU154 underwent considerably higher fluctuations than LY298 with RMSDs ranging from 1.5 to 15 Å for VU154 (*Video 7*) and 0.8–2.1 Å for LY298 (*Video 8*) relative to the cryo-EM structures (*Figure 4D and E*). Therefore, the GaMD simulations corroborate our 3DVA results and suggest that complexes bound to agonists with high affinity or co-bound with agonists and PAMs with high positive cooperativity will exhibit lower dynamic fluctuations.

To investigate why the binding of LY298 was more stable than VU154, we examined the ligand interactions with the receptor. There are three key binding interactions that are shared between both PAMs and the M$_4$ mAChR: (1) a three-way π-stacking interaction between F186[45.51] (ECL2 residues have been numbered 45.X denoting their position between TM4 and TM5 with X.50 being a conserved cysteine residue), the aromatic core of the PAMs, and W435[7.35]; (2) a hydrogen bond between Y439[7.39] of the tyrosine lid and the primary amine of the PAMs; and (3) a hydrogen bond between Y89[2.61] and the carbonyl oxygen of the PAMs (*Figure 4F and G*). While these interactions are conserved for both PAMs in the consensus cryo-EM maps, during GaMD simulations these interactions were more stable with LY298 than VU154 (*Figure 4H–K*, *Figure 4—figure supplement 1*). The importance of these interactions was validated pharmacologically (*Figure 4—figure supplement 2*, *Table 4*), whereby mutation of any of these residues completely abolished the binding affinity modulation mediated by LY298 and VU154 at the M$_4$ mAChR with both Ipx and ACh as agonists.

A potential fourth interaction was observed with residue Q184[45.49] and the amide nitrogen of the PAMs; however, the GaMD simulations suggest that this interaction is relatively weak (*Figure 4—figure supplement 1D and H*), consistent with the fact that mutation of Q184[45.49] to alanine had no effect on the binding affinity modulation of LY298 or VU154 (*Figure 4—figure supplement 2*, *Table 4*). In addition, each PAM had at least one potential unique binding interaction with the receptor (*Figure 4F and G*). For LY298, this is an interaction between the fluorine atom and N423[6.58] that appeared to be

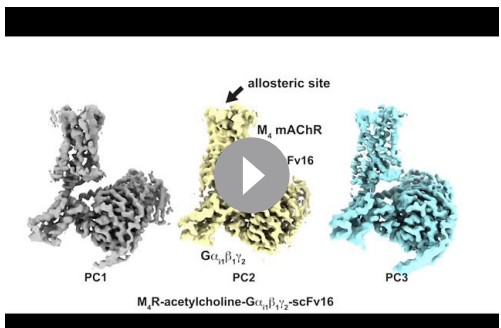

**Video 4.** 3D variability analysis of the ACh-M$_4$R-G$_{i1}$ cryo-electron microscopy (cryo-EM) structure.
https://elifesciences.org/articles/83477/figures#video4

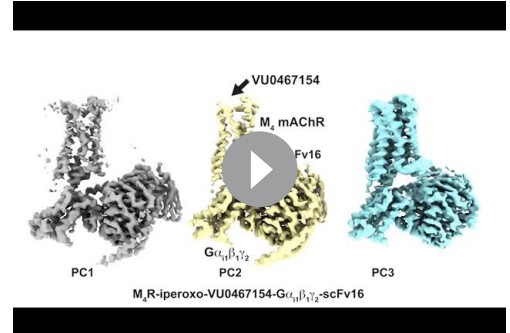

**Video 6.** 3D variability analysis of the VU154-Ipx-M$_4$R-G$_{i1}$ cryo-electron microscopy (cryo-EM) structure.
https://elifesciences.org/articles/83477/figures#video6

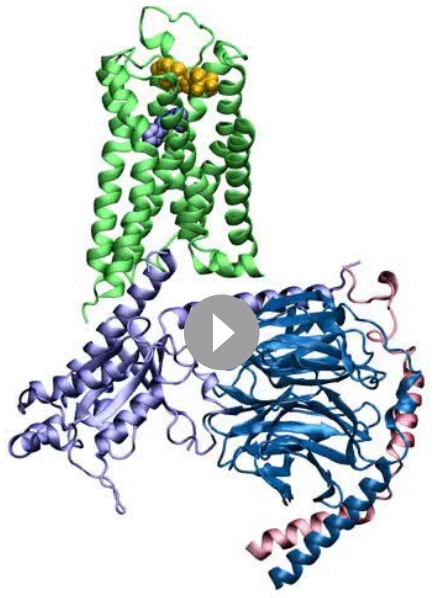

time:   0 ns

**Video 7.** Movie from one VU154-Ipx-M$_4$R-G$_{i1}$ Gaussian accelerated molecular dynamics (GaMD) simulation. https://elifesciences.org/articles/83477/figures#video7

stable during simulation and, when mutated to alanine reduced the binding modulation of LY298 (*Figure 4—figure supplement 1K*, *Figure 4—figure supplement 2*, *Table 4*; *Thal et al., 2016*). For VU154, there were two additional possible hydrogen bonding interactions with residues Y92$^{2.64}$ and T433$^{7.33}$ (*Figure 4G*); however, these interactions were highly fluctuating during GaMD simulations, suggesting they were – at best – transient interactions (*Figure 4—figure supplement 1I and J*). Finally, W435$^{7.35}$ is a key residue in the ECV that changes from a planar rotamer in the agonist-bound structures to a vertical rotamer that π stacks against the PAMs (*Figure 4H*). In GaMD simulations of the Ipx-bound structure, W435$^{7.35}$ is predominantly in a planar conformation that corresponds to its conformation in the cryo-EM structure (*Figure 4I*). In contrast, the binding of LY298 stabilizes W435$^{7.35}$ into a vertical position (*Figure 4K*). However, in the VU154-bound receptor, W435$^{7.35}$ appears to alternate between the planar and vertical positions, consistent with VU154 having a less stable binding pose (*Figure 4J*). These results indicate that the binding of LY298 is more stable than VU154 due to LY298 being able to form stable binding inter-

actions with key residues in the ECV. This provides a likely explanation for why LY298 was able to exert greater positive binding cooperativity on orthosteric agonists than VU154.

## A molecular mechanism of probe dependence

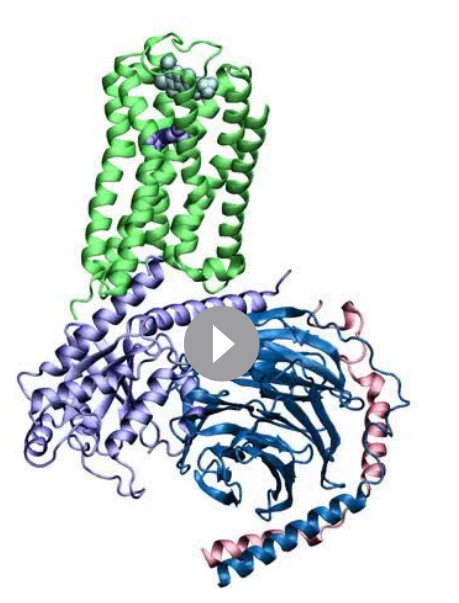

time:   0 ns

**Video 8.** Movie from one LY298-Ipx-M$_4$R-G$_{i1}$ Gaussian accelerated molecular dynamics (GaMD) simulation. https://elifesciences.org/articles/83477/figures#video8

As highlighted above, PAMs, LY298 and VU154, displayed stronger allosteric binding affinity modulation with ACh than Ipx, an example of probe dependence (*Figure 1E*, *Table 1*). These findings are in accord with previous studies where we identified probe dependence in the actions of LY298 when tested against other orthosteric agonists (*Chan et al., 2008*; *Suratman et al., 2011*). To investigate a mechanism for probe dependence at the M$_4$ mAChR, we performed GaMD simulations with LY298 and VU154 co-bound with ACh by replacing Ipx with ACh in the corresponding cryo-EM structures (*Table 3*, *Figure 4—figure supplement 3*). In the absence of PAM, ACh was more dynamic than Ipx with root-mean-square fluctuations (RMSF) of 2.13 Å versus 0.88 Å, reflective of the fact Ipx binds with higher affinity than ACh (*Figure 4—figure supplement 3F*). In the presence of LY298 or VU154, the dynamics of ACh binding was decreased, with RMSFs reduced to 1.23 Å and 1.82 Å, respectively, and with LY298 having the greatest effect (*Figure 4—figure supplement 3F*). This is in line with LY298 having more cooperativity with ACh than VU154 (*Figure 1E*). In comparison to

**Table 4.** Pharmacological parameters of LY298 and VU154 at key $M_4$ muscarinic acetylcholine receptor (mAChR) mutants.

[$^3$H]-NMS saturation binding on stable $M_4$ mAChR Flp-In CHO cells

| Constructs | Sites per cell* | $pK_D$† |
|---|---|---|
| Human WT $M_4$ mAChR (from *Table 1*) | 598,111 ± 43,067 (7) | 9.76 ± 0.05 (7) |
| Y89A$^{2.61}$ | 32,674 ± 4174 (4) | 9.88 ± 0.06 (4) |
| Q184A$^{45.49}$ | 88,728 ± 3056 (3) | 9.99 ± 0.06 (3) |
| F186A$^{45.51}$ | 36,907 ± 4170 (4) | 9.75 ± 0.16 (4) |
| W435A$^{7.35}$ | 34,861 ± 3510 (3) | 9.81 ± 0.22 (3) |
| Y439A$^{7.39}$ | 42,690 ± 4547 (3) | 8.31 ± 0.14 (3) |

[$^3$H]-NMS interaction binding assays between ACh or Ipx and LY298 or VU154 on stable $M_4$ mAChR constructs in Flp-In CHO cells

| Constructs | PAM | $pK_i$ ACh‡ | $pK_i$ Ipx‡ | $pK_B$ PAM‡ | log $\alpha_{ACh}$§ | log $\alpha_{Ipx}$§ | log $\alpha_{NMS}$¶ |
|---|---|---|---|---|---|---|---|
| Human WT $M_4$ | LY298 | 5.09 ± 0.07 (7) | 8.54 ± 0.04 (11) | = 5.65 | 1.57 ± 0.11 | 1.71 ± 0.09 | = 0 |
|  | VU154 | 5.06 ± 0.05 (7) | 8.54 ± 0.03 (11) | = 5.83 | 1.44 ± 0.07 | 1.11 ± 0.06 | = 0 |
| Y89A$^{2.61}$ | LY298 | 5.25 ± 0.05 (6) | 8.48 ± 0.05 (6) | N.D. | N.D. | N.D. | N.D. |
|  | VU154 | 5.27 ± 0.05 (6) | 8.47 ± 0.05 (6) | N.D. | N.D. | N.D. | N.D. |
| Q184A$^{45.49}$ | LY298 | 5.24 ± 0.06 (6) | 8.74 ± 0.04 (10) | 6.23 ± 0.06 | 1.28 ± 0.13 | 1.27 ± 0.11 | −1.10 ± 0.07 |
|  | VU154 | 5.28 ± 0.05 (6) | 8.69 ± 0.04 (10) | 5.87 ± 0.17 | 1.07 ± 0.09 | 0.81 ± 0.07 | = 0 |
| F186A$^{45.51}$ | LY298 | 4.91 ± 0.05 (6) | 8.12 ± 0.05 (8) | N.D. | N.D. | N.D. | N.D. |
|  | VU154 | 4.91 ± 0.05 (6) | 8.12 ± 0.05 (8) | N.D. | N.D. | N.D. | N.D. |
| W435$^{7.35}$ | LY298 | 3.79 ± 0.07 (7) | 6.88 ± 0.07 (7) | N.D. | N.D. | N.D. | N.D. |
|  | VU154 | 3.79 ± 0.07 (7) | 6.88 ± 0.07 (7) | N.D. | N.D. | N.D. | N.D. |
| Y439A$^{7.39}$ | LY298 | 3.23 ± 0.22 (8) | 5.36 ± 0.25 (8) | N.D. | N.D. | N.D. | N.D. |
|  | VU154 | 3.23 ± 0.22 (8) | 5.36 ± 0.25 (8) | N.D. | N.D. | N.D. | N.D. |

Values represent the mean ± SEM with the number of independent experiments shown in parenthesis.

N.D.: not determined; ACh: acetylcholine; Ipx: iperoxo; PAM: positive allosteric modulator.

*Number of [$^3$H]-NMS binding sites per cell.

†Negative logarithm of the radioligand equilibrium dissociation constant.

‡Negative logarithm of the orthosteric ($pK_i$) or allosteric ($pK_B$) equilibrium dissociation constant. $pK_i$ values for ACh and Ipx are shared at each M4 mAChR construct. $pK_B$ values for the PAMs at Q184A are shared across the agonist data sets.

§Logarithm of the binding cooperativity factor between the agonist (ACh or Ipx) and the PAM (LY298 or VU154).

¶Logarithm of the binding cooperativity factor between the [$^3$H]-NMS and the PAM (LY298 or VU154).

ACh, there was a modest increase in the dynamics of Ipx with the addition of LY298 or VU154, likely reflecting the fact Ipx binding to the receptor was already stable (*Figure 3—figure supplement 1I and J*, *Figure 4—figure supplement 3F*). These results provide a plausible mechanism for probe dependence, at least with regard to differences in the magnitude of the allosteric effect depending on the ligand bound. Namely, PAMs manifest higher cooperativity when interacting with agonists, such as ACh, that are inherently less stable on their own when bound to the receptor, in contrast to more stable ligands such as Ipx.

## Structural and dynamic insights into orthosteric and allosteric agonism

In addition to the ability to allosterically modulate the function of orthosteric ligands, it has become increasingly appreciated that allosteric ligands may display variable degrees of direct agonism in their own right, over and above any allosteric modulatory effects (*Changeux and Christopoulos, 2016*). Prior studies have established that the activation process of GPCRs involves conformational changes that extend from the extracellular domains through to the intracellular surface (*Nygaard et al., 2009*). Comparison of the active state ACh-, Ipx-, LY298-Ipx-, and VU154-Ipx-bound $M_4$R-$G_{i1}$ structures to the inactive state tiotropium-bound $M_4$ mAChR structure (Protein Data Bank accession 5DSG) (*Thal et al., 2016*) thus affords an opportunity to gain new insights into the activation process mediated by multiple orthosteric agonists in the presence and absence of two different PAMs that display high (LY298) and low (VU154) degrees of direct allosteric agonism (*Figures 1H and 5A–C*).

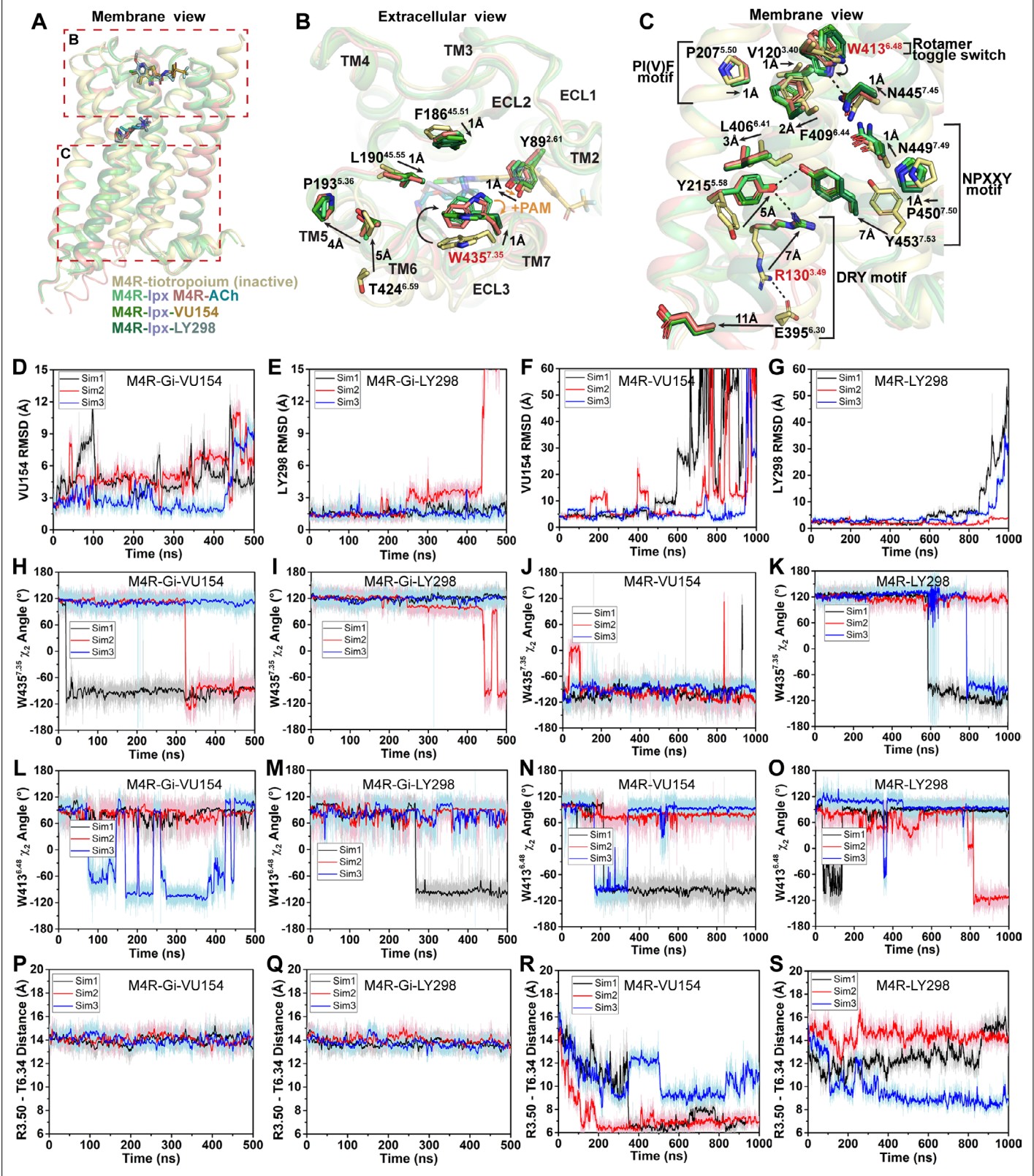

**Figure 5.** Structural and dynamic insights into orthosteric and allosteric agonism. (**A**) Cartoon of the receptor models indicating regions of interest for panels (**B, C**) shown within the red boxes. (**B**) View of the tiotropium-bound, agonist-bound, and positive allosteric modulator (PAM)-agonist-bound conformations from the extracellular surface. (**C**) Membrane view of residues and activation motifs involved in signaling. Residues colored red in (**B, C**) indicate residues of investigated in Gaussian accelerated molecular dynamics (GaMD) simulations. (**D–G**) Time course of the root mean square

*Figure 5 continued on next page*

*Figure 5 continued*

deviations (RMSDs) of the PAMs (**D, E**) from GaMD simulations of the M₄R bound to G protein and no orthosteric agonist, (**F, G**) and in the absence of both G protein and agonist. (**H–K**) Similar to (**D–G**) the time courses of (**H–K**) the W435$^{7.35}$ $\chi_2$ angle, (**L–O**) the W413$^{6.48}$ $\chi_2$ angle, and (**P–S**) the TM3-TM6 distance measured by distance between R130$^{3.50}$ and T399$^{6.34}$. See *Table 3*.

As discussed previously, agonist binding decreases the size of the orthosteric binding site (*Figure 3G and H*). The primary driver of this decrease was the tyrosine lid residue Y416$^{6.51}$, which underwent a large rotation toward Y113$^{3.33}$ creating a hydrogen bond that seals off the tyrosine lid (*Figure 3C*). The closure of the tyrosine lid was further reinforced by a change in the rotamer of W435$^{7.35}$ to a planar position that sits parallel to the tyrosine lid allowing for a π-π interaction with Y416$^{6.51}$ and a positioning of the indole nitrogen of W435$^{7.35}$ to potentially form a hydrogen bond with the hydroxyl of Y89$^{2.61}$ (*Figure 5B*). The contraction of the orthosteric pocket by the inward movement of Y416$^{6.51}$ also led to a contraction of the ECV with a 5 Å inward movement of the top of TM6 and ECL3. As a consequence, the top of TM5 was displaced outward by 4 Å forming a new interface between TM5 and TM6 that was stabilized by a hydrogen bond between T424$^{6.59}$ and the backbone nitrogen of P193$^{5.36}$ along with aromatic interactions between F197$^{5.40}$ and F425$^{6.60}$ (*Figure 5B*). These interactions were specific to the active state structures and appear to be conserved as they were also present in the M₁ and M₂ mAChR active state structures (*Maeda et al., 2019*). In addition to the movements of TM5 and TM6, there was a smaller 1 Å inward movement of ECL2 (*Figure 5B*). The binding of LY298 and VU154 had a minimal impact on the conformation of most ECL residues, implying that the reorganization of residues in the ECV by orthosteric agonists contributes to the increased affinity of the PAMs (*Figure 1G*). There was a slight further inward shift of ECL2 toward the PAMs to facilitate the 3-way π-stacking interaction with F186$^{45.51}$ and W435$^{7.35}$. In addition, in the PAM-bound structures, Y89$^{2.61}$ rotated away from its position in the ACh- and Ipx-bound structures either due to a loss of an interaction with W435$^{7.35}$ or to form a better hydrogen bond with the carbonyl oxygen of the PAMs (*Figure 5B*).

Below the orthosteric binding site are several signaling motifs that are important for the activation of class A GPCRs, including the PIF motif (*Rasmussen et al., 2011*; *Wacker et al., 2013*), the Na⁺ binding site (*Liu et al., 2012a*; *White et al., 2018*), the NPxxY motif (*Fritze et al., 2003*), and the DRY motif (*Figure 5C*; *Ballesteros et al., 2001*). The conformations of these activation motifs were very similar across all four active-state M₄ mAChR structures and were consistent with the position of these motifs across other active-state class A GPCR structures (*Zhou et al., 2019*). Collectively, all of the described activation motifs facilitate an 11 Å outward movement of TM6 that typifies GPCR activation and creation of the G protein binding site. In comparison to the ECV residues (*Figure 5B*), beyond the rotamer toggle switch residue W413$^{6.48}$, there are no discernible differences between the agonist and PAM-agonist-bound structures, suggesting a shared activation mechanism for residues below W413$^{6.48}$ (*Figure 5C*).

As indicated above, LY298 also displays robust allosteric agonism in comparison to VU154 (*Figure 1H*, *Figure 1—figure supplement 2B*). To probe whether the allosteric agonism of LY298 could be related to its ability to better stabilize the M₄ mAChR in an active conformation in comparison to VU154, we performed additional GaMD simulations on the LY298-Ipx- and VU154-Ipx-bound M₄R-G$_{i1}$ structures with the agonist Ipx removed (3 × 500 ns) and with both Ipx and the G protein removed (3 × 1000 ns) (*Figure 5D–S*, *Table 3*). In GaMD simulations, LY298 underwent lower RMSD fluctuations than VU154 before dissociating from the receptor (*Figure 5D–G*). Similarly, the conformations of W435$^{7.35}$ and W413$^{6.48}$ were better stabilized in the LY298-Ipx-bound systems, indicating that LY298 more strongly promotes an active receptor conformation (*Figure 5H–K*). In the presence of the G protein, both PAMs stabilized an active conformation of the receptor based on the distances between TM3 and TM6 (*Figure 5P and Q*). Upon removal of the G protein, the VU154-bound M₄ mAChR quickly transitioned toward the inactive conformation, while the LY298-bound M₄ mAChR was more resistant to deactivation in the GaMD simulations (*Figure 5R and S*). This observation supports LY298 having greater efficacy than VU154 (*Table 1*) as it better stabilizes the active conformation of the M₄ mAChR. Overall, the GaMD simulations show that in the absence of agonist alone, or agonist and G protein, LY298 better stabilizes activation motifs from the top of the receptor (W435$^{7.35}$) all the way down to the intracellular G protein binding pocket (DRY-TM6), providing mechanistic insights into the function of LY298 as a stronger PAM-agonist than VU154.

## Structural insights into allosteric modulation of agonist signaling

In a previous study, we characterized over 40 distinct mutations of $M_4$ mAChR residues that span from the orthosteric site up to the extracellular surface (*Table 5*; *Leach et al., 2011*; *Nawaratne et al., 2010*; *Thal et al., 2016*). As expected, these studies revealed that mutation of residues around the orthosteric and allosteric sites often resulted in a reduction in the binding affinity of either ACh or LY298 at their respective binding sites, though the allosteric site was typically less affected (*Figure 6A and B*, *Table 5*). In contrast, the binding affinity modulation between ACh and LY298 was largely affected by mutation of aromatic residues that link the orthosteric and allosteric sites (*Figure 6C*), implying a network of residues that were responsible for transmitting binding cooperativity between these two sites (*Thal et al., 2016*). Analyzing unpublished data from prior studies allowed an examination of the signaling efficacy of ACh ($\tau_A$) and LY298 ($\tau_B$), but also the functional cooperativity (αβ) in the context of active state structures of the co-complexes (*Figure 6D–F*, *Figure 6—figure supplement 1*, *Table 5*).

Mutation of residues that directly surround ACh primarily decreased the efficacy of ACh (*Figure 6D*, *Table 5*). One exception was W98[23.50] (an ECL1 residue numbered 23.X denoting its position between TM2 and TM3 with X.50 denoting the most conserved residue), a residue that was recently identified in a deep scanning mutagenesis study as a conserved class A residue that is intolerant to mutation (*Jones et al., 2020*) and stabilizes the conserved disulfide bridge between ECL1 and TM3 that is important for the stability of the active state of many GPCRs including mAChRs (*Hulme, 2013*). Interestingly, residues that affect the efficacy of LY298 include nearly all of the residues that also affect ACh efficacy, along with residues that link to the allosteric site and surround the LY298 binding site (*Figure 6E*, *Table 5*). This suggests that the direct signaling of LY298 via the allosteric site is nonetheless linked through a similar network of residues and requires a functional orthosteric site for the transduction of signaling, and that mechanism involves equivalent closure of the orthosteric binding site, consistent with the thermodynamic reciprocity of cooperativity (*Canals et al., 2011*).

Residues Y89[2.61], N432[6.58], W435[7.35], and W440[7.40] were identified as residues that, when mutated to alanine, significantly decreased the functional modulation between ACh and LY298 (*Figure 6F*, *Table 5*). In prior work, all four residues were also shown to contribute to LY298 binding or affinity modulation (*Thal et al., 2016*). Surprisingly, three mutations resulted in increased functional modulation by LY298. Of particular interest was, again, the rotamer toggle switch residue W413[6.48]. Mutation of W413[6.48] to alanine significantly impaired the efficacy of ACh but only reduced the efficacy of LY298 by twofold, such that ACh and LY298 had similar efficacy for this mutant (*Figure 6G–I*, *Table 5*). Interestingly, the functional modulation (αβ) between ACh and LY298 increased to over 3600 (a 20-fold increase vs. WT) at W413A[6.48]. Similar results were observed in the TruPath assay with ACh, Ipx, and LY298 (*Figure 6—figure supplement 2* [mutant], *Figure 1—figure supplement 1B* [WT]). However, with VU154, the functional modulation was considerably reduced with ACh and non-existent with Ipx, in line with our TruPath experiments at the WT $M_4$ mAChR. These results show that, at the $M_4$ mAChR, the rotamer toggle switch residue is important for the signaling efficacy of orthosteric agonists and PAM-agonists but does not impair the process of functional allosteric modulation. Thus, suggesting that the stability of LY298 co-binding with agonists can restore impaired function, while the less stable binding of VU154 does not. Together with the observation that most of the structural differences between the active-state $M_4$ mAChR structures occur at or above W413[6.48], we propose that this residue has a strong role in maintaining the conformational dynamics of the receptor and is a key trigger for robust signal transduction.

## A molecular basis of species selectivity

One of the main advantages of allosteric modulators is the ability to selectivity target highly conserved proteins. The mAChRs are the prime example where allosteric modulators have been designed to selectively target specific subtypes. To date, the only PAM-bound mAChR structures are ones with LY2119620, a PAM that has activity at both the $M_2$ and $M_4$ mAChRs. Similarly, LY298 has activity at the $M_2$ mAChR. However, the allosteric properties of VU154 are differentially affected by the species of the receptor (*Wood et al., 2017b*; *Wood et al., 2017a*). At the human $M_4$ mAChR, LY298 displays robust binding affinity modulation, functional modulation, and allosteric agonism, while VU154 has comparatively weaker allosteric properties (*Figure 1*, *Table 1*). Conversely, at the mouse $M_4$ mAChR, VU154 has a high degree of positive binding modulation, functional modulation, and allosteric agonism that

**Table 5.** Pharmacological parameters of $M_4$ muscarinic acetylcholine receptor (mAChR) mutants.

| Constructs | pERK1/2 interaction assays* | | | [³H]-QNB interaction binding assays† | | | Study |
|---|---|---|---|---|---|---|---|
| | log $\tau_c$ ACh ‡ | log $\tau_c$ LY298 ‡ | log $\alpha\beta$ § | p$K_i$ ACh ¶ | p$K_B$ LY298 ¶ | log $\alpha$** | |
| WT $M_4$ mAChR | 2.96 ± 0.14 (4) | 1.10 ± 0.09 | 2.43 ± 0.14 | 4.51 ± 0.15 | 4.89 ± 0.12 | 1.97 ± 0.11 | Current/Thal†† |
| S85A[2.57] | 3.15 ± 0.11(4) | 0.91 ± 0.07 | 1.75 ± 0.09 | 4.09 ± 0.11 | 5.44 ± 0.14 | 1.43 ± 0.06 | Current/Thal |
| Y89A[2.61] | 2.53 ± 0.17 (5) | = −3 | −0.43 ± 0.27* | 5.07 ± 0.45 | 5.36 ± 0.03 | −0.13 ± 0.08‡‡ | Current/Thal |
| Y92A[2.64] | 2.26 ± 0.15 (4) | −0.06 ± 0.16* | 2.25 ± 0.11 | 4.15 ± 0.25 | 4.53 ± 0.15 | 1.2 ± 0.19‡‡ | Current/Thal |
| I93T, I94V, K95I | 2.57 ± 0.11 | 2.27 ± 0.19‡‡ | N.T. | 4.69 ± 0.11 | 4.82 ± 0.36 | 2.14 ± 0.17‡‡ | Nawaratne §§ |
| I93T[2.65] | 2.34 ± 0.09 | 2.38 ± 0.22‡‡ | N.T. | 4.97 ± 0.04 | 5.36 ± 0.09 | 2.42 ± 0.16‡‡ | Nawaratne |
| I94V[2.66] | 2.34 ± 0.09 | 1.24 ± 0.09 | N.T. | 4.71 ± 0.06 | 5.17 ± 0.08 | 1.74 ± 0.07 | Nawaratne |
| K95I[2.67] | 2.00 ± 0.07 | 0.61 ± 0.09‡‡ | N.T. | 4.86 ± 0.05 | 5.20 ± 0.14 | 1.24 ± 0.04‡‡ | Nawaratne |
| Y97A[23.49] | 2.94 ± 0.11 (4) | 2.19 ± 0.07* | 3.43 ± 0.12* | 4.69 ± 0.17 | 4.25 ± 0.10‡‡ | 2.33 ± 0.12 | Current/Thal |
| W98A[23.50] | 1.45 ± 0.15* (5) | = −3 | 2.29 ± 0.10 | 3.65 ± 0.11‡‡ | 4.39 ± 0.04 | 0.73 ± 0.07‡‡ | Current/Thal |
| G101A[23.53] | 2.58 ± 0.09 (4) | 0.43 ± 0.08* | 2.10 ± 0.07 | 4.37 ± 0.19 | 5.03 ± 0.17 | 1.47 ± 0.01 | Current/Thal |
| D106A[3.26] | 1.24 ± 0.11 | = −3 | N.T. | 3.95 ± 0.09‡‡ | 5.29 ± 0.11 | 1.51 ± 0.15 | Leach ¶¶ |
| W108A[3.28] | 1.49 ± 0.17 | = −3 | N.T. | 4.01 ± 0.06‡‡ | 4.24 ± 0.07‡‡ | 1.23 ± 0.01‡‡ | Leach |
| L109A[3.29] | 1.17 ± 0.14 | = −3 | N.T. | 3.11 ± 0.09‡‡ | 4.28 ± 0.14‡‡ | 2.54 ± 0.10‡‡ | Leach |
| D112E[3.32] | −0.80 ± 0.16‡‡ | = −3 | N.T. | <2 | 5.56 ± 0.13 | 0.39 ± 0.11‡‡ | Leach |
| D112N[3.32] | N.D. | N.D. | N.T. | 3.19 ± 0.02‡‡ | 5.79 ± 0.2‡‡ | 0.74 ± 0.08 | Leach |
| Y113A[3.33] | N.T. | N.T. | N.T. | 2.98 ± 0.12‡‡ | 4.97 ± 0.15 | 0.80 ± 0.10‡‡ | Leach |
| S116A[3.36] | 0.82 ± 0.17‡‡ | −0.35 ± 0.45‡‡ | N.T. | 3.61 ± 0.10‡‡ | 5.12 ± 0.08 | 1.54 ± 0.05 | Leach |
| N117A[3.37] | 0.80 ± 0.27‡‡ | −0.27 ± 0.16‡‡ | N.T. | 3.64 ± 0.04‡‡ | 5.30 ± 0.15 | 1.57 ± 0.13 | Leach |
| V120A[3.40] | 1.47 ± 0.11 | 1.20 ± 0.19 | N.T. | 5.63 ± 0.05‡‡ | 5.41 ± 0.10 | 1.83 ± 0.11 | Leach |
| D129E[3.49] | 1.45 ± 0.24 | 0.78 ± 0.16 | N.T. | 5.04 ± 0.07 | 5.59 ± 0.12 | 1.61 ± 0.16 | Leach |

*Table 5 continued on next page*

*Table 5 continued*

| | pERK1/2 interaction assays* | | | [³H]-QNB interaction binding assays† | | | Study |
|---|---|---|---|---|---|---|---|
| D129N$^{3.49}$ | 2.56 ± 0.39 | 1.86 ± 0.12 | N.T. | 5.54 ± 0.10 ‡‡ | 5.37 ± 0.20 | 1.86 ± 0.22 | Leach |
| W164A$^{4.57}$ | N.D. (3) | = −3 | 2.17 ± 0.64*** | 3.95 ± 0.24 | 5.15 ± 0.28 | ND | Current/Thal |
| F170A$^{4.63}$ | 3.13 ± 0.17 (5) | 2.66 ± 0.12* | 3.58 ± 0.17* | 4.77 ± 0.2 | 4.53 ± 0.06 | 2.23 ± 0.13 | Current/Thal |
| W171A$^{4.64}$ | 2.59 ± 0.17 (5) | 1.31 ± 0.11 | 3.11 ± 0.13 | 3.91 ± 0.21 | 4.56 ± 0.15 | 2.00 ± 0.09 | Current/Thal |
| Q172A$^{4.65}$ | 3.05 ± 0.33 (5) | 1.18 ± 0.31 | 2.71 ± 0.16 | 4.02 ± 0.09 | 4.99 ± 0.03 | 1.54 ± 0.08 | Current/Thal |
| F173A$^{4.66}$ | 3.39 ± 0.11(4) | 2.03 ± 0.10* | 3.31 ± 0.23* | 4.09 ± 0.01 | 4.78 ± 0.19 | 1.90 ± 0.14 | Current/Thal |
| Q184A$^{45.49}$ | 3.01 ± 0.10 (4) | 1.05 ± 0.08 | 2.08 ± 0.12 | 4.25 ± 0.12 | 5.36 ± 0.04 | 1.70 ± 0.05 | Current/Thal |
| F186A$^{45.51}$ | 1.99 ± 0.11 | N.D. | N.T. | 4.85 ± 0.06 | NR | NR | Nawaratne |
| I187A$^{45.52}$ | 2.57 ± 0.10 | 0.62 ± 0.09 | 2.07 ± 0.15 | 3.71 ± 0.12 | 5.46 ± 0.29 | 1.07 ± 0.29 ‡‡ | Current/Thal |
| Q188A$^{45.53}$ | 2.48 ± 0.15 | 0.99 ± 0.11 | 2.35 ± 0.16 | 4.6 ± 0.22 | 4.94 ± 0.08 | 1.49 ± 0.04 | Current/Thal |
| F189A$^{45.54}$ | 2.28 ± 0.11 | 1.25 ± 0.09 | 2.67 ± 0.11 | 4.65 ± 0.02 | 5.09 ± 0.13 | 1.99 ± 0.08 | Current/Thal |
| L190A$^{45.55}$ | 2.50 ± 0.14 | 1.32 ± 0.12 | 2.81 ± 0.12 | 4.20 ± 0.06 | 4.92 ± 0.1 | 2.06 ± 0.09 | Current/Thal |
| W413A$^{6.48}$ | 0.66 ± 0.12***,* (4) | 0.61 ± 0.13*** | 3.54 ± 0.09***,* | 3.47 ± 0.06 ‡‡ | 4.51 ± 0.37 | 2.45 ± 0.36 | Current/Thal |
| Y416A$^{6.51}$ | N.T. | N.T. | N.T. | 2.85 ± 0.10 ‡‡ | NR | NR | Thal |
| N423A$^{6.58}$ | 3.44 ± 0.15 (3) | 0.82 ± 0.10 | 1.43 ± 0.19* | 4.41 ± 0.15 | 5.02 ± 0.06 | 1.18 ± 0.08 ‡‡ | Current/Thal |
| Q427A$^{6.62}$ | 3.15 ± 0.14 (3) | 0.99 ± 0.12 | 1.64 ± 0.12 | 4.46 ± 0.03 | 5.43 ± 0.06 | 1.36 ± 0.04 | Current/Thal |
| S428P$^{6.63}$ | 1.99 ± 0.09 | 1.40 ± 0.19 | N.T. | 5.14 ± 0.03 ‡‡ | 5.17 ± 0.15 | 1.81 ± 0.11 | Nawaratne |
| D432N$^{7.32}$ | 2.26 ± 0.12 | 1.25 ± 0.18 | N.T. | 5.19 ± 0.04 ‡‡ | 5.21 ± 0.2 | 1.37 ± 0.04 | Nawaratne |
| W435A$^{7.35}$ | 2.58 ± 0.17 (4) | = −3 | N.R | 3.37 ± 0.08 ‡‡ | NR | NR | Current/Thal |
| Y439A$^{7.39}$ | 0.60 ± 0.18 ‡‡ | N.D. | N.T. | 3.33 ± 0.10 ‡‡ | 5.84 ± 0.12 | 0.49 ± 0.03 ‡‡ | Nawaratne |
| W440A$^{7.40}$ | 3.69 ± 0.17†††(4) | 0.84 ± 0.11 | 1.52 ± 0.20††† | 4.29 ± 0.24 | 4.94 ± 0.06 | 0.96 ± 0.04 ‡‡ | Current/Thal |
| C442A$^{7.42}$ | 1.49 ± 0.16 ‡‡ | 0.82 ± 0.31 | N.T. | 4.04 ± 0.07 ‡‡ | 5.35 ± 0.06 | 1.81 ± 0.03 | Nawaratne |

*Table 5 continued*

| | pERK1/2 interaction assays* | | | [³H]-QNB interaction binding assays† | | | Study |
|---|---|---|---|---|---|---|---|
| Y443A$^{7.43}$ | 0.50 ± 0.16 ‡‡ | N.D. | N.T. | 3.36 ± 0.01 ‡‡ | 6.22 ± 0.05 ‡‡ | 1.16 ± 0.01 ‡‡ | Nawaratne |

Values represent the mean ± SEM from three or more independent experiments with the number of individual experimental replicates from the current study shown in parenthesis.

N.T.: not tested; N.D.: not determined; N.R.: no response; ACh, acetylcholine.

*Data and analysis from pERK1/2 assays were generated in the current study, Nawaratne et al, J. Bio. Chem. 2010, and Leach et al, Mol. Pharm. 2011. log $\tau$ LY298 and log $\alpha\beta$ were calculated using a simplified operational model of allosterism. log $\tau$ ACh were calculated from the operational model of agonism.

†Data and analysis from [³H]-QNB interaction binding assays were generated in Nawaratne et al, J. Bio. Chem. 2010, Leach et al, Mol. Pharm., and Thal et al, Nature 2016.

‡log $\tau_C$ = logarithm of the operational efficacy parameter corrected for receptor expression using the maximum number of receptor binding sites as previously determined from Nawaratne et al, J. Bio. Chem. 2010, Leach et al, Mol. Pharm., and Thal et al, Nature 2016.

§Logarithm of the functional cooperativity factor between ACh and LY298.

¶Negative logarithm of the orthosteric (pK$_i$) or allosteric (pK$_B$) equilibrium dissociation constant.

**Logarithm of the binding cooperativity factor between ACh and LY298.

††Values of log $\tau_C$ ACh, log $\tau_C$ LY298, and log $\alpha\beta$ that were calculated in this study. Other parameters are from *Thal et al., 2016.*

‡‡Values are significantly different from WT M$_4$ mAChR as determined in previous studies.

§§All values are from Nawaratne et al, J. Bio. Chem. 2010 with log $\tau$ corrected for receptor expression.

¶¶All values are from Leach et al, Mol. Pharm. 2011.

***Parameters determined from the full Operational Model of Allosterism.

†††Values are significantly different from WT M$_4$ mAChR (p<0.05) calculated by a one-way ANOVA with a Dunnett's post-hoc test.

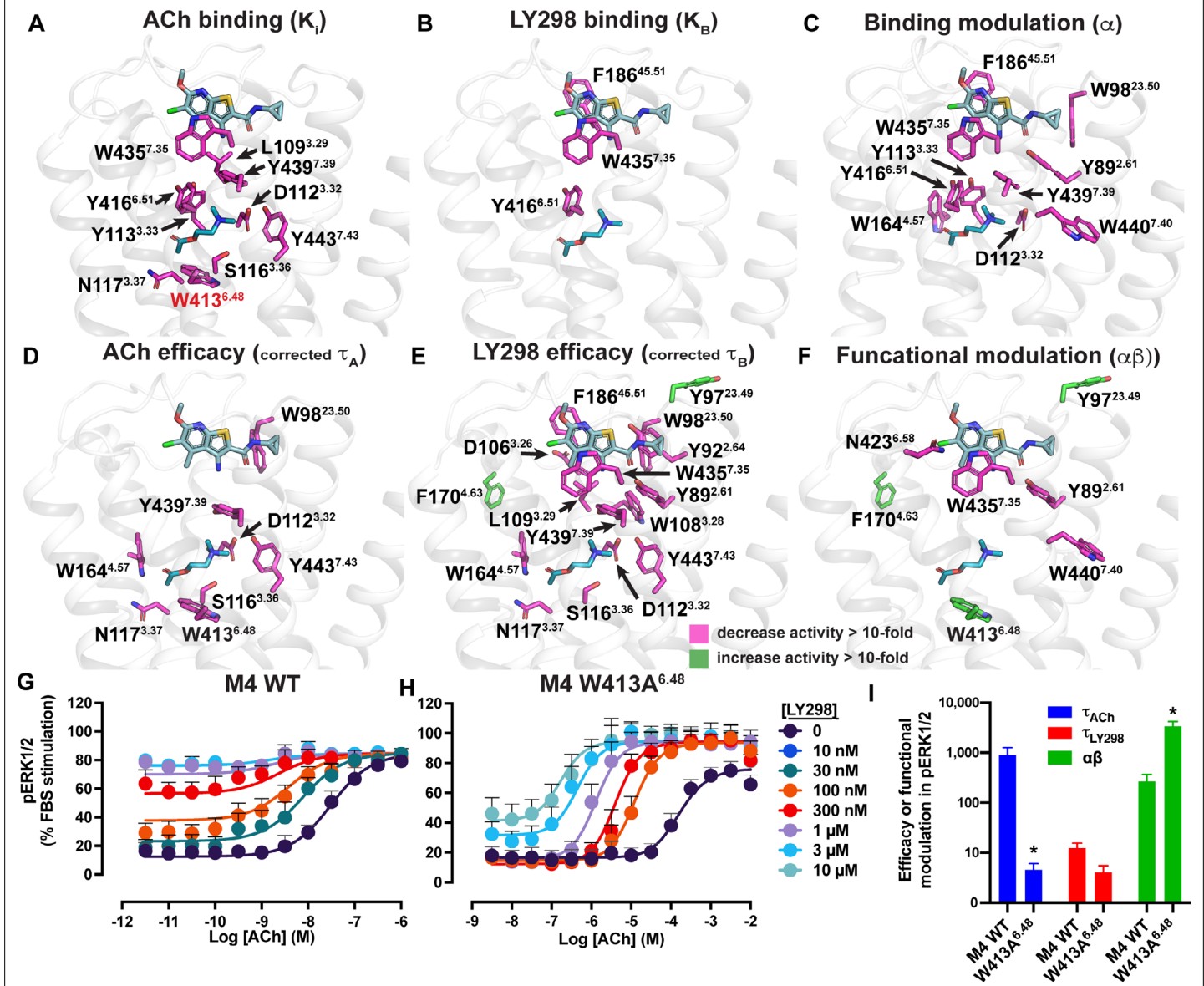

**Figure 6.** Residues involved in binding, agonism, and modulation of acetylcholine (ACh) and LY298. (**A–F**) $M_4$ muscarinic acetylcholine receptor (mAChR) alanine point mutations that increase (green colored sticks) or decrease (pink colored sticks) (**A**) ACh binding, (**B**) LY298 binding, (**C**) binding modulation between ACh and LY298, (**D**) ACh efficacy, (**E**) LY298 efficacy, (**F**) and functional modulation by values more than tenfold. Efficacy values are corrected for receptor expression (*Gregory et al., 2010*) using receptor expression data from *Thal et al., 2016*. Quantitative data used to identify key residues are from both the current study and previous studies as summarized in *Table 5* (*Leach et al., 2011*; *Nawaratne et al., 2010*; *Thal et al., 2016*). (**G–I**) pERK1/2 concentration response curves for interaction of ACh and LY298 at (**G**) WT and (**H**) W413A$^{6.48}$ $M_4$ mAChR with (**I**) values of efficacy and functional modulation. *Indicates statistical significance ($p<0.05$) relative to WT as determined by a one-way ANOVA with a Dunnett's post-hoc test that includes the other $M_4$ mAChR mutants. Data shown are mean ± SEM from three or more experiments performed in duplicate with the pharmacological parameters determined from a global fit of the data.

The online version of this article includes the following figure supplement(s) for figure 6:

**Figure supplement 1.** Concentration–response curves between acetylcholine (ACh) and LY298 at $M_4$ muscarinic acetylcholine receptor (mAChR) mutants.

**Figure supplement 2.** Interaction assays of agonists and positive allosteric modulators (PAMs) at the W413A$^{6.48}$ $M_4$ muscarinic acetylcholine receptor (mAChR) in a TruPath assay.

is comparable to LY298 at the human $M_4$ mAChR (*Figure 7—figure supplements 1 and 2*, *Table 1*). Therefore, we aimed to determine whether our prior findings could be used to explain the selectivity of VU154 between the human and mouse receptors.

The amino acid sequences of the human and mouse $M_4$ mAChRs are highly conserved, with most of the differences occurring between the long third intracellular loop and the N- and C- termini. As shown in *Figure 7A*, only three residues differ between the human and mouse $M_4$ mAChR with respect to the transmembrane domain. Specifically, residue V91 (L in mouse) at the top of TM2 points into the lipid bilayer, and D432 and T433 (E and R in mouse), which are located at the top of TM7 and form part of the allosteric binding site near VU154.

Previous work suggested that residues D432 and T433 were important for differences in the species selectivity of LY298 (*Chan et al., 2008*). As such, we examined two single D432E and T433R mutants and a V91L/D432E/T433R triple mutant of the human receptor, along with the mouse $M_4$ mAChR in radioligand binding and pERK1/2 experiments using Ipx and both PAMs (*Figure 7—figure supplements 1 and 2*, *Table 1*). For LY298, there were no statistically significant differences in binding or function between species and across the mutants that were more than threefold in effect. In contrast, VU154 had a tenfold higher binding affinity for the Ipx-bound mouse $M_4$ mAChR (compare *Figure 1G* with *Figure 7B*). The affinity of VU154 increased by 2.5-fold at the D432E and T433R mutants and the triple mutant matched the affinity of the mouse receptor (*Figure 7B*). In functional assays, similar results were observed for VU154 with Ipx at the mouse $M_4$ mAChR, with significant increases in the efficacy ($\tau_B$ – corrected for receptor expression), transduction coefficients ($\tau_B/K_B$), and functional modulation (αβ) (*Figure 7B*, *Figure 7—figure supplements 1 and 2*, *Table 1*). Relative to the WT $M_4$ mAChR, the efficacy (*Figure 7C*), transduction coefficients, and functional modulation of VU154 increased for all of the mutants (*Figure 7—figure supplements 1 and 2*, *Table 1*); however, none of the values fully matched the mouse receptor. Nevertheless, these results indicate that V91L, D432E, and T433R play a key role in mediating the species selectivity of VU154.

Our prior findings suggest the robust allosteric activity of LY298 at the human $M_4$ mAChR was due to stable interactions with the receptor. As a proof-of-principle, we questioned whether GaMD simulations would produce a stable binding mode for VU154 with D432E and T433R mutations to the VU154-Ipx-bound $M_4$R-$G_{i1}$ cryo-EM structure that was similar to our previously observed stable binding pose of LY298 (*Figure 4*). Excitingly, both the D432E and T433R mutants resulted in a dynamic profile of VU154 that matched our GaMD simulations of LY298 from the LY298-Ipx-bound $M_4$R-$G_{i1}$ cryo-EM structure, including stabilized VU154 binding, constrained $\chi_2$ rotamer conformations of W435[7.35] and W413[6.48], and stable binding interactions with Y89[2.61], Y439[7.39], Q184[45.49], and F186[45.51] (*Figure 7D–K*, *Figure 7—figure supplement 3*, *Videos 9 and 10*). The GaMD simulations also suggest that a potential interaction between the mutant residue T433R and the sulfoxide group of VU154 was more stable (5.2 ± 1.5 Å; *Figure 7—figure supplement 3I*) versus the WT residue T433 (6.56 ± 2.1 Å, *Figure 4—figure supplement 1J*), albeit the distance of this interaction was far apart and would be better validated by structure determination of VU154 with the mouse $M_4$ mAChR.

Collectively, these findings reiterate the importance of receptor dynamics in the determination of allosteric modulator selectivity as even subtle differences in amino acid residues between species may result in profound changes in overall stability of the same PAM-agonist-receptor complex.

## Discussion

Major advances have been made in recent years in the appreciation of the role of GPCR allostery and its relevance to modern drug discovery (*Changeux and Christopoulos, 2016*; *Wootten et al., 2013*). Despite an increase in the number of reported high-resolution GPCR structures bound to allosteric ligands (*Thal et al., 2018*), there remains a paucity of molecular-level details about the interplay between the complex chemical and pharmacological parameters that define allostery at GPCRs. By combining detailed pharmacology studies, multiple high-resolution cryo-EM structures of the $M_4$ mAChR bound to two pharmacologically different agonists and PAMs, and GaMD simulations, we have now provided exquisite in-depth insights into the relationship between both structure and dynamics that govern multiple facets of GPCR allostery (*Figure 8A*).

Comparison of the ACh- and Ipx-bound $M_4$ mAChR structures revealed that Ipx bound in a smaller binding pocket (*Figure 3G and H*), and GaMD simulations showed that Ipx formed more stable interactions with the receptor (*Figure 3—figure supplement 1*). These observations likely explained

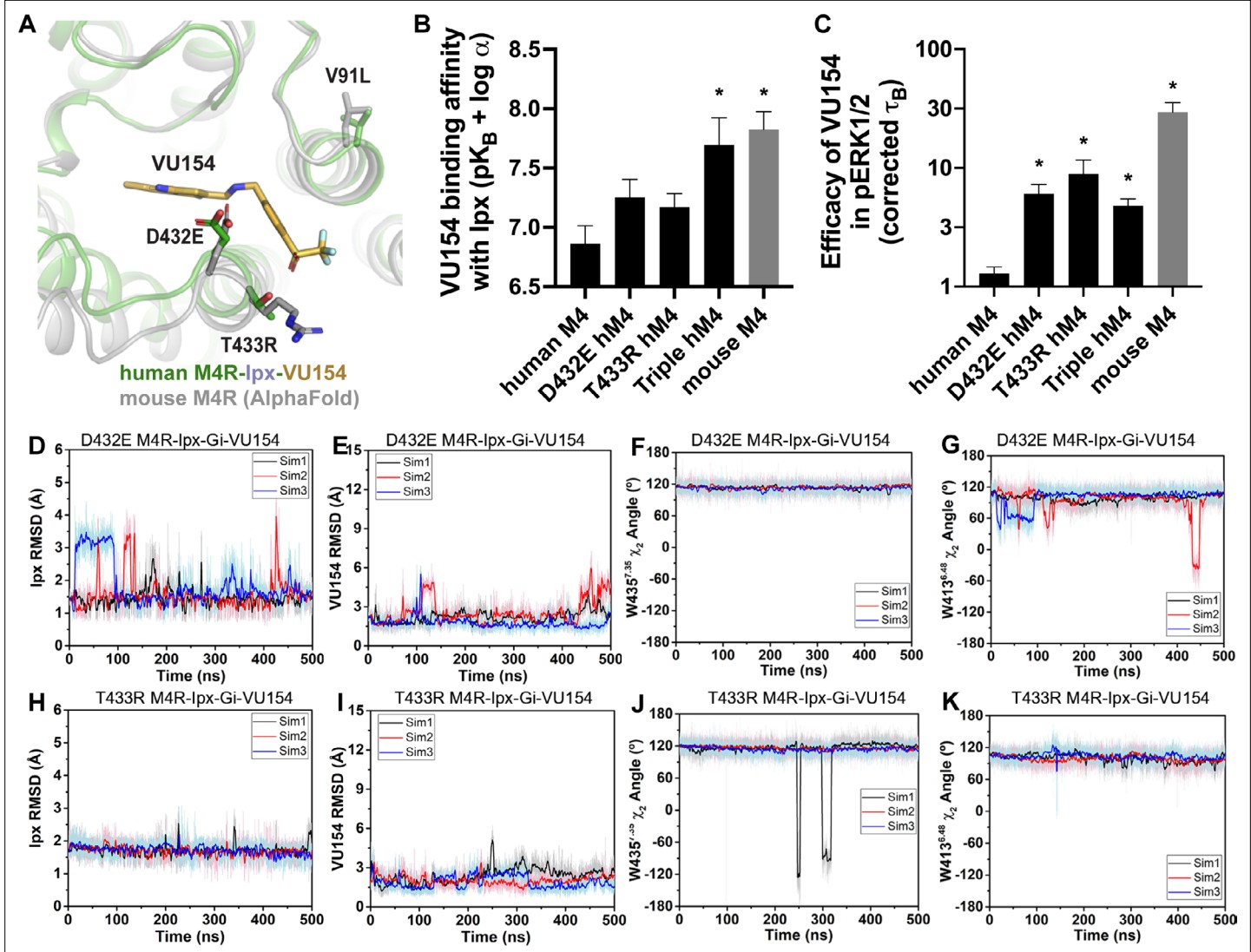

**Figure 7.** A molecular mechanism for the species selectivity for VU154. (**A**) Comparison of the cryo-electron microscopy (cryo-EM) structure of the human M4 muscarinic acetylcholine receptor (mAChR) bound to Ipx-VU154 with the AlphaFold model of the mouse M4 mAChR (*Jumper et al., 2021*; *Varadi et al., 2022*). The three residues that differ between species and within the core 7TM bundle from the human receptor (V91, D432, and T433) are shown as sticks along with the corresponding residues from the mouse receptor. (**B**) The binding affinity of VU154 for the Ipx-bound conformation ($pK_{B-Ipx} = pK_B + \alpha$) determined from [³H]-NMS binding experiments. Values calculated with data from *Figure 7—figure supplement 1* with propagated error. (**C**) Efficacy of VU154 ($\tau_B$ – corrected for receptor expression) of pERK1/2 signaling from data in *Figure 7—figure supplement 2*. (**D–K**) Time courses of obtained from Gaussian accelerated molecular dynamics (GaMD) simulations of the (**D–G**) D432E and (**H–K**) T433R mutant $M_4$R-Ipx-$G_{i1}$-VU154 systems with (**D, H**) Ipx RMSDs, (**E, I**), VU154 root mean square deviations (RMSDs), (**F, J**) W435$^{7.35}$ $\chi_2$ angle, and (**G, K**) W413$^{6.48}$ $\chi_2$ angle. Data shown are mean ± SEM from three or more experiments performed in duplicate with the pharmacological parameters determined from a global fit of the data. *Indicates statistical significance (p<0.05) relative to WT as determined by a one-way ANOVA with a Dunnett's post-hoc test.

The online version of this article includes the following source data and figure supplement(s) for figure 7:

**Source data 1.** Related to *Figure 7*.

**Figure supplement 1.** Binding parameters of positive allosteric modulators (PAMs) at the human and mouse $M_4$ muscarinic acetylcholine receptors (mAChRs).

**Figure supplement 1—source data 1.** Related to *Figure 7—figure supplement 1*.

**Figure supplement 2.** Functional parameters of the positive allosteric modulators (PAMs) at the human and mouse $M_4$ muscarinic acetylcholine receptors (mAChRs) in pERK1/2 signaling assays.

**Figure supplement 2—source data 1.** Related to *Figure 7—figure supplement 2*.

**Figure supplement 3.** Gaussian accelerated molecular dynamics (GaMD) simulations of D432E and T433R human $M_4$ muscarinic acetylcholine receptor (mAChR) mutants.

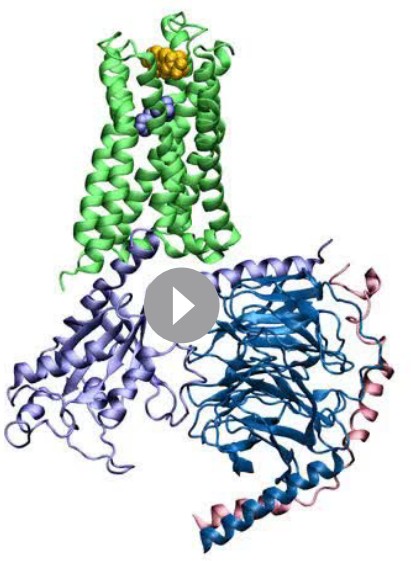

time: 0 ns

**Video 9.** Movie from one VU154-Ipx-M₄R(D432E)-G$_{i1}$ Gaussian accelerated molecular dynamics (GaMD) simulation.

https://elifesciences.org/articles/83477/figures#video9

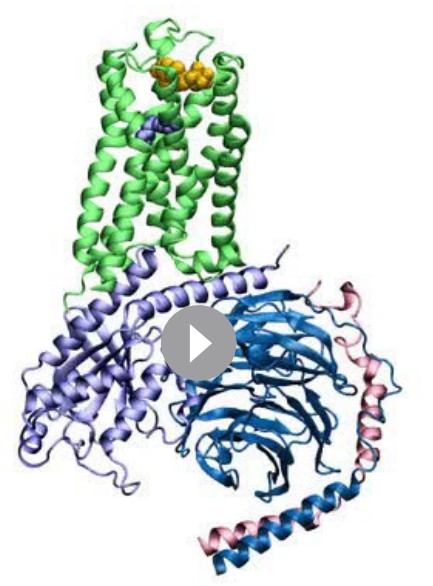

time: 0 ns

**Video 10.** Movie from one VU154-Ipx-M4R(T433R)-G$_{i1}$ Gaussian accelerated molecular dynamics (GaMD) simulation.

https://elifesciences.org/articles/83477/figures#video10

why Ipx exhibited greater than 1000-fold higher binding affinity than ACh (*Figure 1D*), being consistent with studies of other agonists at the β₁-adrenoceptor and the M₁ mAChR (*Brown et al., 2021*; *Warne et al., 2019*; *Figure 8B*). The observation that ACh was a more efficacious agonist than Ipx (*Table 1*) yet bound with lower affinity and less stable interactions than Ipx was paradoxical. *Kenakin and Onaran, 2002* previously opined on the paradox between ligand binding affinity and efficacy and showed via simulations that, in general, there was a negative correlation between binding affinity and efficacy. One interpretation of these results was that the ACh-bound M₄ mAChR more readily sampled receptor conformations that engaged with the transducers (*Manglik et al., 2015*). Similarly, the ACh-bound M₄ mAChR may also have faster G protein turnover than Ipx due to Ipx-M₄R-G$_{i1}$ forming a more stable ternary complex (*Furness et al., 2016*; *Figure 8B*).

It is worth noting that structures of GPCRs bound to agonists with different pharmacological properties (full, partial, and biased agonists) have now been reported for some GPCRs (*Liang et al., 2018a*; *Masureel et al., 2018*; *McCorvy et al., 2018*; *Ring et al., 2013*; *Wacker et al., 2013*; *Warne et al., 2012*; *Wingler et al., 2019*). However, insights gained from such cryo-EM and X-ray crystallography structures may be limited due to the role that the bound transducer plays on the observed final receptor conformation, and not necessarily due solely to the properties of the ligand. The ultimate underlying conformational differences, therefore, are likely to be subtle and dynamic (*Seyedabadi et al., 2022*), requiring application of additional techniques such as NMR spectroscopy, single-molecule FRET and MD simulations for furthering our understanding (*Cao et al., 2021*; *Cong et al., 2021*; *Gregorio et al., 2017*; *Huang et al., 2021*; *Katayama et al., 2021*; *Liu et al., 2012b*; *Solt et al., 2017*; *Sušac et al., 2018*; *Xu et al., 2023*; *Ye et al., 2016*).

Indeed, if considering this issue from the perspective of allosteric modulators of GPCRs, our study highlights that two PAMs with distinctly different pharmacological profiles (*Figure 1*) may bind to and stabilize receptor conformations that were very similar when viewed as static structures (*Figure 4*). Yet, in contrast, the 3DVA analysis from our cryo-EM structures suggested differences in the dynamics of the cryo-EM structures that were explored further in GaMD simulations (*Figure 4C*) and revealed that LY298 had a more stable binding pose and interactions with the receptor than VU154 in the PAM-agonist–receptor–transducer-bound conformation. These

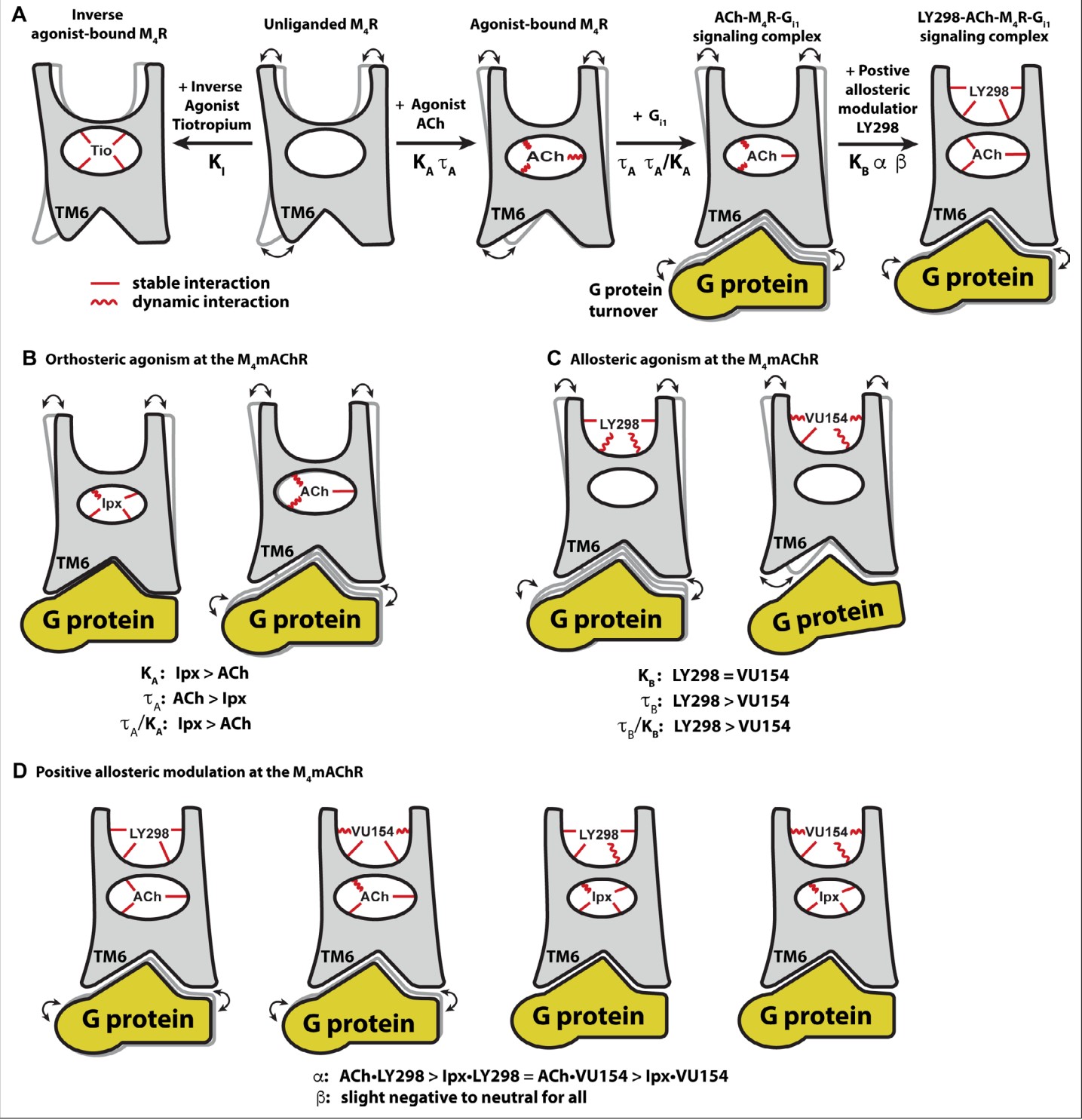

**Figure 8.** Conformational dynamics of the allostery at $M_4$ muscarinic acetylcholine receptor (mAChR) signaling complexes. (**A**) A schematic cartoon illustrating the conformational states of the ligands and the $M_4$ mAChR when bound to different types of ligands and transducer, along with the resulting dynamic profiles. Pharmacological parameters related to each conformational change are shown. Stable ligand–receptor interactions are denoted by a straight line and less-stable (more dynamic) interactions are denoted by a wavy line. (**B**) Iperoxo (Ipx) bound the $M_4$ mAChR with a higher affinity and more stability than ACh but had lower efficacy. ACh being more loosely bound and coupled to G protein may facilitate more G protein turnover accounting for its higher efficacy. (**C**) LY298 and VU154 bound to the $M_4$ mAChR with similar affinity for the receptor, but LY298 was found to bind more stably. LY298 had a higher efficacy than VU154, suggesting that allosteric agonism at the $M_4$ mAChR is mediated by stabilization of the extracellular vestibule (ECV). (**D**) The positive allosteric modulators (PAMs) LY298 and VU154 display robust binding modulation at the $M_4$ mAChR with LY298 having

*Figure 8 continued on next page*

*Figure 8 continued*

a stronger allosteric effect. Both PAMs displayed stronger binding modulation with the agonist ACh versus Ipx, an example of probe dependence. Both PAMs also displayed a slight negative to neutral effect on the efficacy of the agonists, suggesting that their mechanism of action is largely through binding.

observations were consistent with LY298 having greater positive binding cooperativity than VU154 (*Figure 1E*) and suggest that GaMD simulations of GPCRs bound to allosteric ligands could be an extremely valuable tool for drug discovery and optimization (*Bhattarai and Miao, 2018*).

Pharmacological analysis revealed that LY298 is a better PAM-agonist than VU154 with respect to efficacy (*Figure 1H*) in the $G_{i1}$ TruPath and pERK1/2 signaling assays (*Figure 1—figure supplement 2B*). GaMD simulations of the PAM–receptor–transducer and PAM–receptor bound complexes, again showed that LY298 more stably interacted with the receptor (*Figure 4*) and in the absence of G protein better stabilized the duration of the active conformation of the receptor (*Figure 5*). These findings were not contradictory to our above findings that ACh was more efficacious than Ipx despite having weaker interactions with the receptor because when the affinity of the ligands was accounted for in the transduction coupling coefficients, the rank order was Ipx >> ACh ~ LY298 > VU154 (*Figure 1I*). Furthermore, these results were in accordance with the observations of Kenakin and Onaran that ligands with the same binding affinity can also have differing efficacies (and vice versa). In addition, the mechanism of agonism for allosteric ligands that bind to the ECV may differ (*Xu et al., 2021*). Prior work by *DeVree et al., 2016* established that allosteric coupling of G proteins to the unliganded active receptor conformation promoted closure of the ECV region. This allosteric coupling is reciprocal and stabilizing the ECV region by PAMs likely leads to increased efficacy (*Figure 8*).

The PAMs, LY298 and VU154, also displayed stronger allosteric effects with ACh than with Ipx, an observation known as probe dependence (*Figure 1E–G*). Probe dependence can have substantial implications on how allosteric ligands are detected, validated, and their potential therapeutic utility (*Kenakin, 2005*). Examples of probe dependence are not limited to studies on mAChRs and have been observed across multiple receptor families (*Christopoulos, 2014*; *Gentry et al., 2015*; *Pani et al., 2021*; *Slosky et al., 2020*; *Wang et al., 2021b*). GaMD simulations comparing the PAMs co-bound with either Ipx or ACh showed that the PAMs had a stabilizing effect on ACh, whereas the stability of Ipx was slightly reduced by the PAMs likely because the binding of Ipx was already stable. This is a sensible explanation from thermodynamic principles. Another explanation invokes the two-state receptor model (*Canals et al., 2011*), which stipulates that the degree of positive modulation for PAMs increases with an increase in the efficacy of the agonists. The pharmacology data support this model as ACh was more efficacious than Ipx and was better modulated by both PAMs (*Figure 8D*). These observations are also consistent with recent studies that suggest that conformational dynamics between agonist and receptor are important for functional signaling (*Bumbak et al., 2020*; *Cary et al., 2022*; *Deganutti et al., 2022*; *O'Connor et al., 2015*).

The findings presented here provide new insights into the allosteric signaling and allosteric modulation of GPCRs by combining the analytical analysis of multiple pharmacology assays with cryo-EM structures and GaMD simulations. Overall, these results provide a framework for future mechanistic studies and, ultimately, can aid in the discovery, design, and optimization of allosteric drugs as novel therapeutic candidates for clinical progression.

## Limitations of the study

The complexities of GPCR signaling cannot be fully explained by any single receptor or set of experiments. This study was limited to the investigation of two agonists and two PAMs at the human $M_4$ mAChR. Future studies will be required to determine how these results extrapolate to other classes of ligand, mAChR subtypes, and GPCRs. For instance, this study determined the structures of the $M_4$ mAChR bound with the ligands ACh, Ipx, Ipx-LY298, and Ipx-VU154. It is possible that structures of the $M_4$ mAChR bound with ACh-LY298 and ACh-VU154 could reveal different receptor conformations (although GaMD simulations already performed on their docked complexes and the conformational differences between the Ipx-bound cryo-EM structures suggest otherwise). Similarly, structures of the $M_4$ mAChR bound in complex with either PAM alone may provide better insights into direct allosteric agonism. However, we note that our attempt at determining an LY298-bound complex did not have sufficient stability for the determination of a high-resolution structure, as also supported by our GaMD

simulations. Additionally, our cryo-EM structures and MD-simulations utilized an $M_4$ mAChR sequence with a large portion of the third intracellular loop removed and were complexed with a dominant negative mutant of $G\alpha_{i1}$ and stabilized with the antibody scFv16. This contrasts with our pharmacological characterization of the ligands that were performed on the WT $M_4$ mAChR. Further investigation into the molecular determinants of species selectivity is also warranted, as is the need for future experiments that incorporate the combined interplay between dynamics/kinetics of ligands, receptor, transducer recruitment and activation.

## Materials and methods

### Bacterial strains

DH5α (New England Biolabs) and DH10bac (Thermo Fisher Scientific) *Escherichia coli* cells were grown in LB at 37°C.

### Cell culture

Tni and Sf9 cells (Expression Systems) were maintained in ESF-921 media (Expression Systems) at 27°C. Flp-In Chinese hamster ovary (CHO) (Thermo Fisher Scientific) cells stably expressing human $M_4$ mAChR or mutant constructs were maintained in Dulbecco's modified Eagle's medium (DMEM, Invitrogen) containing 5% fetal bovine serum (FBS; ThermoTrace) and 0.6 μg/ml of Hygromycin (Roche) in a humidified incubator (37°C, 5% $CO_2$, 95% $O_2$). HEK293A cells were grown in DMEM supplemented with 5% FBS at 37°C in 5% $CO_2$. Cell lines were authenticated by vendor and confirmed negative for mycoplasma contamination using the Lonza MycoAlert Mycoplasma Detection Kit (#LT07-318).

### Radioligand binding assays

Flp-In CHO cells stably expressing $M_4$ mAChR constructs were seeded at 10,000 cells/well in 96-well white clear bottom isoplates (Greiner Bio-one) and allowed to adhere overnight at 37°C, 5% $CO_2$, and 95% $O_2$. Saturation binding assay was performed to quantify the receptor expression and equilibrium dissociation constant of the radioligand [³H]-NMS (PerkinElmer, specific activity 80 Ci/mmol). Briefly, plates were washed once with phosphate-buffered saline (PBS) and incubated overnight at room temperature (RT) with 0.01–10 nM [³H]-NMS in Hanks's balanced salt solution (HBSS)/10 mM HEPES (pH 7.4) in a final volume of 100 μl. For binding interaction assays, cells were incubated overnight at RT with a specific concentration of [³H]-NMS ($pK_D$ determined at each receptor in saturation binding) and various concentrations of ACh or Ipx in the absence or presence of increasing concentrations of each allosteric modulator. In all cases, nonspecific binding was determined by the coaddition of 10 μM atropine (Sigma). The following day, the assays were terminated by washing the plates twice with ice-cold 0.9% NaCl to remove the unbound radioligand. Cells were solubilized in 100 μl per well of Ultima Gold (PerkinElmer), and radioactivity was measured with a MicroBeta plate reader (PerkinElmer).

### G protein activation assay

Upon 60–80% confluence, HEK293A cells were transfected transiently using polyethylenimine (PEI, Polysciences) and 10 ng per well of each of pcDNA3.1-hM4 mAChR (WT or mutant), pcDNA5/FRT/TO-$G\alpha_{i1}$-RLuc8, pcDNA3.1-$\beta_3$, and pcDNA3.1-$G\gamma_9$-GFP2 at a ratio of 1:1:1:1 ratio with 40 ng of total DNA per well. Cells were plated at 30,000 cells per well into 96-well Greiner CELLSTAR white-walled plates (Sigma-Aldrich). 48 hr later, cells were washed with 200 μl phosphate buffer saline (PBS) and replaced with 70 μL of 1× HBSS with 10 mM HEPES. Cells were incubated for 30 min at 37°C before addition of 10 μl of 1.3 μM Prolume Purple coelenterazine (Nanolight Technology). Cells were further incubated for 10 min at 37C° before BRET measurements were performed on a PHERAstar plate reader (BMG Labtech) using 410/80 nm and 515/30 nm filters. Baseline measurements were taken for 8 min before addition of drugs or vehicle to give a final assay volume of 100 μl and further reading for 30 min. BRET signal was calculated as the ratio of 515/30 nm emission over 410/80 nm emission. The ratio was vehicle corrected using the initial 8 min of baseline measurements and then baseline corrected again using the vehicle-treated wells. Data were normalized using the maximum agonist response to allow for grouping of results using an area under the curve analysis in Prism. Data were analyzed at timepoints of 4, 10, and 30 min yielding similar results.

## Phospho-ERK1/2 assay

The level of phosphorylated extracellular signal-regulated protein kinase 1/2 (pERK1/2) was detected using the AlphaScreen SureFire Kit (PerkinElmer Life and Analytical Sciences). Briefly, FlpIn CHO cells stably expressing the receptor were seeded into transparent 96-well plates at a density of 20,000 cells/well and grown overnight at 37°C, 5% $CO_2$. Cells were washed with PBS and incubated in serum-free DMEM at 37°C for 4 hr to allow FBS-stimulated pERK1/2 levels to subside. Cells were stimulated with increasing concentrations of ACh or Ipx in the absence or presence of increasing concentrations of the allosteric modulator at 37°C for 5 min (the time required to maximally promote ERK phosphorylation for each ligand at each $M_4$ mAChR construct in the initial time-course study; data not shown). For all experiments, stimulation with 10% (*v/v*) FBS for 5 min was used as a positive control. The reaction was terminated by the removal of media and lysis of cells with 50 µl of the SureFire lysis buffer (TGR Biosciences). Plates were then agitated for 5 min and 5 µl of the cell lysate was transferred to a white 384-well ProxiPlate (Greiner Bio-one) followed by the addition of 5 µl of the detection buffer (a mixture of activation buffer:reaction buffer:acceptor beads:donor beads at a ratio of 50:200:1:1). Plates were incubated in the dark for 1 hr at 37°C followed by measurement of fluorescence using an Envision plate reader (PerkinElmer) with standard AlphaScreen settings. Data were normalized to the maximal response mediated by 10 µM ACh, Ipx, or 10% FBS.

## Purification of scFv16

Tni insect cells were infected with scFv16 baculovirus at a density of 4 million cells per ml and harvested at 60 hr post infection by centrifugation for 10 min at 10,000 × *g*. The supernatant was pH balanced to pH 7.5 by the addition of Tris pH 7.5, and 5 mM $CaCl_2$ was added to quench any chelating agents, then left to stir for 1.5 hr at RT. The supernatant was then centrifuged at 30,000 × *g* for 15 min to remove any precipitates. 5 ml of EDTA-resistant Ni resin (Cytivia) was added and incubated for 2 hr at 4°C while stirring. Resin was collected in a glass column and washed with 20 column volumes (CVs) of high salt buffer (20 mM HEPES pH 7.5, 500 mM NaCl, 20 mM imidazole) followed by 20 CVs of low salt buffer (20 mM HEPES pH 7.5, 100 mM NaCl, 20 mM imidazole). Protein was then eluted using 8 CV of elution buffer (20 mM HEPES pH 7.5, 100 mM NaCl, 250 mM imidazole) until no more protein was detected using Bradford reagent (Bio-Rad Laboratories). Protein was concentrated using a 10 kDa Amicon filter device (Millipore) and aliquoted into 1 mg aliquots for further use.

## Expression and purification of $M_4R$-$G_{i1}$-scFv16 complexes

The human $M_4$ mAChR with residues 242–387 of the third intracellular loop removed and the N-terminal glycosylation sites (N3, N9, N13) mutated to D was expressed in Sf9 insect cells, and human $DNG_{\alpha i1}$ and His6-tagged human $G_{\beta 1 \gamma 2}$ were co-expressed in Tni insect cells. Cell cultures were grown to a density of 4 million cell per ml for Sf9 cells and 3.6 million per ml for Tni cells and then infected with either $M_4$ mAChR baculovirus or both $G_{\alpha i1}$ and $G_{\beta 1 \gamma 2}$ baculovirus, at a ratio of 1:1. $M_4$ mAChR expression was supplemented with 10 mM atropine. Cultures were grown at 27°C and harvested by centrifugation 60–72 hr (48 hr for Hi5 cells) post infection. Cells were frozen and stored at –80°C for later use. 1–2 l of the frozen cells were used for each purification.

Cells expressing $M_4$ mAChR were thawed at RT and then dounced in the solubilization buffer containing 20 mM HEPES pH 7.5, 10% glycerol, 750 mM NaCl, 5 mM $MgCl_2$, 5 mM $CaCl_2$, 0.5% LMNG, 0.02% CHS, 10 µM atropine, and cOmplete Protease Inhibitor Cocktail (Roche) until homogeneous. The receptor was solubilized for 2 hr at 4°C while stirring. The insoluble material was removed by centrifugation at 30,000 × *g* for 30 min followed by filtering the supernatant and batch-binding immobilization to M1 anti-flag affinity resin, previously equilibrated with high salt buffer, for 1 hr at RT. The resin with immobilized receptor was then washed using a peristaltic pump for 30 min at 2 ml/min with high salt buffer: 20 mM HEPES pH 7.5, 750 mM NaCl, 5 mM $MgCl_2$, 5 mM $CaCl_2$, 0.5% lauryl maltose neopentyl glycol (LMNG, Anatrace), 0.02% cholesterol hemisuccinate (CHS, Anatrace) followed by low salt buffer: 20 mM HEPES pH 7.5, 100 mM NaCl, 5 mM $MgCl_2$, 5 mM $CaCl_2$, 0.5% LMNG, 0.02% CHS, and an agonist (5 µM Ipx, 1 µM Ipx with 10 µM VU154, or 100 µM ACh). While the receptor was immobilized on anti-FLAG resin, the $DNG\alpha_{i1}$ cell pellet was thawed, dounced, and solubilized in the solubilization buffer containing 20 mM HEPES pH 7.5, 100 mM NaCl, 5 mM $MgCl_2$, 5 mM $CaCl_2$, 0.5% LMNG, 0.02% CHS, apyrase (five units), and cOmplete Protease Inhibitor Cocktail. $DNG\alpha_{i1}$ was solubilized for 2 hr at 4°C followed by the centrifugation at 30,000 × *g* for 30 min to

remove the insoluble material. Supernatant was filtered through a glass fiber filter (Millipore) and then added to the receptor bound to anti-Flag resin. Apyrase (five units), scFv16, and agonist (either 1 µM Ipx, 1 µM Ipx with 10 µM VU154, or 100 µM ACh) were added and incubated for 1 hr at RT with gentle mixing. The anti-FLAG resin was then loaded onto a glass column and washed with approximately 20 CVs of washing buffer: 20 mM HEPES pH 7.4, 100 mM NaCl, 5 mM MgCl$_2$, 5 mM CaCl$_2$, 0.01% LMNG, 0.001% CHS, agonist (1 µM Ipx, 1 µM Ipx with 10 µM VU154, or 100 µM ACh). Complex was eluted with size-exclusion chromatography (SEC) buffer: 20 mM HEPES pH 7.5, 100 mM NaCl, 5 mM MgCl$_2$, 0.01% LMNG, 0.001% CHS and agonist (1 µM Ipx, or 1 µM Ipx with 10 µM VU154, or 100 µM ACh) with the addition of 10 mM EGTA and 0.1 mg/mL FLAG peptide. After the elution, an additional 1–2 mg of scFv16 was added and shortly incubated on ice before concentrating using a 100 kDa Amicon filter to a final volume of 500 µl. The sample was filtered using a 0.22 µm filter followed by SEC using a Superdex 200 increase 10/300 column (Cytivia) using SEC buffer. For the ACh- and VU154-Ipx-bound samples, the fractions containing protein were concentrated again and re-run over SEC using a buffer with half the amount of detergent in order to remove empty micelles. Samples were concentrated and flash frozen using liquid nitrogen. In case of the LY298-Ipx-bound sample, the sample was purified with 1 µM Ipx only. After SEC, the sample was then split in half, where one half was incubated with approximately 1.6 µM LY298 at 4°C overnight, and then concentrated and flash frozen in liquid nitrogen.

## EM sample preparation and data acquisition

Samples (3 µl) were applied to glow-discharged Quantifoil R1.2/1.3 Cu/Rh 200 mesh grids (Quantifoil) (M4R-G$_{i1}$-Ipx and M4R-G$_{i1}$-Ipx-LY298) or UltrAuFoil R1.2/1.3 Au 300 mesh grids (Quantifoil) (M4R-G$_{i1}$-Ipx-VU154 and M4R-G$_{i1}$-Ach) and were vitrified on a Vitrobot Mark IV (Thermo Fisher Scientific) set to 4°C and 100% humidity and 10 s blot time. Data were collected on a Titan Krios G3i 300 kV electron microscope (Thermo Fisher Scientific) equipped with GIF Quantum energy filter and K3 detector (Gatan). Data acquisition was performed in EFTEM NanoProbe mode with a 50 µM C2 aperture at an indicated magnification of ×105,000 with zero-loss slit width of 25 eV. The data were collected automatically with homemade scripts for SerialEM performing a nine-hole beam-image shift acquisition scheme with one exposure in the center of each hole. Experimental parameters specific to each collected data set is listed in *Table 2*.

## Image processing

Specific details for the processing of each cryo-EM data set are shown in *Figure 2—figure supplement 2*. Image frames for each movie were motion corrected using MotionCor2 (*Zheng et al., 2017*) and contrast transfer function (CTF)-estimated using GCTF (*Zhang, 2016*). Particles were picked from corrected micrographs using crYOLO (*Wagner et al., 2019*) or RELION-3.1 software *Zivanov et al., 2018* followed by reference-free 2D and 3D classifications. Particles within bad classes were removed and remaining particles subjected to further analysis. Resulting particles were subjected to Bayesian polishing, CTF refinement, 3D auto-refinement in RELION, followed by another round of 3D classification and 3D refinement that yielded the final maps (*Zivanov et al., 2018*). Local resolution was determined from RELION using half-reconstructions as input maps. Due to the high degree of conformational flexibility between the receptor and G protein, a further local refinement was performed in cryoSPARC for the ACh-bound M$_4$R-complex. A receptor-focused map was generated (2.75 Å), which was used to generate a PDB model of the ACh-bound M$_4$R.

## Model building and refinement

An initial M$_4$R template model was generated from our prior modeling studies of the M$_4$ mAChR that was based on an active state M$_2$ mAChR structure (PBD: 4MQT) (*Kruse et al., 2013*). An initial model for dominant negative Gα$_{i1}$Gβ$_1$Gγ$_2$ was from a structure in complex with Smoothend (PDB: 6OT0) (*Qi et al., 2019*) and scFv16 from the X-ray crystal structure in complex with heterotrimeric G protein (PDB: 6CRK) (*Maeda et al., 2018*). Models were fit into EM maps using UCSF Chimera (*Pettersen et al., 2004*), and then rigid-body-fit using PHENIX (*Liebschner et al., 2019*), followed by iterative rounds of model rebuilding in Coot (*Casañal et al., 2020*) and ISOLDE (*Croll, 2018*), and real-space refinement in PHENIX. Restrains for all ligands were generated from the GRADE server (https://grade.globalphasing.org). Model validation was performed with MolProbity (*Williams et al., 2018*) and

the wwPDB validation server (*Berman et al., 2003*). Figures were generated using UCSF Chimera (*Pettersen et al., 2004*), Chimera X (*Pettersen et al., 2021*), and PyMOL (Schrödinger).

## Cryo-EM 3D variability analysis

3D variability analysis (3DVAR) was performed to access and visualize the dynamics within the cryo-EM datasets of the $M_4$ mAChR complexes, as previously described using cryoSPARC (*Punjani and Fleet, 2021*). The polished particle stacks were imported into cryoSPARC, followed by 2D classification and 3D refinement using the respective low-pass-filtered RELION consensus maps as an initial model. 3DVA was analyzed in three components with 20 volume frames of data per component of motion. Output files were visualized using UCSF Chimera (*Pettersen et al., 2004*).

## Gaussian accelerated molecular dynamics (GaMD)

GaMD enhances the conformational sampling of biomolecules by adding a harmonic boost potential to reduce the system energy barriers (*Miao et al., 2015*). When the system potential $V(\vec{r})$ is lower than a reference energy E, the modified potential $V(\vec{r})$ of the system is calculated as

$$V(\vec{r}) = V(\vec{r}) + \Delta V(\vec{r})$$

$$\Delta V(\vec{r}) = \begin{cases} \frac{1}{2}k\left(E - V(\vec{r})\right)^2, & V(\vec{r}) < E \\ 0, & V(\vec{r}) \geq E, \end{cases} \tag{1}$$

where k is the harmonic force constant. The two adjustable parameters E and k are automatically determined on three enhanced sampling principles. First, for any two arbitrary potential values $v_1(\vec{r})$ and $v_2(\vec{r})$ found on the original energy surface, if $V_1(\vec{r}) < V_2(\vec{r})$, $\Delta V$ should be a monotonic function that does not change the relative order of the biased potential values; that is, $V_1(\vec{r}) < V_2(\vec{r})$. Second, if $V_1(\vec{r}) < V_2(\vec{r})$, the potential difference observed on the smoothened energy surface should be smaller than that of the original; i.e., $V_2(\vec{r}) - V_1(\vec{r}) < V_2(\vec{r}) - V_1(\vec{r})$. By combining the first two criteria and plugging in the formula of $V(\vec{r})$ and $\Delta V$, we obtain

$$V_{max} \leq E \leq V_{min} + \frac{1}{k}, \tag{2}$$

where $V_{min}$ and $V_{max}$ are the system minimum and maximum potential energies. To ensure that *Equation 2* is valid, k has to satisfy $k \leq 1/\left(V_{max} - V_{min}\right)$. Let us define $k = k_0 \cdot 1/\left(V_{max} - V_{min}\right)$, then $0 k_0 \leq 1$. Third, the standard deviation (SD) of $\Delta V$ needs to be small enough (i.e. narrow distribution) to ensure accurate reweighting using cumulant expansion to the second order: $\sigma_{\Delta V} = k\left(E - V_{avg}\right)\sigma_V \leq \sigma_0$, where $V_{avg}$ and $\sigma_V$ are the average and SD of $\Delta V$ with $\sigma_0$ as a user-specified upper limit (e.g. $10k_BT$) for accurate reweighting. When E is set to the lower bound $E = V_{max}$ according to *Equation 2*, $k_0$ can be calculated as

$$k_0 = min\left(1.0, k_0'\right) = min\left(1.0, \frac{\sigma_0}{\sigma_V} \cdot \frac{V_{max} - V_{min}}{V_{max} - V_{avg}}\right), \tag{3}$$

Alternatively, when the threshold energy E is set to its upper bound $E = V_{min} + 1/k$, $k_0$ is set to

$$k_0 = k_0'' \equiv \left(1 - \frac{\sigma_0}{\sigma_V}\right) \cdot \frac{V_{max} - V_{min}}{V_{avg} - V_{min}}, \tag{4}$$

If $k_0''$ is calculated between 0 and 1. Otherwise, $k_0$ is calculated using *Equation 3*.

## Energetic reweighting of GaMD simulations

For energetic reweighting of GaMD simulations to calculate potential of mean force (PMF), the probability distribution along a reaction coordinate is written as $p(A)$. Given the boost potential $\Delta V(r)$ of each frame, $p(A)$ can be reweighted to recover the canonical ensemble distribution $p(A)$, as

$$p\left(A_j\right) = p\left(A_j\right) \frac{\left\langle e^{\beta \Delta V(r)}\right\rangle_j}{\sum_{i=1}^{M}\left\langle p\left(A_i\right) e^{\beta \Delta V(r)}\right\rangle_i}, j = 1, \ldots, M, \tag{5}$$

where $M$ is the number of bins, $\beta = k_B T$, and $\left\langle e^{\beta \Delta V(r)}\right\rangle_j$ is the ensemble-averaged Boltzmann factor of $\Delta V(r)$ for simulation frames found in the $j$th bin. The ensemble-averaged reweighting factor can be approximated using cumulant expansion:

$$\left\langle e^{\beta \Delta V(r)}\right\rangle = exp\left\{\sum_{k=1}^{\infty} \frac{\beta^k}{k!} C_k\right\}, \tag{6}$$

where the first two cumulants are given by

$$C_1 = \langle \Delta V \rangle,$$
$$C_2 = \left\langle \Delta V^2 \right\rangle - \langle \Delta V \rangle^2 = \sigma_v^2. \tag{7}$$

The boost potential obtained from GaMD simulations usually follows near-Gaussian distribution (**Miao and McCammon, 2017**). Cumulant expansion to the second order thus provides a good approximation for computing the reweighting factor (**Miao et al., 2015**; **Miao et al., 2014**). The reweighted free energy $F(A) = -k_B T l n p(A)$ is calculated as

$$F(A) = F(A) - \sum_{k=1}^{2} \frac{\beta^k}{k!} C_k + F_c, \tag{8}$$

where $F(A) = -k_B T l n p(A)$ is the modified free energy obtained from GaMD simulation and $F_c$ is a constant.

## System setup

The M$_4$R-ACh-G$_{i1}$, M$_4$R-Ipx-G$_{i1}$, M$_4$R-Ipx-G$_{i1}$-VU154, and M$_4$R-Ipx-G$_{i1}$-LY298 cryo-EM structures were used for setting up simulation systems. The scFv16 in the cryo-EM structures was omitted in all simulations. The initial structures of single mutant D432E and T433R mutant of M$_4$R-Ipx-G$_{i1}$-VU154 were obtained by mutating the corresponding residues in the M$_4$R-Ipx-G$_{i1}$-VU154 cryo-EM structure. The initial structures of M$_4$R-ACh-G$_{i1}$-VU154 and M$_4$R-ACh-G$_{i1}$-LY298 were obtained from M$_4$R-Ipx-G$_{i1}$-VU154 and M$_4$R-Ipx-G$_{i1}$-LY298 cryo-EM structures by replacing Ipx with ACh through alignment of receptors to the M4R-ACh-G$_{i1}$ cryo-EM structure. The initial structures of M$_4$R-G$_{i1}$-VU154 and M$_4$R-G$_{i1}$-LY298 were obtained by removing the corresponding Ipx agonist from the M$_4$R-Ipx-G$_{i1}$-VU154 and M$_4$R-Ipx-G$_{i1}$-LY298 cryo-EM structures. The initial structures of M$_4$R-VU154 and M$_4$R-LY298 were obtained by removing the corresponding Ipx agonist and G$_{i1}$ protein from the M$_4$R-Ipx-G$_{i1}$-VU154 and M$_4$R-Ipx-G$_{i1}$-LY298 cryo-EM structures. According to previous findings, intracellular loop (ICL) 3 is highly flexible and removal of ICL3 does not appear to affect GPCR function (**Dror et al., 2015**; **Dror et al., 2011**). The ICL3 was thus omitted as in the current GaMD simulations. Similar to a previous study, helical domains of the G$_{i1}$ protein missing in the cryo-EM structures were not included in the simulation models. This was based on earlier simulation of the β$_2$AR-G$_s$ complex, which showed that the helical domain fluctuated substantially (**Dror et al., 2015**). All chain termini were capped with neutral groups (acetyl and methylamide). All the disulfide bonds in the complexes (i.e. Cys108[3.25]-Cys185[45x50] and Cys426[ECL3]-Cys429[ECL3] in the M4R) that were resolved in the cryo-EM structures were maintained in the simulations. Using the *psfgen* plugin in VMD (**Humphrey et al., 1996**), missing atoms in protein residues were added and all protein residues were set to the standard CHARMM protonation states at neutral pH. For each of the complex systems, the receptor was inserted into a palmitoyl-oleoyl-phosphatidyl-choline (POPC) bilayer with all overlapping lipid molecules removed using the membrane plugin in VMD. The system charges were then neutralized at 0.15 M NaCl using the *solvate* plugin in VMD (**Humphrey et al., 1996**). The simulation systems were summarized in *Table 3*.

## Simulation protocol

The CHARMM36M parameter set (*Huang et al., 2017*; *Klauda et al., 2010*; *Vanommeslaeghe and MacKerell, 2015*) was used for the $M_4$ mAChRs, $G_{i1}$ proteins, and POPC lipids. Force field parameters of agonists ACh and Ipx, PAMs LY298 and VU154 were obtained from the CHARMM Param-Chem web server (*Vanommeslaeghe et al., 2012b*; *Vanommeslaeghe and MacKerell, 2012a*). Force field parameters with high penalty were optimized with FFParm (*Kumar et al., 2020*). GaMD simulations of these systems followed a similar protocol used in previous studies of GPCRs (*Draper-Joyce et al., 2021*; *Miao and McCammon, 2018*; *Miao and McCammon, 2016*). For each of the complex systems, initial energy minimization, thermalization, and 20 ns cMD equilibration were performed using NAMD2.12 (*Phillips et al., 2005*). A cutoff distance of 12 Å was used for the van der Waals and short-range electrostatic interactions and the long-range electrostatic interactions were computed with the particle-mesh Ewald summation method (*Darden et al., 1993*). A 2-fs integration time step was used for all MD simulations, and a multiple-time-stepping algorithm was used with bonded and short-range non-bonded interactions computed every time step and long-range electrostatic interactions every two-time steps. The SHAKE algorithm (*Ryckaert et al., 1977*) was applied to all hydrogen-containing bonds. The NAMD simulation started with equilibration of the lipid tails. With all other atoms fixed, the lipid tails were energy minimized for 1000 steps using the conjugate gradient algorithm and melted with a constant number, volume, and temperature (NVT) run for 0.5 ns at 310 K. The 12 systems were further equilibrated using a constant number, pressure, and temperature (NPT) run at 1 atm and 310 K for 10 ns with 5 kcal/(mol. Å$^2$) harmonic position restraints applied to the protein and ligand atoms. Final equilibration of each system was performed using a NPT run at 1 atm pressure and 310 K for 0.5 ns with all atoms unrestrained. After energy minimization and system equilibration, conventional MD simulations were performed on each system for 20 ns at 1 atm pressure and 310 K with a constant ratio constraint applied on the lipid bilayer in the X-Y plane.

With the NAMD output structure, along with the system topology and CHARMM36M force field files, the *ParmEd* tool in the AMBER package was used to convert the simulation files into the AMBER format. The GaMD module implemented in the GPU version of AMBER20 (Case et al. 2020) was then applied to perform the GaMD simulation. GaMD simulations of systems with $G_{i1}$ protein ($M_4$R-ACh-$G_{i1}$, $M_4$R-Ipx-$G_{i1}$, $M_4$R-Ipx-$G_{i1}$-VU154, $M_4$R-Ipx-$G_{i1}$-LY298, $M_4$R-ACh-$G_{i1}$-VU154, $M_4$R-ACh-$G_{i1}$-LY298, single mutant D432E and T433R mutants of $M_4$R-Ipx-$G_{i1}$-VU154) included an 8-ns short cMD simulation used to collect the potential statistics for calculating GaMD acceleration parameters, a 48-ns equilibration after adding the boost potential, and finally three independent 500-ns GaMD production simulations with randomized initial atomic velocities. The average and SD of the system potential energies were calculated every 800,000 steps (1.6 ns). GaMD simulations of $M_4$R-VU154 and $M_4$R-LY298 included a 2.4-ns short cMD simulation used to collect the potential statistics for calculating GaMD acceleration parameters, a 48-ns equilibration after adding the boost potential, and finally three independent 1000-ns GaMD production simulations with randomized initial atomic velocities. The average and SD of the system potential energies were calculated every 240,000 steps (0.48 ns). All GaMD simulations were run at the 'dual-boost' level by setting the reference energy to the lower bound. One boost potential is applied to the dihedral energetic term and the other to the total potential energetic term. The upper limit of the boost potential SD, $\sigma_0$ was set to 6.0 kcal/mol for both the dihedral and the total potential energetic terms. Similar temperature and pressure parameters were used as in the NAMD simulations.

## Simulation analysis

CPPTRAJ (*Roe and Cheatham, 2013*) and VMD (*Humphrey et al., 1996*) were used to analyze the GaMD simulations. The RMSDs of the agonist ACh and Ipx, PAM VU154 and LY298 relative to the simulation starting structures, the interactions between receptor and agonists/PAMs, distances between the receptor TM3 and TM6 intracellular ends were selected as reaction coordinates. Particularly, distances were calculated between the Cα atoms of residues Arg$^{3.50}$ and Thr$^{6.30}$, N atom of residue N117$^{3.37}$ and carbon atom (C5) in the acetyl group of ACh or oxygen atom (O09) in the ether bond of Ipx, NE1 atom of residue W164$^{4.67}$ and carbon atom (C5) in the acetyl group of ACh or oxygen atom (O09) in the ether bond of Ipx, indole ring of residue W413$^{6.48}$ and acetyl group of ACh or heterocyclic isoazoline group of Ipx, OH atom of residue Y89$^{2.61}$ and oxygen atom in the amide group of VU154/LY298, benzene ring of residue F186$^{45.51}$ and aromatic core of the PAMs VU154/LY298, OH

atom of residue Y439[7.39] and nitrogen atoms in the amine group of the PAMs VU154/LY298, CD atom of residue Q184[45.49] and nitrogen atom in the amide group of VU154/LY298, CG atom of residue N423[6.58] and chlorine atom in PAM LY298, OH atom of residue Y92[2.64] and nitrogen atom in the amide group of VU154, OG1 atom of residue T433[7.33] and sulfur atom in the trifluoromethylsulfonyl group of VU154. In addition, the $\chi_2$ angle of residue W413[6.48] and W435[7.35] were calculated. Time courses of these reaction coordinates obtained from the GaMD simulation were plotted in the respective figures. The PyReweighting (*Miao et al., 2014*) toolkit was applied to reweight GaMD simulations to recover the original free energy or PMF profiles of the simulation systems. PMF profiles were computed using the combined trajectories from all the three independent 500 ns GaMD simulations for each system. A bin size of 1.0 Å was used for RMSD. The cutoff was set to 500 frames for 2D PMF calculations. The 2D PMF profiles were obtained for wildtype $M_4R$-Ipx-$G_{i1}$-LY298, $M_4R$-Ipx-$G_{i1}$-VU154, and the D432E and T433R single mutants of the $M_4R$-Ipx-$G_{i1}$-VU154 system regarding the RMSDs of the agonist Ipx and the RMSDs of the PAMs relative to the cryo-EM conformation.

## Data analysis

All pharmacological data was fit using GraphPad Prism 9.2.0. Saturation binding experiments to determine $B_{max}$ and $pK_d$ values were determined as previously described (*Leach et al., 2011*; *Nawaratne et al., 2010*; *Thal et al., 2016*). Detailed equations and analysis details can be found in Appendix 1. Interaction inhibition binding curves between [³H]-NMS, agonists (ACh or Ipx), and PAMs (LY298 or VU154) were analyzed using the allosteric ternary complex model to calculate binding affinity values for each ligand ($pK_A$ – for ACh/Ipx and $pK_B$ for LY298/VU154) and the degree of binding modulation between agonist and PAM (log α) (*Christopoulos and Kenakin, 2002*). The $pK_B$ values for LY298 and VU154 were determined from global fits of the ACh and Ipx curves to generate one $pK_B$ value per ligand (*Ehlert, 1988*; *Leach et al., 2011*; *Nawaratne et al., 2010*; *Thal et al., 2016*). All pERK1/2 and TruPath assays were analyzed using the operational model allosterism and agonism to determine values of orthosteric ($\tau_A$) or allosteric efficacy ($\tau_B$) and the functional modulation (log αβ) between the agonists and PAMs (*Leach et al., 2011*; *Nawaratne et al., 2010*). Binding affinities of the agonists and the PAMs were fixed to values determined from equilibrium binding assays. The $\tau_B$ values for LY298 and VU154 were determined from global fits of the ACh and Ipx curves (when possible) to generate one value per ligand. For comparison between WT human $M_4$ mAChR and other $M_4$ mAChR constructs, the log $\tau$ values were corrected (denoted log $\tau_C$) by normalizing to $B_{max}$ values from saturation binding experiments (*Leach et al., 2011*; *Nawaratne et al., 2010*; *Thal et al., 2016*). All affinity, potency, and cooperativity values were estimated as logarithms, and statistical analysis between WT and mutant $M_4$ mAChR was determined by one-way ANOVA using a Dunnett's post-hoc test with a value of $p < 0.05$ considered as significant in this study.

## Acknowledgements

This work was supported by a Wellcome Trust Collaborative Award (201529/Z/16/Z; PMS, ABT, AC), the National Health and Medical Research Council of Australia (1055134, 1150083, and 1138448), the Australian Research Council (DE170100152, DP190102950, and IC200100052), and the National Institutes of Health (GM132572). PMS is a Senior Principal Research Fellow (1154434), DW a Senior Research Fellow (1155302), DMT an Early Career Research Fellow (1196951), and KL a Future Fellow (160100075). RD was supported by Takeda Science Foundation 2019 Medical Research Grant and Japan Science and Technology Agency PRESTO (18069571). This work was partially supported by the Monash University Ramaciotti Centre for cryo-electron microscopy and the Monash University MASSIVE high-performance computing facility and supercomputing resources with the XSEDE allocation award TG-MCB180049, BIO220137 from the Advanced Cyberinfrastructure Coordination Ecosystem: Services & Support (ACCESS) program, and NERSC project M2874. We thank John Tesmer for discussion and the suggestion of calculating DAQ scores.

## Additional information

### Competing interests

Denise Wootten, Patrick Sexton, Arthur Christopoulos: P.M.S, D.W., and A.C. are shareholders of Septerna Inc. The other authors declare that no competing interests exist.

## Funding

| Funder | Grant reference number | Author |
| --- | --- | --- |
| Wellcome Trust | 201529/Z/16/Z | Andrew Tobin<br>Patrick Sexton<br>Arthur Christopoulos |
| National Health and Medical Research Council | 1055134 | Arthur Christopoulos |
| National Health and Medical Research Council | 1150083 | Arthur Christopoulos |
| National Health and Medical Research Council | 1138448 | David M Thal |
| Australian Research Council | DE170100152 | David M Thal |
| Australian Research Council | DP190102950 | Celine Valant |
| Australian Research Council | IC200100052 | Denise Wootten |
| National Institutes of Health | GM132572 | Yinglong Miao |
| National Health and Medical Research Council | 1154434 | Patrick Sexton |
| National Health and Medical Research Council | 1155302 | Denise Wootten |
| National Health and Medical Research Council | 1196951 | David M Thal |
| Australian Research Council | 160100075 | Katie Leach |
| Takeda Science Foundation | 2019 Medical Research Grant | Radostin Danev |
| Japan Science and Technology Agency | 18069571 | Radostin Danev |

The funders had no role in study design, data collection and interpretation, or the decision to submit the work for publication. For the purpose of Open Access, the authors have applied a CC BY public copyright license to any Author Accepted Manuscript version arising from this submission.

## Author contributions

Ziva Vuckovic, Apurba Bhattarai, Investigation, Methodology, Writing – review and editing; Jinan Wang, Software, Formal analysis, Validation, Investigation, Visualization, Methodology, Writing – review and editing; Vi Pham, Wessel AC Burger, Formal analysis, Investigation, Writing – review and editing; Jesse I Mobbs, Validation, Investigation, Visualization, Writing – review and editing; Matthew J Belousoff, Resources, Formal analysis; Geoff Thompson, Mahmuda Yeasmin, Vindhya Nawaratne, Yi-Lynn Liang, Investigation; Katie Leach, Writing – review and editing; Emma T van der Westhuizen, Elham Khajehali, Alisa Glukhova, Investigation, Writing – review and editing; Denise Wootten, Resources, Supervision, Project administration, Writing – review and editing; Craig W Lindsley, Resources, Writing – review and editing; Andrew Tobin, Conceptualization, Resources, Funding acquisition, Project administration, Writing – review and editing; Patrick Sexton, Conceptualization, Resources, Supervision, Funding acquisition, Project administration, Writing – review and editing; Radostin Danev, Resources, Funding acquisition, Investigation, Writing – review and editing; Celine Valant, Conceptualization, Formal analysis, Supervision, Validation, Visualization, Writing – original draft, Project administration, Writing – review and editing; Yinglong Miao, Conceptualization, Resources, Software, Formal analysis, Supervision, Funding acquisition, Validation, Visualization,

Methodology, Writing – original draft, Project administration, Writing – review and editing; Arthur Christopoulos, Conceptualization, Resources, Supervision, Funding acquisition, Writing – original draft, Project administration, Writing – review and editing; David M Thal, Conceptualization, Resources, Software, Formal analysis, Supervision, Funding acquisition, Validation, Investigation, Visualization, Methodology, Writing – original draft, Project administration, Writing – review and editing

### Author ORCIDs
Emma T van der Westhuizen 
Andrew Tobin 
Arthur Christopoulos 
David M Thal 

### Decision letter and Author response
Decision letter https://doi.org/10.7554/eLife.83477.sa1
Author response https://doi.org/10.7554/eLife.83477.sa2

## Additional files

### Supplementary files
• MDAR checklist

### Data availability
All data generated or analysed during this study are included in the manuscript. Structural data has been deposited in the Protein Data Bank (PDB) and Electron Microscopy Data Bank (EMDB) under the following codes:(1) M4R-Gi1-Ipx PDB: 7TRK and EMD-26099 (2) M4R-Gi1-Ipx-LY298 PDB: 7TRP and EMD-26100 (3) M4R-Gi1-Ipx-VU154 PDB: 7TRQ and EMD-26101 (4) M4R-Gi1-ACh PDB: 7TRS and EMD-26102.

The following datasets were generated:

| Author(s) | Year | Dataset title | Dataset URL | Database and Identifier |
|---|---|---|---|---|
| Vuckovic Z, Mobbs JI, Belousoff MJ, Glukhova A, Sexton PM, Danev R, Thal DM | 2023 | Human M4 muscarinic acetylcholine receptor complex with Gi1 and the agonist iperoxo | https://www.rcsb.org/structure/7TRK | RCSB Protein Data Bank, 7TRK |
| Vuckovic Z, Mobbs JI, Belousoff MJ, Glukhova A, Sexton PM, Danev R, Thal DM | 2023 | Human M4 muscarinic acetylcholine receptor complex with Gi1 and the agonist iperoxo and positive allosteric modulator LY2033298 | https://www.rcsb.org/structure/7TRP | RCSB Protein Data Bank, 7TRP |
| Vuckovic Z, Mobbs JI, Belousoff MJ, Glukhova A, Sexton PM, Danev R, Thal DM | 2023 | Human M4 muscarinic acetylcholine receptor complex with Gi1 and the agonist iperoxo and positive allosteric modulator VU0467154 | https://www.rcsb.org/structure/7TRQ | RCSB Protein Data Bank, 7TRQ |
| Vuckovic Z, Mobbs JI, Belousoff MJ, Glukhova A, Sexton PM, Danev R, Thal DM | 2023 | Human M4 muscarinic acetylcholine receptor complex with Gi1 and the endogenous agonist acetylcholine | https://www.rcsb.org/structure/7TRS | RCSB Protein Data Bank, 7TRS |

*Continued on next page*

*Continued*

| Author(s) | Year | Dataset title | Dataset URL | Database and Identifier |
|---|---|---|---|---|
| Vuckovic Z, Mobbs JI, Belousoff MJ, Glukhova A, Sexton PM, Danev R, Thal DM | 2023 | Human M4 muscarinic acetylcholine receptor complex with Gi1 and the agonist iperoxo | https://www.ebi.ac.uk/emdb/search/EMD-26099 | Electron Microscopy Data Bank, EMD-26099 |
| Vuckovic Z, Mobbs JI, Belousoff MJ, Glukhova A, Sexton PM, Danev R, Thal DM | 2023 | Human M4 muscarinic acetylcholine receptor complex with Gi1 and the agonist iperoxo and positive allosteric modulator LY2033298 | https://www.ebi.ac.uk/emdb/search/EMD-26100 | Electron Microscopy Data Bank, EMD-26100 |
| Vuckovic Z, Mobbs JI, Belousoff MJ, Glukhova A, Sexton PM, Danev R, Thanl DM | 2023 | Human M4 muscarinic acetylcholine receptor complex with Gi1 and the agonist iperoxo and positive allosteric modulator VU0467154 | https://www.ebi.ac.uk/emdb/search/EMD-26101 | Electron Microscopy Data Bank, EMD-26101 |
| Vuckovic Z, Mobbs JI, Belousoff MJ, Glukhova A, Sexton PM, Danev R, Thal DM | 2023 | Human M4 muscarinic acetylcholine receptor complex with Gi1 and the endogenous agonist acetylcholine | https://www.ebi.ac.uk/emdb/search/EMD-26102 | Electron Microscopy Data Bank, EMD-26102 |

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

## Appendix 1

### Data analysis

### Pharmacological parameters related to ligand binding

Interaction radioligand binding data were analyzed according to the following adapted form of an allosteric ternary complex model that accounts for the interaction of two orthosteric ligands and one allosteric ligand on a receptor (*Christopoulos and Kenakin, 2002*; *Leach et al., 2007*; *Leach et al., 2011*):

$$Y = \frac{B_{max} \cdot [A]}{[A] + [(K_A \cdot K_B)/(\alpha_A[B] + K_B)] \cdot \{1 + ([I]/K_I) + ([B]/K_B) + ([\alpha_I[I][B]/K_I K_B])\}}$$

- [A], [B], and [I] represent the concentrations of the radioligand ([$^3$H]-NMS), allosteric ligand, and orthosteric inhibitor, respectively.
- $K_A$, $K_B$, and $K_I$ represent their respective equilibrium dissociation constants. The value $K_A$ was fixed to the value determined from saturation binding experiments.
- $B_{max}$ is the total number of receptors.
- $\alpha_A$ and $\alpha_I$ represent the affinity cooperativity values between the allosteric ligand and the radioligand or orthosteric inhibitor, respectively. Values greater than 1 indicate positive cooperativity, values <1 (but >0) indicate negative cooperativity, and values of unity indicate neutral cooperativity.
- All potency, affinity, and cooperativity parameters were estimated as logarithms.

Agonist binding affinity with a PAM bound:

$$Y = pK_I + \log \alpha_I$$

$$\Delta Y = ((\Delta pK_I)^2 + (\Delta \log \alpha_I)^2)^{1/2}$$

PAM binding affinity for the agonist-bound state:

$$Y = pK_B + \log \alpha_I$$

$$\Delta Y = ((\Delta pK_B)^2 + (\Delta \log \alpha_I)^2)^{1/2}$$

Global analysis for sharing the $pK_B$ value between interaction experiments using two different agonists:

Define:

Ag1=agonist 1 (e.g. ACh)

Ag2=agonist 2 (e.g. Ipx)

alpha = $\alpha_A$

betta = $\alpha_I$

Data for agonist 1 in columns <A:F>

Data for agonist 2 in columns <G:L>

Prism equation:

$$KA = 10^{\wedge}LogKA$$

$$KB = 10^{\wedge}LogKB$$

$$I = 10^{\wedge}X$$

$$A = 10^{\wedge}LogHotA$$

$$KIAg1 = 10^{\wedge}LogKIAg1 \text{ (Ki for agonist 1)}$$

$$KIAg2 = 10^{\wedge}LogKIAg2 \text{ (Ki for agonist 2)}$$

$$alphaAg1 = 10^{\wedge}LogalphaAg1$$

$$bettaAg1 = 10^{\wedge}LogbettaAg1$$

$$alphaAg2 = 10^{\wedge}LogalphaAg2$$

$$bettaAg2 = 10^{\wedge}LogbettaAg2$$

$$Part1Ag1 = (KA^{*}KB)/(alphaAg1^{*}B + KB)$$

$$Part2Ag1 = 1 + I/KIAg1 + B/KB + (bettaAg1^{*}I^{*}B)/(KIAg1^{*}KB)$$

$$KAppAg1 = Part1Ag1^{*}Part2Ag1$$

$$Part1Ag2 = (KA^{*}KB)/(alphaAg2^{*}B + KB)$$

$$Part2Ag2 = 1 + I/KIAg2 + B/KB + (bettaAg2^{*}I^{*}B)/(KIAg2^{*}KB)$$

$$KAppAg2 = Part1Ag2^{*}Part2Ag2$$

$$< A : F > Y = (BmaxAg1^{*}A)/(A + KAppAg1)$$

$$< G : L > Y = (BmaxAg1^{*}A)/(A + KAppAg2)$$

## Pharmacological parameters related to function

To determine efficacy values ($\tau_A$) of ACh and Ipx in TruPath and pERK1/2 assays, data were directly fit to the operational model of agonism (*Black and Leff, 1983*):

$$Y = Basal + \frac{E_m - Basal}{1 + \frac{K_A + [A]}{\tau_A \times [A]}}$$

where $E_m$ is the maximal response of the system, Basal is the basal level of response in the absence of agonist, [A] is the concentration of agonist, $K_A$ denotes the equilibrium dissociation constant of the agonist (A), and $\tau_A$ is the index of the coupling efficiency/efficacy of the agonist. Values of $K_A$ were constrained to the corresponding $K_i$ values from interaction radioligand binding experiments.

The determination of $\tau_B$ and $\alpha\beta$ values for LY298 and VU154 in TruPath and pERK1/2 assays data was directly fit to the following operational model of allosterism and agonism (*Leach et al., 2007*):

$$Y = Basal + \frac{(E_m - Basal)\left([A]K_B + \alpha\beta[A][B] + \tau_B[A][B]EC_{50}\right)^n}{\left([A]K_B + \alpha\beta[A][B] + \tau_B[A][B]EC_{50}\right)^n + EC_{50}^n\left(K_B + [B]\right)^n}$$

where $E_m$ is the maximal response of the system, Basal is the basal level of response in the absence of agonist, [A] and [B] are the concentrations of agonist and the PAM, respectively. $K_B$ denotes the equilibrium dissociation constant of the PAM. $\tau_B$ is the index of the coupling efficiency/efficacy of the PAM. $\alpha$ denotes the binding cooperativity between the agonist and PAM, whereas $\beta$ denotes a scaling factor that quantifies the allosteric effect of the PAM on the orthosteric ligand efficacy. This model assumes that the agonists (A) are full agonists. n is the transducer slope that describes the stimulus–response coupling of the ligand–receptor to the signaling pathway and was constrained to 1. The allosteric binding affinity ($K_B$) was constrained to the value determined from radioligand binding experiments. All potency, affinity, efficacy, and cooperativity parameters were estimated as logarithms.

Transduction coupling coefficients were calculated as follows *Kenakin et al., 2012*:

$$Y = log\frac{\tau_A}{K_A} = log\tau_A - logK_A$$

$$\Delta Y = ((\Delta \log \tau_A)^2 + (\Delta K_A)^2)^{1/2}$$

Efficacy modulation ($\beta$) was calculated as follows:

$$Y = \log \beta = \log \alpha\beta - \log \alpha$$

$$\Delta Y = ((\Delta \log \alpha\beta)^2 + (\Delta \log \alpha)^2)^{1/2}$$

Global analysis for sharing $\tau_B$ between interaction functional experiments using two different agonists:

Define:
Ag1=agonist 1 (e.g. ACh)
Ag2=agonist 2 (e.g. Ipx)

$$ab = \alpha\beta$$

Data for agonist 1 in columns <A:F>
Data for agonist 2 in columns <G:L>
Prism equation:

$$KB = 10^{\wedge}LogKB$$

$$tauB = 10^{\wedge}LogtauB$$

$$A = 10^{\wedge}X$$

$$B = ConcMod$$

$$EC50Ag1 = 10^{\wedge}LogEC50Ag1$$

$$EC50Ag2 = 10^{\wedge}LogEC50Ag2$$

$$abAg1 = 10^{\wedge}LogAlphaBetaAg1$$

$$abAg2 = 10^{\wedge}LogAlphaBetaAg2$$

$$Part1Ag1 = (A^*(KB + abAg1^*B) + tauB^*B^*EC50Ag1)^{\wedge}n$$

$$Part1Ag2 = (A^*(KB + abAg2^*B) + tauB^*B^*EC50Ag2)^{\wedge}n$$

$$Part2Ag1 = (EC50Ag1^{\wedge}n)^*((KB + B)^{\wedge}n)$$

$$Part2Ag2 = (EC50Ag2^{\wedge}n)^*((KB + B)^{\wedge}n)$$

$$SpanAg1 = EmAg1BasalAg1$$

$$SpanAg2 = EmAg2BasalAg2$$

$$<A : F> Y = BasalAg1 + (SpanAg1^*Part1Ag1)/(Part1Ag1 + Part2Ag1)$$

$$<G : L> Y = BasalAg2 + (SpanAg2^*Part1Ag2)/(Part1Ag2 + Part2Ag2)$$

Correcting efficacy $\tau_X$ values using receptor expression (*Leach et al., 2011*; *Nawaratne et al., 2010*):
$B_{max-WT}$ = maximal number of receptors for the WT receptor
$B_{max-X}$ = maximal number of receptors for the receptor construct X
$\tau_{WT}$ = efficacy at the WT receptor

$\tau_X$ = efficacy at receptor X

$$\log\tau_{X-corr} = \log\tau_X - (\log B_{max-X} - \log B_{max-WT})$$

$$\Delta\log\tau_{X-corr} = \left( \left(\Delta\log\tau_X\right)^2 + \left(0.432 \times \left(\frac{\Delta B_{max-X}}{B_{max-X}}\right)\right)^2 + \left(0.432 \times \left(\frac{\Delta B_{max-WT}}{B_{max-WT}}\right)\right)^2 \right)^{1/2}$$

## Appendix 1—key resources table

| Reagent type (species) or resource | Designation | Source or reference | Identifiers | Additional information |
|---|---|---|---|---|
| Antibody | Anti-FLAG M1 (mouse polyclonal IgG2a) | Gift from Prof. Brian Kobilka (PMID::17962520) | | Antibody was used to make anti-FLAG mAb resin that was used for the purification of FLAG-tagged M4 mAChR |
| Strain, strain background (*Escherichia coli*) | DH5α | New England Biolabs | C2987H | |
| Strain, strain background (*E. coli*) | DH10bac | Thermo Fisher Scientific | 10361012 | |
| Cell line (*Spodoptera frugiperda*) | Sf9 | Expression Systems | 94-001S | |
| Cell line (*Trichoplusia ni*) | Tni | Expression Systems | 94-002S | |
| Cell line (Chinese hamster ovary) | Flp-In CHO human M4 mAChR WT | PMID:26958838 | | |
| Cell line (Chinese hamster ovary) | CHO K1 mouse M4 mAChR WT | PMID:21198541 | | |
| Cell line (Chinese hamster ovary) | Flp-In CHO human M4 mAChR D432E | This study | | pEF5-FRT-V5-DEST plasmid |
| Cell line (Chinese hamster ovary) | Flp-In CHO human M4 mAChR T433R | This study | | pEF5-FRT-V5-DEST plasmid |
| Cell line (Chinese hamster ovary) | Flp-In CHO human M4 mAChR V91L, D432E, T433R | This study | | pEF5-FRT-V5-DEST plasmid |
| Cell line (Chinese hamster ovary) | Flp-In CHO human M4 mAChR Y89A | PMID:26958838 | | |
| Cell line (Chinese hamster ovary) | Flp-In CHO human M4 mAChR Q184A | PMID:26958838 | | |
| Cell line (Chinese hamster ovary) | Flp-In CHO human M4 mAChR F186A | PMID:20406819 | | |
| Cell line (Chinese hamster ovary) | Flp-In CHO human M4 mAChR W435A | PMID:26958838 | | |
| Cell line (Chinese hamster ovary) | Flp-In CHO human M4 mAChR W439A | PMID:20406819 | | |
| Cell line (Chinese hamster ovary) | Flp-In CHO human M4 mAChR S85A | PMID:26958838 | | |
| Cell line (Chinese hamster ovary) | Flp-In CHO human M4 mAChR Y89A | PMID:26958838 | | |

*Appendix 1 Continued on next page*

*Appendix 1 Continued*

| Reagent type (species) or resource | Designation | Source or reference | Identifiers | Additional information |
|---|---|---|---|---|
| Cell line (Chinese hamster ovary) | Flp-In CHO human M4 mAChR Y92A | PMID:26958838 | | |
| Cell line (Chinese hamster ovary) | Flp-In CHO human M4 mAChR I93T, I94V, K95I | PMID:20406819 | | |
| Cell line (Chinese hamster ovary) | Flp-In CHO human M4 mAChR I93T | PMID:20406819 | | |
| Cell line (Chinese hamster ovary) | Flp-In CHO human M4 mAChR I94V | PMID:20406819 | | |
| Cell line (Chinese hamster ovary) | Flp-In CHO human M4 mAChR K95I | PMID:20406819 | | |
| Cell line (Chinese hamster ovary) | Flp-In CHO human M4 mAChR Y97A | PMID:26958838 | | |
| Cell line (Chinese hamster ovary) | Flp-In CHO human M4 mAChR W98A | PMID:26958838 | | |
| Cell line (Chinese hamster ovary) | Flp-In CHO human M4 mAChR G101A | PMID:26958838 | | |
| Cell line (Chinese hamster ovary) | Flp-In CHO human M4 mAChR D106A | PMID:21300722 | | |
| Cell line (Chinese hamster ovary) | Flp-In CHO human M4 mAChR W108A | PMID:21300722 | | |
| Cell line (Chinese hamster ovary) | Flp-In CHO human M4 mAChR L109A | PMID:21300722 | | |
| Cell line (Chinese hamster ovary) | Flp-In CHO human M4 mAChR D112E | PMID:21300722 | | |
| Cell line (Chinese hamster ovary) | Flp-In CHO human M4 mAChR D112N | PMID:21300722 | | |
| Cell line (Chinese hamster ovary) | Flp-In CHO human M4 mAChR S116A | PMID:21300722 | | |
| Cell line (Chinese hamster ovary) | Flp-In CHO human M4 mAChR N117A | PMID:21300722 | | |
| Cell line (Chinese hamster ovary) | Flp-In CHO human M4 mAChR V120A | PMID:21300722 | | |
| Cell line (Chinese hamster ovary) | Flp-In CHO human M4 mAChR D129E | PMID:21300722 | | |
| Cell line (Chinese hamster ovary) | Flp-In CHO human M4 mAChR D129N | PMID:21300722 | | |
| Cell line (Chinese hamster ovary) | Flp-In CHO human M4 mAChR W164A | PMID:26958838 | | |
| Cell line (Chinese hamster ovary) | Flp-In CHO human M4 mAChR F170A | PMID:26958838 | | |
| Cell line (Chinese hamster ovary) | Flp-In CHO human M4 mAChR W171A | PMID:26958838 | | |
| Cell line (Chinese hamster ovary) | Flp-In CHO human M4 mAChR Q172A | PMID:26958838 | | |
| Cell line (Chinese hamster ovary) | Flp-In CHO human M4 mAChR F173A | PMID:26958838 | | |
| Cell line (Chinese hamster ovary) | Flp-In CHO human M4 mAChR Q184A | PMID:26958838 | | |

*Appendix 1 Continued on next page*

*Appendix 1 Continued*

| Reagent type (species) or resource | Designation | Source or reference | Identifiers | Additional information |
|---|---|---|---|---|
| Cell line (Chinese hamster ovary) | Flp-In CHO human M4 mAChR F186A | PMID:20406819 | | |
| Cell line (Chinese hamster ovary) | Flp-In CHO human M4 mAChR I187A | PMID:26958838 | | |
| Cell line (Chinese hamster ovary) | Flp-In CHO human M4 mAChR Q188A | PMID:26958838 | | |
| Cell line (Chinese hamster ovary) | Flp-In CHO human M4 mAChR F189A | PMID:26958838 | | |
| Cell line (Chinese hamster ovary) | Flp-In CHO human M4 mAChR L190A | PMID:26958838 | | |
| Cell line (Chinese hamster ovary) | Flp-In CHO human M4 mAChR W413A | PMID:26958838 | | |
| Cell line (Chinese hamster ovary) | Flp-In CHO human M4 mAChR Y416A | PMID:26958838 | | |
| Cell line (Chinese hamster ovary) | Flp-In CHO human M4 mAChR N423A | PMID:26958838 | | |
| Cell line (Chinese hamster ovary) | Flp-In CHO human M4 mAChR Q427A | PMID:26958838 | | |
| Cell line (Chinese hamster ovary) | Flp-In CHO human M4 mAChR S428P | PMID:20406819 | | |
| Cell line (Chinese hamster ovary) | Flp-In CHO human M4 mAChR D432N | PMID:20406819 | | |
| Cell line (Chinese hamster ovary) | Flp-In CHO human M4 mAChR W435A | PMID:26958838 | | |
| Cell line (Chinese hamster ovary) | Flp-In CHO human M4 mAChR Y439A | PMID:20406819 | | |
| Cell line (Chinese hamster ovary) | Flp-In CHO human M4 mAChR Y440A | PMID:26958838 | | |
| Cell line (Chinese hamster ovary) | Flp-In CHO human M4 mAChR C442A | PMID:20406819 | | |
| Cell line (Chinese hamster ovary) | Flp-In CHO human M4 mAChR Y443A | PMID:20406819 | | |
| Cell line (Chinese hamster ovary) | Flp-In CHO cell line | Thermo Fisher Scientific | R75807 | |
| Cell line (*Homo sapiens*) | 293A cell line | Thermo Fisher Scientific | R70507 | |
| Recombinant DNA reagent | Human FLAG-M4Δi3-His | This study | | pVL1392 vector |
| Recombinant DNA reagent | Human Gαi1 dominant negative mutant | PMID:32193322 | | pFastBac vector |
| Recombinant DNA reagent | Human Gβ1γ2 | Gift from Prof. Brian Kobilka (PMID:24256733) | | pVL1392 vector |
| Recombinant DNA reagent | scFv16 | PMID:30213947 | | pFastBac vector |

*Appendix 1 Continued on next page*

*Appendix 1 Continued*

| Reagent type (species) or resource | Designation | Source or reference | Identifiers | Additional information |
|---|---|---|---|---|
| Recombinant DNA reagent | pcDNA5/FRT/TO-GAlphai1-RLuc8 | Gift from Prof. Bryan Roth (PMID:32367019) | | TRUPATH assay |
| Recombinant DNA reagent | pcDNA3.1-Beta3 | Gift from Prof. Bryan Roth (PMID:32367019) | | TRUPATH assay |
| Recombinant DNA reagent | pcDNA3.1-GGamma9-GFP2 | Gift from Prof. Bryan Roth (PMID:32367019) | | TRUPATH assay |
| Chemical compound, drug | Acetylcholine | Sigma-Aldrich | | |
| Chemical compound, drug | Iperoxo | Sigma-Aldrich | | |
| Chemical compound, drug | LY2033298 | Sigma-Aldrich | | |
| Chemical compound, drug | VU0467154 | Gift from Prof. Craig Lindsley (PMID:25137629) | | PMID:25137629 |
| Chemical compound, drug | Prolume Purple | Nanolight Technology | | |
| Chemical compound, drug | [$^3$H]-NMS | PerkinElmer | | |
| Chemical compound, drug | Polyethylenimine (PEI) | Sigma-Aldrich | | |
| Chemical compound, drug | Atropine | Sigma-Aldrich | | |
| Commercial assay or kit | AlphaScreen SureFire pERK 1/2 (Thr202/Tyr204) Assay Kits | PerkinElmer | | |
| Software, algorithm | Prism 8.0 | GraphPad | | |
| Software, algorithm | PyMOL | Schrödinger | | |
| Software, algorithm | GaMD | PMID:26300708 | | |
| Software, algorithm | AMBER20 | https://ambermd.org | | |
| Software, algorithm | CPPTRAJ | PMID:26300708 | | |
| Software, algorithm | PyReweighting | PMID:25061441 | | |
| Software, algorithm | Phenix suite | PMID:20124702 | | |
| Software, algorithm | Coot | PMID:31730249 | | |
| Software, algorithm | Chimera | PMID:15264254 | | |

*Appendix 1 Continued on next page*

*Appendix 1 Continued*

| Reagent type (species) or resource | Designation | Source or reference | Identifiers | Additional information |
|---|---|---|---|---|
| Software, algorithm | Chimera X | PMID:32881101 | | |
| Software, algorithm | cryoSPARC | PMID:28165473 | | |
| Software, algorithm | Relion 3.1 | PMID:30412051 | | |
| Software, algorithm | Motioncor2 | PMID:28250466 | | |
| Software, algorithm | GCTF | PMID:26592709 | | |
| Software, algorithm | crYOLO | PMID:31240256 | | |
| Software, algorithm | ISOLDE | PMID:29872003 | | |
| Software, algorithm | MolProbity | PMID:29067766 | | |
| Software, algorithm | DAQ score | PMID:35953671 | | |

