## [Editor Report]

This important work advances our understanding of the structural basis of allosteric modulation of the M4 muscarinic receptor but has broad implications for GPCRs. The evidence supporting the conclusions is exceptional, with multiple cryo-EM structures that are complemented by excellent pharmacological and dynamics studies.

---

## [Decision Letter]

**Decision letter after peer review:**

Thank you for submitting your article "Pharmacological hallmarks of allostery at the M4 muscarinic receptor elucidated through structure and dynamics" for consideration by *eLife*. Your article has been reviewed by 3 peer reviewers, including Sudarshan Rajagopal as the Reviewing Editor and Reviewer #1, and the evaluation has been overseen by Richard Aldrich as the Senior Editor.

Essential revisions:

1) Please address the comment from reviewer 1 regarding log tau vs log tau/KD for assessing the efficacy of ligands.

2) Please address the comments from reviewers 2-3, especially with the presentation of the data/figures for improved readability.

*Reviewer #1 (Recommendations for the authors):*

I have a concern about the use of a transducer coupling coefficient (log(tau/K)) in the manuscript. As the authors note, this is typically used to quantify agonist bias, usually by comparing it to a coupling coefficient for a different signaling pathway by the same agonist. But coupling coefficients really can't be used to compare different agonists because of different dissociation constants for each agonist – a reference is made to Kenakin, 2012 (which I went back to and didn't find this argument) that the coupling coefficient characterizes agonism of a specific pathway defined as the interaction between an agonist, receptor, and transducer. This leads to some questionable interpretations in lines 177-181. I believe the proper interpretation is that ACh is more efficacious as it has a larger log tau (which has been shown to be related to proportional efficacy by Onaran et al. – Sci Rep. 2017 Mar 14;7:44247. doi: 10.1038/srep44247.) – the transducer coupling coefficient doesn't matter as it is being driven by the tighter binding. Using the ternary complex model, efficacy is proportional to the differences in affinity between the transducer-bound and unbound states, while affinity largely reflects the interaction with the transducer-unbound state (depending on the experimental platform).

The same issue arises in lines 613-619 where an argument is made that "structures of GPCRs in a ternary complex… are better represented by their transducer coupling coefficient than the efficacy of the agonist…." Essentially, this is making the argument that efficacy/affinity is a better representation of the ternary complex than efficacy. Let's take an example of the b2 adrenergic receptor (which I know better than the M4) – isoproterenol has a log KD of -6 but is a full agonist hitting Emax. Pindolol is a very weak partial agonist but binds tightly with log KD -9.3 but with an efficacy of 10% of isoproterenol. If I calculate tau/KD for these two compounds, pindolol would have a much higher value – but it would be completely driven by the high binding affinity. It would not reflect the stability of the ternary complex. The interpretation that the transducer coupling coefficient is in contradiction to basic tenets of pharmacology.

I think the section on "Structural and dynamic insights into orthosteric and allosteric agonism" proposes some plausible ideas but probably overstates the insights that can be obtained from transducer-bound structures. As with other crystal or cryoEM structures, the observed conformation of the receptor is largely driven by the bound transducer and not by the pharmacological characteristics of the agonist (partial vs full, etc. – although I doubt you could get an antagonist bound to a transducer-bound receptor). This limitation has to be highlighted as it is a major one for these structural studies (and is theoretically less of an issue with solution-based studies).

Figure 6. It would be helpful to label some of the microswitches in the figure, as it may not be obvious to all readers, e.g., me, what specific residues they are looking at.

Figure S2. Any comment on the different populations of complexes with the PAMs? It is interesting that VU154 induces a large population of presumably a different conformation.

- Consider focusing on log tau as a readout of efficacy as opposed to the transduction coefficients, which have limited utility when comparing agonists.

- A sentence or two on limitations of the cryoEM approach in studying agonism due to the stabilization of the A-R-T complex.

*Reviewer #2 (Recommendations for the authors):*

1. The abstract focuses on the questions and the methods used to address them, with relatively little in terms of specific details about what is found. Simply saying the work offers "in-depth insights", while true, is a bit vague and leaves the reader guessing. Some more specific statements about key findings (e.g., the superior coupling to G protein induced by ACh) would be appreciated.

2. Figure 2 panels B-E and G-J might be made larger, and the data points for the other panels smaller, for clarity. The data points obscure error bars in all but a few cases.

3. Figure 3 shows a huge amount of structural data, but this means all panels in the figure are very small. The authors may want to consider a more focused presentation, with fewer essential components left for the SI. Panels A-C are somewhat redundant, for example, and a single representative example might suffice.

4. The discussion of Wang et al. 2022 and the corresponding figure S5 are very important. Figure S5 was difficult to interpret and see clearly, perhaps in part due to image compression by the journal. Even so, a more focused presentation would still be helpful, ideally with a finer map mesh. A transparent surface view may be even easier to interpret.

5. The discussion is clearly written and helpful, but reads as being targeted to a specialist pharmacology audience. Including more of a broad perspective and relating this to allostery in other GPCRs and other protein families could enhance the overall breadth of appeal.

6. The final sentence about signaling bias (line 624) is important, and the authors may want to state explicitly that the structural and mechanistic basis for biased signaling (and allostery as well) is likely to vary from one receptor to another. This is implied, but if the authors feel justified in making a stronger statement it may be helpful for clarity and emphasis.

*Reviewer #3 (Recommendations for the authors):*

1. I would propose merging Figures 1 and 2 to clarify which aspect is covered for each experiment in Figure 2. It would be great to use a color code (or any other sort of label) for each parameter (K, α, etc.) and highlight the respective plots (including supplements), for readers to follow the discussion.

2. It would also be helpful to show the structures of all four compounds in the main text, for clarity. These could be used to highlight which parts of the ligands were observed in the structure, vs. which ones were disordered (i.e. which bonds are rotating freely), as discussed for both Ipx and VU154.

3. All pharmacological experiments have been performed with a full-length wild type, however, the cryoEM structure contains a major deletion of ICL3. While I understand that previous experiments have used similar constructs for structure determination, I believe it would be helpful to confirm binding affinities, as well as the efficacy of the respective drugs with the truncated cryoEM construct. If this is not feasible, please highlight the caveat that all structures, and consequently all MD simulations, are based on data obtained from an engineered construct, rather than the wildtype sequence.

4. Line 190: What is a "NAL" (i presume it should spell 'NAM')? Please also discuss the impact of this observation.

5. The pharmacological experiments were conducted with all possible combinations of ligands (ACh, Ipx with LY298, VU154). I understand that it could have been merely a factor of resources, but were any attempts made to elucidate structures of the PAM-complexes with ACh?

6. It is not clear to the reader, whether the above pharmacological experiments (Figure 2) have been carried out for the very first time, or if others have attempted similar studies. Please clarify exactly what part of the work is novel.

7. Figure 2A: the fact that increasing concentrations of LY298 appear to block the overall binding of ACh is not described or discussed anywhere. Based on this plot LY298 would be a PAM-antagonist (see for example Figure 1, Grundmann et al. 2021, 10.3390/ijms22041763). This would be an important aspect, which would need to be addressed.

8. I would suggest placing some of the MD simulation traces into the supplementary materials, as these currently take up a large fraction of all figures. Alternatively, different complexes could be color coded and overlaid in one figure to highlight differences.

9. Figure 4 B and H: given the structure in Figure 4 H, it means that the binding pocket around Ipx leaves no room for any movement. Overall, I am not able to follow the discussion regarding the alkyne group/linker of Ipx not being visible. If the start and end points are fixed (visible densities), and have a rigid, planar triple bond involved (alkyne), I find it hard to imagine that the linker is flexible enough to wash out the signal for the linker. Also, the representative Figure S3F is not very convincing, as (A) all iperoxo densities seem to be of rather poor quality and (B) the average of all aligned structures would still likely result in an 'average-able' linker density. I would suggest either elaborating on or omitting this claim.

10. Line 301, The predominate χ2 angle of W413 was approximately 60◦ and 105◦ in the ACh-bound and Ipx-bound simulations, respectively, corresponding to the cryo-EM conformations. As depicted in Figure 4L, W413 when bound to ACh also samples angles close to 90 degrees or higher in the majority of Sim2 and part of Sim1. What is the significance of this conformational sampling?

11. On the note of ligands having flexible parts in VU154, and therefore no resolved densities in the maps, is there any pharmacological evidence (i.e. SAR) for these regions not contributing to binding/signaling? Analogously, are there any SAR data for replacing the alkyne bond in Ipx? It would be conceivable that a rigid replacement linker would affect (either positively or negatively) receptor binding and signaling.

12. According to the GaMD simulation results in Figure S7C, the minimum distance for the T433 and VU154 seems to be close to 4 Å while the predominant distance is around 7 Å. Therefore, I am wondering what the significance of this hydrogen bond is observed in the cryoEM structure. Additionally, T433R mutant showed increased binding to VU154 and showed the importance of T433 in species selectivity. Is this increased binding of VU154 with T433R mutant a result of a more "stable" hydrogen bond between receptor and VU154?

13. Regarding the species selectivity aspect, there is no mention of the V91L mutant in Figure 7, only as part of the triple mutant. It is hard to judge which mutations are responsible for species selectivity without either showing results for the D432E/T433R double mutant, or the additional V91L single mutant.

---

## [Author Response]

Essential revisions:1) Please address the comment from reviewer 1 regarding log tau vs log tau/KD for assessing the efficacy of ligands.2) Please address the comments from reviewers 2-3, especially with the presentation of the data/figures for improved readability.

We appreciate and thank the editors and reviewers for their time and effort providing positive and constructive feedback on the manuscript. For (1) we now have further clarified how and why we estimate both ligand efficacy and transducer coefficients of ligands in the manuscript, taking into account the reviewer concerns and suggestions (see detailed response below). For (2) we have made substantial revisions to the figures and text in accordance with the suggestions of the reviewers. These changes were done in accordance with the general editorial recommendations of *eLife*.

Reviewer #1 (Recommendations for the authors):I have a concern about the use of a transducer coupling coefficient (log(tau/K)) in the manuscript. As the authors note, this is typically used to quantify agonist bias, usually by comparing it to a coupling coefficient for a different signaling pathway by the same agonist. But coupling coefficients really can't be used to compare different agonists because of different dissociation constants for each agonist – a reference is made to Kenakin, 2012 (which I went back to and didn't find this argument) that the coupling coefficient characterizes agonism of a specific pathway defined as the interaction between an agonist, receptor, and transducer. This leads to some questionable interpretations in lines 177-181. I believe the proper interpretation is that ACh is more efficacious as it has a larger log tau (which has been shown to be related to proportional efficacy by Onaran et al. – Sci Rep. 2017 Mar 14;7:44247. doi: 10.1038/srep44247.) – the transducer coupling coefficient doesn't matter as it is being driven by the tighter binding. Using the ternary complex model, efficacy is proportional to the differences in affinity between the transducer-bound and unbound states, while affinity largely reflects the interaction with the transducer-unbound state (depending on the experimental platform).The same issue arises in lines 613-619 where an argument is made that "structures of GPCRs in a ternary complex… are better represented by their transducer coupling coefficient than the efficacy of the agonist…." Essentially, this is making the argument that efficacy/affinity is a better representation of the ternary complex than efficacy. Let's take an example of the b2 adrenergic receptor (which I know better than the M4) – isoproterenol has a log KD of -6 but is a full agonist hitting Emax. Pindolol is a very weak partial agonist but binds tightly with log KD -9.3 but with an efficacy of 10% of isoproterenol. If I calculate tau/KD for these two compounds, pindolol would have a much higher value – but it would be completely driven by the high binding affinity. It would not reflect the stability of the ternary complex. The interpretation that the transducer coupling coefficient is in contradiction to basic tenets of pharmacology.I think the section on "Structural and dynamic insights into orthosteric and allosteric agonism" proposes some plausible ideas but probably overstates the insights that can be obtained from transducer-bound structures. As with other crystal or cryoEM structures, the observed conformation of the receptor is largely driven by the bound transducer and not by the pharmacological characteristics of the agonist (partial vs full, etc. – although I doubt you could get an antagonist bound to a transducer-bound receptor). This limitation has to be highlighted as it is a major one for these structural studies (and is theoretically less of an issue with solution-based studies).

We appreciate the reviewer’s concern about our use and description of transducer coupling coefficients in the manuscript (as originally written), but respectfully disagree with the comment that “…coupling coefficients really can't be used to compare different agonists because of different dissociation constants for each agonist…”. This is because the latter statement is dependent on the *context* in which the transducer coupling coefficient is used, and we apologise for not being clearer in terms of both its definition and the reason why we included this additional parameter in our analyses of our functional data.

In essence, we feel that there are two main concerns that are raised by the reviewer regarding the transducer coupling coefficient:

1. The purpose of this parameter when it comes to quantifying biased agonism.

2. The relevance of this parameter for interpreting structural biology data of the ternary complex of agonist (+/- PAM)-receptor-transducer.

In terms of the first concern, the transducer coupling coefficient is used as a *starting point* for the ultimate quantification of biased agonism in cellular signaling pathways; it is, in-and-of-itself, not the ‘end point’ for the comparison of agonists. As described in Kenakin and Christopoulos, 2013, *Nat. Rev. Drug Discovery*, 12: 205, this parameter is used specifically because bias may arise from different tau (efficacy) values for a given pathway linked to a single (common) affinity value and/or a different affinity value linked to a given pathway (in the intact cellular environment). The affinity (K) parameter in this operational model-based approach thus represents a ‘conditional’ or ‘apparent/observed’ affinity, even for the ‘ground state’, because it subsumes the potential for the receptor to isomerize between different states (transducer-bound or otherwise) prior to initiating a cellular response. To date (and to our knowledge), in most instances in the literature, the measured affinity of an agonist for a given receptor is essentially identical between pathways (irrespective of rapid isomerisation between more than one state, i.e., K_D_ = K_Observed_) – as would normally be expected and, in which case, we completely agree with the reviewer that a comparison of efficacy (tau) values alone is sufficient for the quantification of biased agonism and for interpreting the classic ternary complex. However, there are some examples of where the estimated affinity of an agonist for a specific pathway is *significantly* different between biochemical and cell-based assays – in which case a comparison of tau values alone would yield errors in bias estimates if a single affinity value is assumed/used in calculations. For example, please see Gregory et al., 2010 *JBC*, 285: 7459 and Davey et al., 2012, *Endocrinology*, 153: 1232. In the former paper, the estimated affinity of the agonist, McN-A-343 for the M_2_ muscarinic receptor based on biochemical (radioligand binding) assays is approx.10-fold lower than any functional estimate of its affinity across different pathways. In the latter paper, the estimated affinity (pK_B_) of the PAM, cinacalcet, is identical when determined in cellular assays of intracellular calcium mobilization or downstream pERK1/2 (pK_B_ = 6.73), but significantly different (pK_B_ = 8.14) in an assay of plasma membrane ruffling – all determined in the same cellular background. Although the mechanisms underlying these observations remain unclear, they nonetheless suggest that GPCRs can sometimes adopt a different affinity state depending on the signaling pathway that is retained over the time course of a given experiment (e.g., via compartmentalisation), such that a single affinity value that would normally represent rapid isomerization of the ground state between any number of different states would fail to account for the data on the basis of tau values alone.

It is for this latter reason that we choose to use the tau/K_A_ ratio, rather than comparison of tau values alone, as the starting point for quantifying bias – while also using a reference agonist across different pathways. Based on the method for estimating bias as described in Kenakin et al., 2012, *ACS Chem. Neurosci.* 3: 193 and summarized in Kenakin and Christopoulos, 2013, *Nat. Rev. Drug Discovery*, 12: 205, if the affinity values of the agonists are identical between pathways (as would be expected in most instances according to classic receptor theory) then the method essentially becomes a comparison of tau values. However, if this assumption is *not* met, then errors will arise using tau values alone, but not when using tau/K_A_ ratios (Kenakin and Christopoulos, 2013, *Nat. Rev. Drug Discovery*, 12: 205).

Nonetheless, in order to better address this issue and the reviewer’s concern, we have re-written the original description in the Results of the transducer coupling coefficient (original lines 177 – 181) to more explicitly define the meaning of this parameter, as well as addressing the ambiguity in our original statement that this parameter ‘…accounts for the ground state binding affinity of the agonist….and characterises the agonism of a specific pathway defined as the interaction between agonist, receptor and transducer…etc.’ to state the following:

“These signaling assays allowed us to determine the efficacy of the agonists (τA) and the PAMs (τB) (Figure 1H, Figure 1 —figure supplement 2B). Importantly, efficacy (τ ), as defined from the Black-Leff operational model of agonism (Black and Leff, 1983), is determined by the ability of an agonist to promote an active receptor conformation, the receptor density (Bmax), and the subsequent ability of a cellular system to generate a response (Figure 1B). Notably, in both signalling assays, the rank order of efficacy was ACh > Ipx > LY298 > VU154. We subsequently calculated the transducer coupling coefficient (τ/K) (Figure 1I, Figure 1 —figure supplement 2C), a parameter often used as a starting point to quantify biased agonism (Kenakin et al., 2012) and that is specific to the intact cellular environment in which a given response occurs. Thus the dissociation constant (K) in the transduction coefficient subsumes the affinity for the ground state (non-bound) receptor, in addition to any isomerisation states of the receptor that ultimately yield cellular responses (Kenakin and Christopoulos, 2013).”

We also totally agree with the reviewer’s comment regarding the transduction coefficients of isoproterenol (high efficacy) and pindolol (low efficacy) but, nonetheless, using these values as a starting point in the ultimate quantification of bias (not efficacy) of these ligands between pathways would still yield the correct result.

In terms of the second concern, we agree with the reviewer that our suggestion that the transducer coupling coefficient is a ‘better’ representation of efficacy in an atomic resolution structure of the ternary complex than, e.g., the tau parameter alone is likely an overstatement. We have thus removed this statement altogether and re-written that section of the Discussion accordingly.

We also agree with the reviewer that most GPCR structures to date suggest that the observed conformation is largely driven by the bound transducer, but feel that the field is still ‘relatively’ nascent in a number of key areas: for instance, the fact that many important structures (e.g., unliganded states; antagonist-bound receptor-transducer states; PAM-bound, agonist-free states, etc.) have not been readily solved; the fact that (as shown in our MD simulations) the dynamics of the complex must also be incorporated into the understanding of the mechanisms governing stability of any observed structure and, ultimately, the need to strive to improve our ability to link solved structures to biochemistry and to intact cell pharmacology. It is for this reason that we have chosen to include transducer coupling coefficient estimates, in addition to independent affinity and efficacy (tau) estimates, because both parameters play different roles in ultimately defining the stability and signaling capacity of a given complex; the transducer coupling coefficient is another means of reinforcing this point.

Figure 6. It would be helpful to label some of the microswitches in the figure, as it may not be obvious to all readers, e.g., me, what specific residues they are looking at.

We thank the reviewer for this suggestion, and have labelled the microswitches in this figure accordingly.

Figure S2. Any comment on the different populations of complexes with the PAMs? It is interesting that VU154 induces a large population of presumably a different conformation.

Figure S2 shows the cryo-EM data processing workflow that was done for each complex. The 3D classification doesn’t necessarily reflect differences in conformations or populations of receptor complexes. The “different populations” were likely due to ‘bad particles’ (i.e., receptor complexes) that arise from differences in the purification, stability of the complexes, vitrification, imaging, and processing of the samples.

- Consider focusing on log tau as a readout of efficacy as opposed to the transduction coefficients, which have limited utility when comparing agonists.

In general, throughout the text we have now focused more on efficacy, and in Figure 7C we have replaced the transduction coefficients with efficacy estimates alone.

- A sentence or two on limitations of the cryoEM approach in studying agonism due to the stabilization of the A-R-T complex.

We have amended the discussion to include this limitation:

“It is worth noting that structures of GPCRs bound to agonists with different pharmacological properties (full, partial, and biased agonists) have now been reported for some GPCRs (Liang et al., 2018a; Masureel et al., 2018; McCorvy et al., 2018; Ring et al., 2013; Wacker et al., 2013; Warne et al., 2012; Wingler et al., 2019). However, insights gained from such cryo-EM and x-ray crystallography structures may be limited, due to the role that the bound transducer plays on the observed final receptor conformation, and not necessarily due solely to the properties of the ligand. The ultimate underlying conformational differences, therefore, are likely to be subtle and dynamic (Seyedabadi et al., 2022), requiring application of additional techniques such as NMR spectroscopy, single molecule FRET and MD simulations for furthering our understanding (Cao et al., 2021; Cong et al., 2021; Gregorio et al., 2017; Huang et al., 2021; Katayama et al., 2021; J. J. Liu et al., 2012; Solt et al., 2017; Sušac et al., 2018; Xu et al., 2023; Ye et al., 2016, p. 19).”

Reviewer #2 (Recommendations for the authors):1. The abstract focuses on the questions and the methods used to address them, with relatively little in terms of specific details about what is found. Simply saying the work offers "in-depth insights", while true, is a bit vague and leaves the reader guessing. Some more specific statements about key findings (e.g., the superior coupling to G protein induced by ACh) would be appreciated.

We appreciate the reviewer’s concerns and have amended the abstract to more explicitly highlight the key findings while also trying to remain within the constraints of abstract length.

“Allosteric modulation of G protein-coupled receptors (GPCRs) is a major paradigm in drug discovery. Despite decades of research, a molecular-level understanding of the general principles that govern the myriad pharmacological effects exerted by GPCR allosteric modulators remains limited. The M4 muscarinic acetylcholine receptor (M4 mAChR) is a validated and clinically relevant allosteric drug target for several major psychiatric and cognitive disorders. In this study, we rigorously quantified the affinity, efficacy and magnitude of modulation of two different positive allosteric modulators, LY2033298 (LY298) and VU0467154 (VU154), combined with the endogenous agonist acetylcholine (ACh) or the high-affinity agonist iperoxo (Ipx), at the human M4 mAChR. By determining the cryo-electron microscopy (cryo-EM) structures of the M4 mAChR, bound to a cognate Gi1 protein and in complex with ACh, Ipx, LY298-Ipx, and VU154-Ipx, and applying molecular dynamics (MD) simulations we determine key molecular mechanisms underlying allosteric pharmacology. In addition to delineating the contribution of spatially distinct binding sites on observed pharmacology, our findings also revealed a vital role for orthosteric and allosteric ligand-receptor-transducer complex stability, mediated by conformational dynamics between these sites, in the ultimate determination of affinity, efficacy, cooperativity, probe-dependence and species variability. There results provide a holistic framework for further GPCR mechanistic studies and can aid the discovery and design of future allosteric drugs.”

2. Figure 2 panels B-E and G-J might be made larger, and the data points for the other panels smaller, for clarity. The data points obscure error bars in all but a few cases.

We have decreased the size of the data points and increased the thickness of the error bars. We have increased the size of panels (B-E, G-J) and moved the concentration response curves to supplemental: Figure 1 —figure supplement 1.

3. Figure 3 shows a huge amount of structural data, but this means all panels in the figure are very small. The authors may want to consider a more focused presentation, with fewer essential components left for the SI. Panels A-C are somewhat redundant, for example, and a single representative example might suffice.

We now only show the M4R-Gi-Ipx panel as a single representative example. The remaining structures and G protein comparison have been moved to Figure 2 —figure supplement 1.

4. The discussion of Wang et al. 2022 and the corresponding figure S5 are very important. Figure S5 was difficult to interpret and see clearly, perhaps in part due to image compression by the journal. Even so, a more focused presentation would still be helpful, ideally with a finer map mesh. A transparent surface view may be even easier to interpret.

We thank the reviewer for suggesting a transparent surface view. Indeed, this is easier to interpret. We have thus removed one set of the panels, allowing an increase in the size of the others.

5. The discussion is clearly written and helpful, but reads as being targeted to a specialist pharmacology audience. Including more of a broad perspective and relating this to allostery in other GPCRs and other protein families could enhance the overall breadth of appeal.

We have trimmed and amended the discussion such that it is balanced towards a broader audience.

6. The final sentence about signaling bias (line 624) is important, and the authors may want to state explicitly that the structural and mechanistic basis for biased signaling (and allostery as well) is likely to vary from one receptor to another. This is implied, but if the authors feel justified in making a stronger statement it may be helpful for clarity and emphasis.

We have added this statement to a section describing the limitations of the study.

Reviewer #3 (Recommendations for the authors):1. I would propose merging Figures 1 and 2 to clarify which aspect is covered for each experiment in Figure 2. It would be great to use a color code (or any other sort of label) for each parameter (K, α, etc.) and highlight the respective plots (including supplements), for readers to follow the discussion.

We have merged Figures 1 and 2 together and included the chemical structures of the compounds into Figure 1. The concentration response curves have been moved to Figure 1 —figure supplement 1 and 2. We’ve experimented with colour coding, but have not found a way to do this with sufficient clarity. Instead, we’ve linked the parameters in panel B to their respective panels below in the text of the schematic.

2. It would also be helpful to show the structures of all four compounds in the main text, for clarity. These could be used to highlight which parts of the ligands were observed in the structure, vs. which ones were disordered (i.e. which bonds are rotating freely), as discussed for both Ipx and VU154.

Chemical structures have been included in Figure 1. The cryo-EM density surrounding the ligands at an appropriate contour level have been added to Figure 2.

3. All pharmacological experiments have been performed with a full-length wild type, however, the cryoEM structure contains a major deletion of ICL3. While I understand that previous experiments have used similar constructs for structure determination, I believe it would be helpful to confirm binding affinities, as well as the efficacy of the respective drugs with the truncated cryoEM construct. If this is not feasible, please highlight the caveat that all structures, and consequently all MD simulations, are based on data obtained from an engineered construct, rather than the wildtype sequence.

We agree that this is an important caveat, and have added a comment to this effect in a ‘limitations’ section at the end of the manuscript.

4. Line 190: What is a "NAL" (i presume it should spell 'NAM')? Please also discuss the impact of this observation.

We apologize for the use of this abbreviation without context. A ‘NAL’ is a ‘neutral allosteric ligand’ i.e., a ligand that occupies the allosteric site but has *neutral* cooperativity (ab = 1) with the orthosteric ligand (but can still compete with other allosteric ligands binding to the same site – see Christopoulos et al., 2014, *Pharmacol. Rev.*66: 918). We have thus amended the sentence to explicitly state:

“on this basis, VU154 would be classified as a *‘neutral’ allosteric ligand* (not a PAM) with Ipx in the TruPath assay, i.e., VU154 still binds to the allosteric site, but displays neutral cooperativity (ab=1) with Ipx”.

5. The pharmacological experiments were conducted with all possible combinations of ligands (ACh, Ipx with LY298, VU154). I understand that it could have been merely a factor of resources, but were any attempts made to elucidate structures of the PAM-complexes with ACh?

Ipx was the initial ligand of choice for the structural studies, largely due to its high affinity and prior use in structural studies of the M_2_ mAChR. Structures were not attempted with ACh and LY298/VU154 due to limited cryo-EM resources.

6. It is not clear to the reader, whether the above pharmacological experiments (Figure 2) have been carried out for the very first time, or if others have attempted similar studies. Please clarify exactly what part of the work is novel.

We apologize for lack of clarity. The pharmacology of LY298 with ACh has been relatively well characterized in binding and functional assays by us and other groups as has, albeit to a lesser extent, the pharmacology of VU154. As far as we are aware, however, the TruPath data are new for both ligands ,as well as the effects of VU154 on pERK1/2. Moreover, the pharmacological effects of LY298 and VU154 on Ipx have not been reported to our knowledge. We’ve added the following lines to the manuscript to make this issue clearer:

“The pharmacology of LY298 or VU154 interacting with ACh have been well characterized in binding and functional assays at the M4 mAChR (Bubser et al., 2014; Chan et al., 2008; Gould et al., 2016; Leach et al., 2010; Suratman et al., 2011; Thal et al., 2016). However, their pharmacology with Ipx has not been reported. Therefore, we characterized both PAMs with ACh and Ipx in binding and in two different functional assays to provide a thorough foundational comparative characterization of the pharmacological parameters of these ligands from the same study.”

7. Figure 2A: the fact that increasing concentrations of LY298 appear to block the overall binding of ACh is not described or discussed anywhere. Based on this plot LY298 would be a PAM-antagonist (see for example Figure 1, Grundmann et al. 2021, 10.3390/ijms22041763). This would be an important aspect, which would need to be addressed.

We apologise, but the reviewer has misinterpreted these data. The experiments in figure 2A describe a three-way interaction between a fixed concentration of radiolabeled orthosteric *antagonist*, [^3^H]-NMS, and increasing concentrations of unlabeled orthosteric *agonist* (ACh) and allosteric modulator, LY298. What is actually occurring, therefore, is that the reduced overall specific binding ‘window’ *is due to negative cooperativity between LY298 and the radiolabel, [^3^H]NMS* (because the antagonist prefers the inactive state, whereas LY298 prefers the active state). *At the same time*, the potency of the ACh competition curve is actually increasing because the modulator is increasing ACh affinity (positive cooperativity) i.e., it is not ‘blocking the overall binding of ACh’ but, rather, *it is actually blocking the overall binding of [^3^H]NMS*. This is an example of ‘probe dependence’, whereby the same modulator can have a different allosteric effect depending on the orthosteric ligand (e.g., antagonist vs agonist). Thus, LY298 is *not* a PAM-antagonist; it is a *NAM* of [^3^H]NMS and a *PAM* of ACh; a true ‘PAM-antagonist’ is a modulator that increases agonist affinity while concomitantly reducing agonist signaling efficacy (e.g., see Price et al., 2005, Mol. Pharm. 67: 1484), which is not the case here (indeed, binding assays alone cannot be used to determine whether a compound is a PAM-antagonist). We have added the following line to the main text:

“A probe-dependent effect was also observed with the radioligand, [3H]-NMS, evidenced by a reduction in specific radioligand binding due to negative cooperativity between the antagonist probe and LY298, which has been previously reported (Chan et al., 2008; Leach et al., 2010; Suratman et al., 2011; Thal et al., 2016).”

8. I would suggest placing some of the MD simulation traces into the supplementary materials, as these currently take up a large fraction of all figures. Alternatively, different complexes could be color coded and overlaid in one figure to highlight differences.

We agree and have moved some of the MD simulation traces to supplemental (e.g. Figure 3 —figure supplement 1; figure 4 —figure supplement 1).

9. Figure 4 B and H: given the structure in Figure 4 H, it means that the binding pocket around Ipx leaves no room for any movement. Overall, I am not able to follow the discussion regarding the alkyne group/linker of Ipx not being visible. If the start and end points are fixed (visible densities), and have a rigid, planar triple bond involved (alkyne), I find it hard to imagine that the linker is flexible enough to wash out the signal for the linker. Also, the representative Figure S3F is not very convincing, as (A) all iperoxo densities seem to be of rather poor quality and (B) the average of all aligned structures would still likely result in an 'average-able' linker density. I would suggest either elaborating on or omitting this claim.

We apologise for the ambiguity here. The trimethyl ammonium ion is making a cation-pi interaction with nearby residues. The relative orientation of the “ion” does not need to be specific relative to the receptor, as the interaction is driven by the position of the charge, which overlays nicely across the structures. There is enough room in the active site for the alkyne bond to rotate circularly around the position of the ion (similar to a ‘crankshaft’ rotating) and the effect of this would be a blurring of the density around this bond. Nevertheless, we acknowledge the esoteric nature of this point and that it is not necessarily relevant to the main text; as such we have removed most of this statement from the text and supplemental figure.

10. Line 301, The predominate χ2 angle of W413 was approximately 60◦ and 105◦ in the ACh-bound and Ipx-bound simulations, respectively, corresponding to the cryo-EM conformations. As depicted in Figure 4L, W413 when bound to ACh also samples angles close to 90 degrees or higher in the majority of Sim2 and part of Sim1. What is the significance of this conformational sampling?

Ipx stabilizes the active conformation/rotamer of W413 (~105◦) by directly interacting with the residue, whereas ACh is too small to directly interact with W413. As such, the conformation of W413 matches the inactive conformation (60◦) that was observed in the tiotropium-bound structure. As commented, during GaMD simulations with ACh, the conformation of W413 appears to fluctuate between the inactive and active conformations. One may speculate that this could relate to differences in efficacy between ACh and Ipx, as the residue is known to be important for biased signaling at some GPCRs (Cong et al., *Molecular Cell* 2021). Indeed, we have studied residue W413 in more detail in pERK1/2 signalling assays and the G_i1_ TruPath assay –data that was not included in the original submitted version of this manuscript, but now added back to a new section: “Structural insights into allosteric modulation of agonist signaling” as we believe that these data are important to the overall manuscript.

11. On the note of ligands having flexible parts in VU154, and therefore no resolved densities in the maps, is there any pharmacological evidence (i.e. SAR) for these regions not contributing to binding/signaling? Analogously, are there any SAR data for replacing the alkyne bond in Ipx? It would be conceivable that a rigid replacement linker would affect (either positively or negatively) receptor binding and signaling.

We have not thoroughly investigated the SAR around VU154 and other M4-PAMs. We agree that this is an interesting idea, but beyond the scope of the current manuscript. Instead, we looked at the mouse receptor, where we knew VU154 had better PAM-agonist activity. It is plausible that the unresolved / flexible region of VU154, which is negatively charged, is able to interact with an Arg residue that is present at the mouse receptor and not at the human receptor (Figure 7) – but this would require a cryo-EM structure with the mouse receptor for proper validation. With respect to Ipx, we note that it is one of the most potent agonists discovered for mAChRs to date (Schrage et al. *BJP* 2013) and has ‘supraphysiological’ activity, i.e., higher intrinsic efficacy than the cognate agonist, ACh. As such, we would expect analogs that removed the linker to have lower potency and binding, but we have not investigated this to date.

12. According to the GaMD simulation results in Figure S7C, the minimum distance for the T433 and VU154 seems to be close to 4 Å while the predominant distance is around 7 Å. Therefore, I am wondering what the significance of this hydrogen bond is observed in the cryoEM structure. Additionally, T433R mutant showed increased binding to VU154 and showed the importance of T433 in species selectivity. Is this increased binding of VU154 with T433R mutant a result of a more "stable" hydrogen bond between receptor and VU154?

We apologize for not being clear here. We meant to imply there were potential interactions between VU154 and residue T433 based on inspection of the structure. However, during GaMD simulations these distances were too far apart and highly fluctuating suggesting the interaction would be transient at best. We have amended the manuscript as such:

“For VU154, there were two additional hydrogen bonding interactions were possible with residues Y922.64 and T4337.33 (Figure 4G); however, these interactions were highly fluctuating during GaMD simulations suggesting they were at best transient interactions (Figure 4 —figure supplement 1I,J).

With respect to the T433R mutant we apologize for not including that data. GaMD simulations suggest the interaction between residue R433 and the sulfoxide group of VU154 is more stable (5.2 ± 1.5 Å) vs the WT residue T433 (6.56 ± 2.1 Å), but the interaction distance is still a bit far. Ultimately, this is an interaction that would be better confirmed by structure as suggested above. The GaMD data for T433R has been added to X and included in the text as:

“The GaMD simulations also suggest that a potential interaction between the mutant residue T433R and the sulfoxide group of VU154 were more stable (5.2 ± 1.5 Å; Figure 7 —figure supplement 3I) vs the WT residue T433 (6.56 ± 2.1 Å, Figure 4 —figure supplement 1J), albeit the distance of this interaction far apart and would be better validated by structure determination of VU154 with the mouse M4 mAChR.”

13. Regarding the species selectivity aspect, there is no mention of the V91L mutant in Figure 7, only as part of the triple mutant. It is hard to judge which mutations are responsible for species selectivity without either showing results for the D432E/T433R double mutant, or the additional V91L single mutant.

We do apologize for not having data regarding either V91L or a D432E/T433R double mutant. This is a limitation of the study that we highlight in the manuscript text. However, V91 does not face the allosteric site and note that the V to L mutation is conserved; we thus do not expect this residue to have a strong role in species selectivity. Hence, the biggest contributors are likely to be D432E and T433R.